# Phased Consistency Models

Fu-Yun Wang[1] Zhaoyang Huang[2] Alexander William Bergman[3,6] Dazhong Shen[4]

Peng Gao[4] Michael Lingelbach[3,6] Keqiang Sun[1] Weikang Bian[1]

Guanglu Song[5] Yu Liu[4] Xiaogang Wang[1] Hongsheng Li[1,4,7]

[1]MMLab, CUHK   [2]Avolution AI   [3]Hedra

[4]Shanghai AI Lab   [5]Sensetime Research   [6]Stanford University   [7]CPII under InnoHK

{fywang@link, xgwang@ee, hsli@ee}.cuhk.edu.hk

## Abstract

Consistency Models (CMs) have made significant progress in accelerating the generation of diffusion models. However, their application to high-resolution, text-conditioned image generation in the latent space remains unsatisfactory. In this paper, we identify three key flaws in the current design of Latent Consistency Models (LCMs). We investigate the reasons behind these limitations and propose **P**hased **C**onsistency **M**odels (PCMs), which generalize the design space and address the identified limitations. Our evaluations demonstrate that PCMs outperform LCMs across 1–16 step generation settings. While PCMs are specifically designed for multi-step refinement, they achieve comparable 1-step generation results to previously state-of-the-art specifically designed 1-step methods. Furthermore, we show the methodology of PCMs is versatile and applicable to video generation, enabling us to train the state-of-the-art few-step text-to-video generator. Our code is available at `https://github.com/G-U-N/Phased-Consistency-Model`.

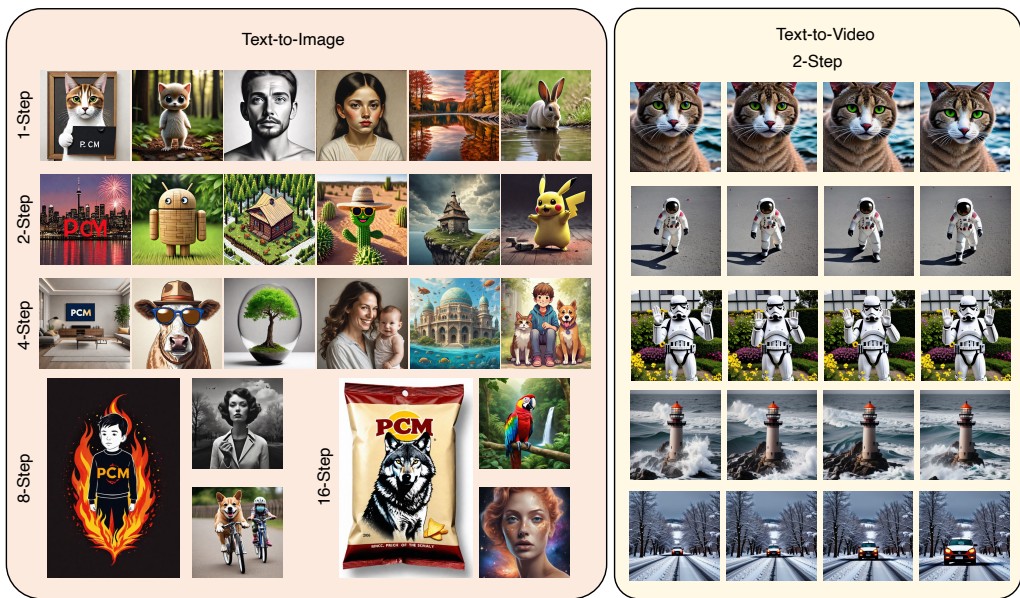

Figure 1: *PCMs: Towards stable and fast image and video generation.*

38th Conference on Neural Information Processing Systems (NeurIPS 2024).

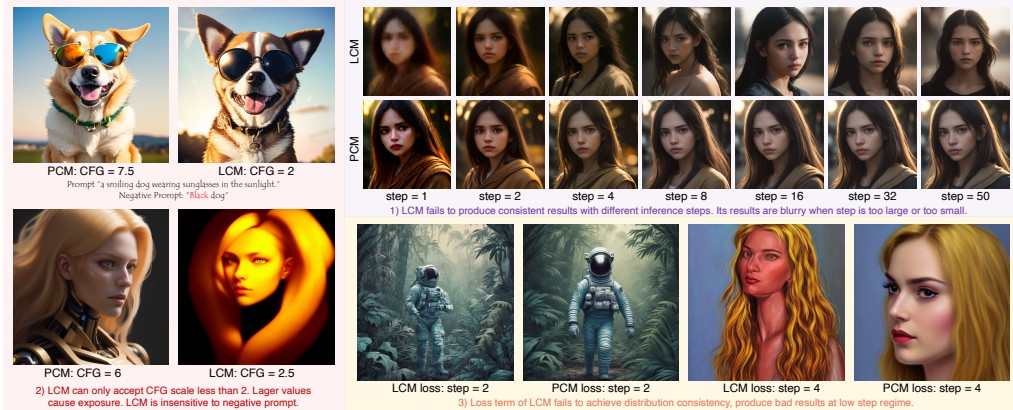

Figure 2: **Summative motivation.** We observe and summarize three crucial limitations for (latent) consistency models, and generalize the design space, well tackling all these limitations.

# 1 Introduction

Diffusion models [20, 15, 57, 68] have emerged as the dominant methodology for image synthesis [45, 41, 7] and video synthesis [22, 53, 52, 63]. These models have shown the ability to generate high-quality and diverse samples conditioned on varying signals. At their core, diffusion models rely on an iterative evaluation to generate new samples. This iterative evaluation trajectory models the probability flow ODE (PF-ODE) [56, 57] that transforms an initial normal distribution to a target real data distribution. However, the iterative nature of diffusion models makes the generation of new samples time-intensive and resource-consuming. To address this challenge, consistency models [56, 11, 63, 55] have emerged to reduce the number of iterative steps required to generate samples. These models work by training a model that enforces the self-consistency [56] property: *any point along the same PF-ODE trajectory shall be mapped to the same solution point.* These models have been extended to high-resolution text-to-image synthesis with latent consistency models (LCMs) [35]. Despite the improvements in efficiency and the ability to generate samples in a few steps, the sample quality of such models is still limited.

We show that the current design of LCMs is flawed, causing inevitable drawbacks in controllability, consistency, and efficiency during image sampling. Fig. 2 illustrates our observations of LCMs. The limitations are listed as follows:

(1) **Consistency.** Due to the specific consistency property, CMs can only use the purely stochastic multi-step sampling algorithm, which assumes that the accumulated noise variable in each generative step is independent and causes varying degrees of stochasticity for different inference-step settings. As a result, we can find inconsistency among the samples generated with the same seeds in different inference steps.

(2) **Controllability.** Even though diffusion models can adopt classifier-free guidance (CFG) [16] in a wide range of values (i.e. 2–15), equipped with weights of LCMs, they can only accept values of CFG within range of 1–2. Larger values of CFG would cause the exposure problem. This brings difficulty for the hyper-parameter selection. Additionally, we find that LCMs are insensitive to the negative prompt. As shown in the figure, LCMs still generate black dogs even when the negative prompt is set to "black dog". Both phenomenon reduce the controllability on generation. We show the reason behind this is the CFG-augmented ODE solver adopted in the consistency distillation stage.

(3) **Efficiency.** LCMs tend to generate much inferior samples at the few-step settings, especially in less than 4 inference steps, which limits the sampling efficiency. We argue that the reason lies in the traditional L2 loss or the Huber loss used in the LCMs procedure, which is insufficient for the fine-grained supervision in few-step settings.

To this end, we propose Phased Consistency Models (PCMs), which can tackle the discussed limitations of LCMs and are easy to train. Specifically, instead of mapping all points along the ODE trajectory to the same solution, PCMs phase the ODE trajectory into several sub-trajectories and only enforce the self-consistency property on each sub-trajectory. Therefore, PCMs can sample

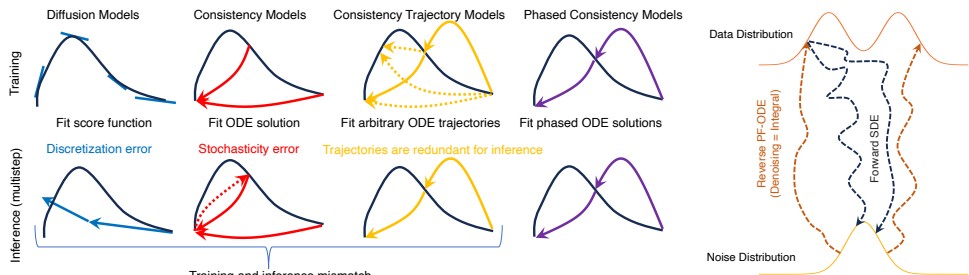

Figure 3: (Left) Illustrative comparison of diffusion models [15], consistency models [56], consistency trajectory models [21], and our phased consistency model. (Right) Simplified visualization of the forward SDE and reverse-time PF-ODE trajectories.

samples along the solution points of different sub-trajectories in a deterministic manner without error accumulation. As an example shown in Fig. 3, we train PCMs with two sub-trajectories, with three edge points including $\mathbf{x}_T$, $\mathbf{x}_0$, and $\mathbf{x}_{\lfloor T/2 \rfloor}$ selected. Thereby, we can achieve 2-step deterministic sampling (i.e., $\mathbf{x}_T \to \mathbf{x}_{\lfloor T/2 \rfloor} \to \mathbf{x}_0$ ) for generation. Moreover, the phased nature of PCMs leads to an additional advantage. For distillation, we can choose to use a normal ODE solver without the CFG alternatively in the consistency distillation stage, which is not viable in LCMs. As a result, PCMs can optionally use larger values of CFG for inference and be more responsive to the negative prompt. Additionally, to improve the sample quality for few-step settings, we propose an adversarial loss in the latent space for more fine-grained supervision.

To conclude, we dive into the design of (latent) consistency models, analyze the reasons for their unsatisfactory generation characteristics, and propose effective strategies for tackling these limitations. We validate the effectiveness of PCMs on widely recognized image generation benchmarks with stable diffusion v1-5 (0.9 B) [45] and stable diffusion XL (3B) [41] and video generation benchmarks following AnimateLCM [63]. Vast experimental results show the effectiveness of PCMs.

## 2 Preliminaries

**Diffusion models** define a forward conditional probability path, with a general representation of $\alpha_t \mathbf{x}_0 + \sigma_t \boldsymbol{\epsilon}$ for intermediate distribution $\mathbb{P}_t(\mathbf{x}|\mathbf{x}_0)$ conditioned on $\mathbf{x}_0 \sim \mathbb{P}_0$, which is equivalent to the stochastic differential equation $\mathrm{d}\mathbf{x}_t = f_t \mathbf{x}_t \mathrm{d}t + g_t \mathrm{d}\boldsymbol{w}_t$ with $\boldsymbol{w}_t$ denoting the standard Winer process, $f_t = \frac{\mathrm{d}\log\alpha_t}{\mathrm{d}t}$ and $g_t^2 = \frac{\mathrm{d}\sigma_t^2}{\mathrm{d}t} - 2\frac{\mathrm{d}\log\alpha_t}{\mathrm{d}t}\sigma_t^2$. There exists a deterministic flow for reversing the transition, which is represented by the probability flow ODE (PF-ODE) $\mathrm{d}\mathbf{x}_t = \left[ f_t \mathbf{x}_t - g_t^2/2 \nabla_{\mathbf{x}_t} \log \mathbb{P}_t(\mathbf{x}_t) \right] \mathrm{d}t$. In standard diffusion training, a neural network is typically learned to estimate the score of marginal distribution $\mathbb{P}_t(\mathbf{x}_t)$, which is equivalent to $\boldsymbol{s}_\phi(\mathbf{x},t) \approx \nabla_\mathbf{x} \log \mathbb{P}_t(\mathbf{x}) = \mathbb{E}_{\mathbf{x}_0 \sim \mathbb{P}(\mathbf{x}_0|\mathbf{x})} \left[ \nabla_\mathbf{x} \log \mathbb{P}_t(\mathbf{x}|\mathbf{x}_0) \right]$. Substitue the $\nabla_\mathbf{x} \log \mathbb{P}_t(x)$ with $\boldsymbol{s}_\phi(x,t)$, and we get the empirical PF-ODE. Famous solvers including DDIM [54], DPM-solver [33], Euler and Heun [20] can be generally perceived as the approximation of the PF-ODE with specific orders and forms of diffusion for sampling in finite discrete inference steps. However, when the number of discrete steps is too small, they face inevitable discretization errors.

**Consistency models** [56], instead of estimating the score of marginal distributions, learn to directly predict the solution point of ODE trajectory by enforcing the self-consistency property: all points at the same ODE trajectory map to the same solution point. To be specific, given a ODE trajectory $\{\mathbf{x}_t\}_{t\in[\epsilon,T]}$, the consistency models $\boldsymbol{f}_\theta(\cdot, t)$ learns to achieve $\boldsymbol{f}_\theta(\mathbf{x}_t, t) = \mathbf{x}_\epsilon$ by enforcing $\boldsymbol{f}_\theta(\mathbf{x}_t, t) = \boldsymbol{f}_\theta(\mathbf{x}_{t'}, t')$ for all $t, t' \in [\epsilon, T]$. A boundary condition $\boldsymbol{f}_\theta(\mathbf{x}_\epsilon, \epsilon) = \mathbf{x}_\epsilon$ is set to guarantee the successful convergence of consistency training. During the sampling procedure, we first sample from the initial distribution $\mathbf{x}_T \sim \mathcal{N}(\mathbf{0}, \mathbf{I})$. Then, the final sample can be generated/refined by alternating denoising and noise injection steps, i.e,

$$\hat{\mathbf{x}}_\epsilon^{\tau_k} = \boldsymbol{f}_\theta(\hat{\mathbf{x}}_{\tau_k}, \tau_k), \ \ \hat{\mathbf{x}}_{\tau_{k-1}} = \hat{\mathbf{x}}_\epsilon^{\tau_k} + \boldsymbol{\eta}, \ \ \boldsymbol{\eta} \sim \mathcal{N}(\mathbf{0}, \mathbf{I}) \tag{1}$$

where $\{\boldsymbol{\tau}_k\}_{k=1}^K$ are selected time points in the ODE trajectory and $\tau_K = T$. Actually, because only one solution point is used to conduct the consistency model, it is inevitable to introduce stochastic error, i.e., $\boldsymbol{\eta}$, for multiple sampling procedures.

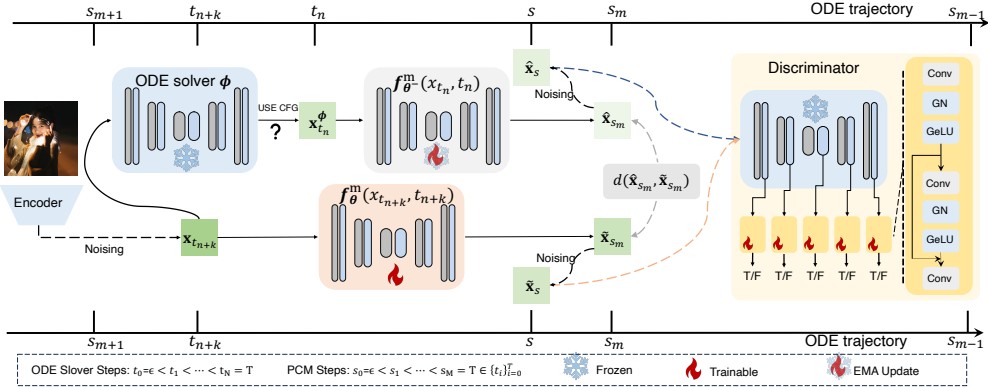

Figure 4: **Training paradigm of PCMs**. '?' means optional usage.

Recently, Consistency trajectory models (CTMs) [21] propose a more flexible framework. specifically, CTMs $f_\theta(\cdot, t, s)$ learns to achieve $f_\theta(\mathbf{x}_t, t, s) = f(x_t, t, s; \phi)$ by enforcing $f_\theta(\mathbf{x}_t, t, s) = f_\theta(\mathbf{x}_{t'}, t', s)$ with $s \leq \min(t, t')$ and $t, t' \in [\epsilon, T]$. However, the learning objectives of CTMs are redundant, including many trajectories that will never be applied for inference. More specifically, if we split the continuous time trajectory into $N$ discrete points, diffusion models learn $\mathcal{O}(N)$ objectives (i.e., each point learns to move to its adjacent point), consistency models learn $\mathcal{O}(N)$ objectives (i.e., each point learns to move to solution point), and consistency trajectory models learn $\mathcal{O}(N^2)$ objectives (i.e., each point learns to move to all the other points in the trajectory). Hence, except for the current timestep embedding, CTMs should additionally learn a target timestep embedding, which is not comprised of the design space of diffusion models.

Different from the above approaches, PCMs can be optimized efficiently and support deterministic sampling without additional stochastic error. Overall, Fig. 3 illustrates the difference in training and inference processes among diffusion models, consistency models, consistency trajectory models, and phased consistency models.

## 3 Method

In this section, we introduce the technical details of PCMs, which overcome the limitations of LCMs in terms of consistency, controllability, and efficiency. **Consistency**: Specifically, we first introduce the main framework of PCMs, consisting of definition, parameterization, the distillation objective, and the sampling procedure in Sec. 3.1. In particular, by enforcing the self-consistency property in multiple ODE sub-trajectories respectively, PCMs can support deterministic sampling to preserve image consistency with varying inference steps. **Controllability**: Secondly, in Sec. 3.2, to improve the controllability of text guidance, we revisit the potential drawback of the guided distillation adopted in LCMs, and propose to optionally remove the CFG for consistency distillation. **Efficiency**: Thirdly, in Sec. 3.3, to further improve inference efficiency, we introduce an adversarial consistency loss to enforce the modeling of data distribution, which facilitates 1-step generation.

### 3.1 Main Framework

**Definition.** For a solution trajectory of a diffusion model $\{\mathbf{x}_t\}_{t \in [\epsilon, T]}$ following the PF-ODE, we split the trajectory into multiple sub-trajectories with hyper-defined edge timesteps $s_0, s_1, \ldots, s_M$, where $s_0 = \epsilon$ and $s_M = T$. The $M$ sub-trajectories can be represented as $\{\mathbf{x}_t\}_{t \in [s_m, s_{m+1}]}$ with $m = 0, 1, \ldots, M - 1$. We treat each sub-trajectory as an independent CM and define the consistency function as $f^m : (\mathbf{x}_t, t) \to \mathbf{x}_{s_m}, \quad t \in [s_m, s_{m+1}]$. We learn $f_\theta^m$ to estimate $f$ by enforcing the self-consistency property on each sub-trajectory that its outputs are consistent for arbitrary pairs on the same sub-trajectory. Namely, $f^m(\mathbf{x}_t, t) = f^m(\mathbf{x}_{t'}, t')$ for all $t, t' \in [s_m, s_{m+1}]$. Note that the consistency function is only defined on the sub-trajectory. However, for sampling, it is necessary to define a transition from timestep $T$ (i.e., $s_M$) to $\epsilon$ (i.e., $s_0$). Thereby, we defined $f^{m,m'} = f^{m'} \left( \cdots f^{m-2} \left( f^{m-1} \left( f^m (\mathbf{x}_t, t), s_m \right), s_{m-1} \right) \cdots, s_{m'} \right)$ that transforms any point $\mathbf{x}_t$ on $m$-th sub-trajectory to the solution point of $m'$-th trajectory.

**Parameterization.** Following the definition, the corresponding consistency function of each sub-trajectory should satisfy boundary condition $f^m(\mathbf{x}_{s_m}, s_m) = \mathbf{x}_{s_m}$, which is crucial for guaranteeing

the successful training of consistency models. To satisfy the boundary condition, we typically need to explicitly parameterize $\boldsymbol{f}_{\boldsymbol{\theta}}^m(\mathbf{x}, t)$ as $\boldsymbol{f}_{\boldsymbol{\theta}}^m(\mathbf{x}_t, t) = c_{\text{skip}}^m(t)\mathbf{x}_r + c_{\text{out}}^m(t)F_{\boldsymbol{\theta}}(\mathbf{x}_t, t, s_m)$, where $c_{\text{skip}}^m(t)$ gradually increases to 1 and $c_{\text{out}}^m(t)$ gradually decays to 0 from timestep $s_{m+1}$ to $s_m$.

Another important thing is how to parameterize $F_{\boldsymbol{\theta}}(\mathbf{x}_t, t, s)$, basically it should be able to indicate the target prediction at timestep $s$ given the input $\mathbf{x}$ at timestep $t$. Since we build upon the epsilon prediction models, we hope to maintain the epsilon-prediction learning target.

For the above-discussed PF-ODE, there exists an exact solution [33] from timestep $t$ to $s$

$$\mathbf{x}_s = \frac{\alpha_s}{\alpha_t}\mathbf{x}_t + \alpha_s \int_{\lambda_t}^{\lambda_s} e^{-\lambda}\sigma_{t_\lambda(\lambda)}\nabla \log \mathbb{P}_{t_\lambda(\lambda)}(\mathbf{x}_{t_\lambda(\lambda)})\mathrm{d}\lambda \tag{2}$$

where $\lambda_t = \ln \frac{\alpha_t}{\sigma_t}$ and $t_\lambda$ is a inverse function with $\lambda_t$. The equation shows that the solution of the ODE from $t$ to $s$ is the scaling of $\mathbf{x}_t$ and the weighted sum of scores. Given a epsilon prediction diffusion network $\boldsymbol{\epsilon}_{\boldsymbol{\phi}}(\mathbf{x}, t)$, we can estimate the solution as $\mathbf{x}_s = \frac{\alpha_s}{\alpha_t}\mathbf{x}_t - \alpha_s \int_{\lambda_t}^{\lambda_s} e^{-\lambda}\boldsymbol{\epsilon}_{\boldsymbol{\phi}}(\mathbf{x}_{t_\lambda(\lambda)}, t_\lambda(\lambda))\mathrm{d}\lambda$. However, note that the solution requires knowing the epsilon prediction at each timestep between $t$ and $s$, but consistency modes need to predict the solution with only $\mathbf{x}_t$ available with single network evaluation. Thereby, we parameterize the $F_{\boldsymbol{\theta}}(\mathbf{x}, t, s)$ as following,

$$\mathbf{x}_s = \frac{\alpha_s}{\alpha_t}\mathbf{x}_t - \alpha_s \hat{\boldsymbol{\epsilon}}_{\boldsymbol{\theta}}(\mathbf{x}_t, t) \int_{\lambda_t}^{\lambda_s} e^{-\lambda}\mathrm{d}\lambda \, . \tag{3}$$

One can show that the parameterization has the same format with DDIM $\mathbf{x}_s = \alpha_s(\frac{\mathbf{x}_t - \sigma_t\boldsymbol{\epsilon}_{\boldsymbol{\theta}}(\mathbf{x}_t, t)}{\alpha_t}) + \sigma_s\boldsymbol{\epsilon}_{\boldsymbol{\theta}}(\mathbf{x}_t, t)$ (see Theorem 3). But here we clarify that the parameterization has an intrinsic difference from DDIM. DDIM is the first-order approximation of solution ODE, which works because we assume the linearity of the score in small intervals. This causes the DDIM to degrade dramatically in few-step settings since the linearity is no longer satisfied. Instead, our parameterization is not approximation but exact solution learning. The learning target of $\hat{\boldsymbol{\epsilon}}_{\boldsymbol{\theta}}(\mathbf{x}, t)$ is no more the scaled score $-\sigma_t\nabla_{\mathbf{x}} \log \mathbb{P}_t(\mathbf{x})$ (which epsilon-prediction diffusion models learn to estimate) but $\frac{\int_{\lambda_t}^{\lambda_s} e^{-\lambda}\boldsymbol{\epsilon}_{\boldsymbol{\phi}}(\mathbf{x}_{t_\lambda(\lambda)}, t_\lambda(\lambda))\mathrm{d}\lambda}{\int_{\lambda_t}^{\lambda_s} e^{-\lambda}\mathrm{d}\lambda}$ . Actually, we can define the parameterization in other formats, but we find this format is simple and has a small gap between the original diffusion models. The parameterization of $F_{\boldsymbol{\theta}}$ also allows for a better property that we can drop the introduced $c_{\text{skip}}^m$ and $c_{\text{out}}^m$ in consistency models to ease the complexity of the framework. Note that following Eq. 3, we can get $F_{\boldsymbol{\theta}}(\mathbf{x}_{s_m}, s_m, s_m) = \frac{\alpha_{s_m}}{\alpha_{s_m}}\mathbf{x}_{s_m} - 0 = \mathbf{x}_{s_m}$. Therefore, the boundary condition is already satisfied. Hence, we can simply define $\boldsymbol{f}_{\boldsymbol{\theta}}^m(\mathbf{x}, t) = F_{\boldsymbol{\theta}}(\mathbf{x}, t, s_m)$. This parameterization also aligns with several previous diffusion distillation techniques [46, 3, 74] utilizing DDIM format, building a deep connection with previous distillation methods. The difference is that we provide a more fundamental explanation of the meaning and learning objective of parameterizations.

**Phased consistency distillation objective.** Denote the pre-trained diffusion models as $\boldsymbol{s}_{\boldsymbol{\phi}}(\mathbf{x}, t) = -\frac{\boldsymbol{\epsilon}_{\boldsymbol{\phi}}(\mathbf{x}, t)}{\sigma_t}$, which induces an empirical PF-ODE. We firstly discretize the whole trajectory into $N$ sub-intervals with $N + 1$ discrete timesteps from $[\epsilon, T]$, which we denote as $t_0 = \epsilon < t_1 < t_2 < \cdots < t_N = T$. Typically $N$ should be sufficiently large to make sure the ODE solver approximates the ODE trajectory correctly. Then we sample $M + 1$ timesteps as edge timesteps $s_0 = t_0 < s_1 < s_2 < \cdots < s_M = t_N \in \{t_i\}_{i=0}^N$ to split the ODE trajectory into $M$ sub-trajectories. Each sub-trajectory $[s_i, s_{i+1}]$ consists of the set of sub-intervals $\{[t_j, t_{j+1}]\}_{t_j \geq s_i, t_{j+1} \leq s_{i+1}}$.

Here we define $\Phi(\mathbf{x}_{t_{n+k}}, t_{n+k}, t_n; \boldsymbol{\phi})$ as the $k$-step ODE solver that approximate $\mathbf{x}_{t_n}^{\boldsymbol{\phi}}$ from $\mathbf{x}_{t_{n+k}}$ on the same sub-trajectory following Equation 2, namely,

$$\hat{\mathbf{x}}_{t_n}^{\boldsymbol{\phi}} = \Phi(\mathbf{x}_{t_{n+k}}, t_{n+k}, t_n; \boldsymbol{\phi}). \tag{4}$$

Following CMs [56], we set $k = 1$ to minimize the ODE solver cost. The training loss is defined as

$$\mathcal{L}_{\text{PCM}}(\boldsymbol{\theta}, \boldsymbol{\theta}^-; \boldsymbol{\phi}) = \mathbb{E}_{\mathbb{P}(m), \mathbb{P}(n|m), \mathbb{P}(\mathbf{x}_{t_{n+1}}|n, m)} \left[ \lambda(t_n) d \left( \boldsymbol{f}_{\boldsymbol{\theta}}^m(\mathbf{x}_{t_{n+1}}, t_{n+1}), \boldsymbol{f}_{\boldsymbol{\theta}^-}^m(\hat{\mathbf{x}}_{t_n}^{\boldsymbol{\phi}}, t_n) \right) \right] \tag{5}$$

where $\mathbb{P}(m) := \text{uniform}(\{0, 1, \ldots, M - 1\})$, $\mathbb{P}(n|m) := \text{uniform}(\{n + 1 | t_{n+1} \leq s_{m+1}, t_n \geq s_m\})$, $\mathbb{P}(\mathbf{x}_{t_{n+1}}|n, m) = \mathbb{P}_{t_{n+1}}(\mathbf{x})$, and $\boldsymbol{\theta}^- = \mu\boldsymbol{\theta}^- + (1 - \mu)\boldsymbol{\theta}$.

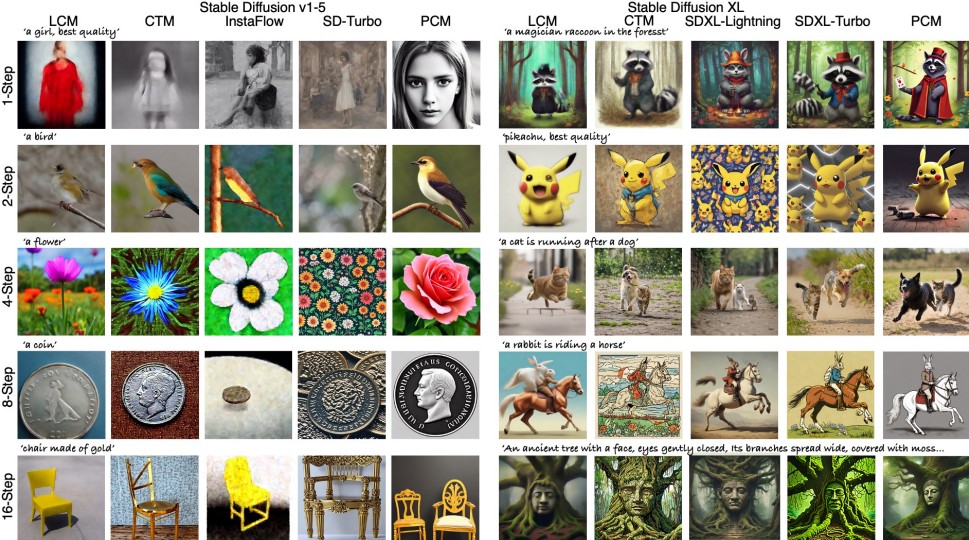

Figure 5: **Qualitative Comparison.** Our method achieves top-tier performance.

We show that when $\mathcal{L}_{\mathrm{PCM}}(\boldsymbol{\theta}, \boldsymbol{\theta}^-; \boldsymbol{\phi}) = 0$ and local errors of ODE solvers uniformly bounded by $\mathcal{O}((\Delta t)^{p+1})$, the solution estimation error within the arbitrary sub-trajectory $\boldsymbol{f}_{\boldsymbol{\theta}}^m(\mathbf{x}_{t_n}, t_n)$ is bounded by $\mathcal{O}((\Delta t)^p)$ in Theorem 1. Additionally, the solution estimation error across any sets of sub-trajectories (i.e., the error between $\boldsymbol{f}_{\boldsymbol{\theta}}^{m,m'}(\mathbf{x}_{t_n}, t_n)$ and $\boldsymbol{f}^{m,m'}(\mathbf{x}_{t_{n+1}, t_{n+1}}; \boldsymbol{\phi})$) is also bounded by $\mathcal{O}((\Delta t)^p)$ in Theorem 2.

**Sampling.** For a given initial sample at timestep $t$ which belongs to the sub-trajectory $[s_m, s_{m+1}]$, we can support deterministic sampling following the definition of $\boldsymbol{f}^{m,0}$. Previous work [21, 67] reveals that introducing a certain degree of stochasticity might lead to better generation quality. We show that our sampling method can also introduce randomness through a simple modification, which we discuss at Sec. IV.1

## 3.2 Guided Distillation

For convenience, we take the epsilon-prediction format with text conditions $\boldsymbol{c}$ for the following discussion. The consistency model is denoted as $\boldsymbol{\epsilon}_{\boldsymbol{\theta}}(\mathbf{x}_t, t, \boldsymbol{c})$, and the diffusion model is denoted as $\boldsymbol{\epsilon}_{\boldsymbol{\phi}}(\mathbf{x}_t, t, \boldsymbol{c})$. A commonly applied strategy for text-conditioned diffusion models is classifier-free guidance (CFG) [16, 51, 1]. At training, $\boldsymbol{c}$ is randomly substituted with null text embedding $\varnothing$. At each inference step, the model computes $\boldsymbol{\epsilon}_{\boldsymbol{\phi}}(\mathbf{x}_t, t, \boldsymbol{c})$ and $\boldsymbol{\epsilon}_{\boldsymbol{\phi}}(\mathbf{x}_t, t, \boldsymbol{c}_{\mathrm{neg}})$ simultaneously, and the actual prediction is the linear combination of them. Namely,

$$\boldsymbol{\epsilon}_{\boldsymbol{\phi}}(\mathbf{x}_t, t, \boldsymbol{c}, \boldsymbol{c}_{\mathrm{neg}}; w) = \boldsymbol{\epsilon}_{\boldsymbol{\phi}}(\mathbf{x}_t, t, \boldsymbol{c}_{\mathrm{neg}}) + w(\boldsymbol{\epsilon}_{\boldsymbol{\phi}}(\mathbf{x}_t, t, \boldsymbol{c}) - \boldsymbol{\epsilon}_{\boldsymbol{\phi}}(\mathbf{x}_t, t, \boldsymbol{c}_{\mathrm{neg}})), \tag{6}$$

where $w$ controlling the strength and $\boldsymbol{c}_{\mathrm{neg}}$ can be set to $\varnothing$ or text embeddings of unwanted characteristics. A noticeable phenomenon is that diffusion models with this strategy can not generate content with good quality without using CFG. That is, the empirical ODE trajectory induced with pure $\boldsymbol{\epsilon}_{\boldsymbol{\phi}}(\mathbf{x}_t, t, \boldsymbol{c})$ deviates away from the ODE trajectory to real data distribution. Thereby, it is necessary to apply CFG-augmented prediction $\boldsymbol{\epsilon}_{\boldsymbol{\phi}}(\mathbf{x}_t, t, \boldsymbol{c}, \boldsymbol{c}_{\mathrm{neg}}; w)$ for ODE solvers. Recall that we have shown that the consistency learning target of the consistency model $\boldsymbol{\epsilon}_{\boldsymbol{\theta}}(\mathbf{x}_t, t, \boldsymbol{c})$ is the weighted sum of epsilon prediction on the trajectory. Thereby, we have $\boldsymbol{\epsilon}_{\boldsymbol{\theta}}(\mathbf{x}_t, t, \boldsymbol{c}) \propto \boldsymbol{\epsilon}_{\boldsymbol{\phi}}(\mathbf{x}_{t'}, t', \boldsymbol{c}, \varnothing; w)$, for all $t' \leq t$. On this basis, if we additionally apply the CFG for the consistency models, we can prove that

$$\boldsymbol{\epsilon}_{\boldsymbol{\theta}}(\mathbf{x}_t, t, \boldsymbol{c}, \boldsymbol{c}_{\mathrm{neg}}; w') \propto ww'(\boldsymbol{\epsilon}_{\boldsymbol{\phi}}(\mathbf{x}_{t'}, t', \boldsymbol{c}) - \boldsymbol{\epsilon}_{\boldsymbol{\phi}}^{\mathrm{merge}})) + \boldsymbol{\epsilon}_{\boldsymbol{\phi}}(\mathbf{x}_{t'}, t', \boldsymbol{c}_{\mathrm{neg}}), \tag{7}$$

where $\boldsymbol{\epsilon}_{\boldsymbol{\phi}}^{\mathrm{merge}} = (1 - \alpha)\boldsymbol{\epsilon}_{\boldsymbol{\phi}}(\mathbf{x}_{t'}, t', \boldsymbol{c}_{\mathrm{neg}}) + \alpha\boldsymbol{\epsilon}_{\boldsymbol{\phi}}(\mathbf{x}_{t'}, t', \varnothing)$ and $\alpha = \frac{(w-1)}{ww'}$ (See Theorem 4). This equation indicates that applying CFG $w'$ to the consistency models trained with CFG-augmented ODE solver confined with $w$, is equivalent to scaling the prediction of original diffusion models by $w'w$, which explains the the exposure problem. We can also observe that the epsilon prediction with negative prompts is *diluted* by the prediction with null text embedding, which reveals that the impact of negative prompts is reduced.

Table 1: Comparison of FID-SD and FID-CLIP with Stable Diffusion v1-5 based methods under different steps.

| METHODS | FID-SD | | | | | | | | | | FID-CLIP | | | | |
| | COCO-30K | | | | | CC12M-30K | | | | | COCO-30K | | | | |
| | 1 | 2 | 4 | 8 | 16 | 1 | 2 | 4 | 8 | 16 | 1 | 2 | 4 | 8 | 16 |
|---|---|---|---|---|---|---|---|---|---|---|---|---|---|---|---|
| InstaFlow [31] | 11.51 | 69.79 | 102.15 | 122.20 | 139.29 | 11.90 | 62.48 | 88.64 | 105.34 | 113.56 | **9.56** | 29.38 | 38.24 | 43.60 | 47.32 |
| SD-Turbo [49] | 10.62 | 12.22 | 16.66 | 24.30 | 30.32 | 10.35 | 12.03 | 15.15 | 19.91 | 23.34 | 12.08 | 15.07 | 15.12 | 14.90 | 15.09 |
| LCM [35] | 53.43 | 11.03 | 6.66 | 6.62 | 7.56 | 42.67 | 10.51 | 6.40 | 5.99 | 9.02 | 21.04 | 10.18 | 10.64 | 12.15 | 13.85 |
| CTM [21] | 63.55 | 9.93 | 9.30 | 15.62 | 21.75 | 28.47 | 8.98 | 8.22 | 13.27 | 17.43 | 30.39 | 10.32 | 10.31 | 11.27 | 12.75 |
| Ours | **8.27** | **9.79** | **5.81** | **5.00** | **4.70** | **7.91** | **8.93** | **4.93** | **3.88** | **3.85** | 11.66 | 9.17 | 9.07 | 9.49 | 10.13 |
| Ours* | - | - | 7.46 | 6.49 | 5.78 | - | - | 4.99 | 5.01 | 5.13 | - | - | **8.85** | **8.33** | **8.09** |

We ask the question: *Is it possible to conduct consistency distillation with diffusion models trained for CFG usage without applying CFG-augmented ODE solver?* Our finding is that it is not applicable for original CMs but works well for PCMs especially when the number of sub-trajectory $M$ is large. We empirically find that $M = 4$ is sufficient for successful training. As we discussed, the text-conditioned diffusion models trained for CFG usage fail at achieving good generation quality when removing CFG for inference. That is, target data distribution induced by PF-ODE with $\epsilon_\phi(\mathbf{x}_t, t, \mathbf{c})$ has a large distribution distance to real data distribution. Therefore, fitting the spoiled ODE trajectory is only to make the generation quality bad. In contrast, when phasing the whole ODE trajectory into several sub-trajectories, the negative influence is greatly alleviated. On one hand, the starting point $\mathbf{x}_{s_{m+1}}$ of sub-trajectories is replaced by adding noise to the real data. On the other hand, the distribution distance between the distribution of solution points $\mathbf{x}_{s_m}$ of sub-trajectories and real data distribution $\mathbb{P}^{\text{data}}_{s_m}$ at the same timestep is much smaller proportional to the noise level introduced. To put it straightforwardly, even though the distribution gap between the real data and the samples generated from $\epsilon_\phi(\mathbf{x}_t, t, \mathbf{c})$ is large, adding noise to them reduce the gap. Thereby, we can optionally train a consistency model whether supporting larger values of CFG or not.

### 3.3 Adversarial Consistency Loss

We introduce an adversarial loss to enforce the distribution consistency, which greatly improves the generation quality in few-step settings. For convenience, we introduce an additional symbol $\mathcal{T}_{t \to s}$ which represents a flow from $\mathbb{P}_t$ to $\mathbb{P}_s$. Let $\mathcal{T}^\phi_{t \to s}$, $\mathcal{T}^\theta_{t \to s}$ be the transition mapping following the ODE trajectory of pre-trained diffusion and our consistency models. Additionally, let $\mathcal{T}^-_{s \to t}$ be the distribution transition following the forward process SDE (adding noise). The loss function is defined as the following

$$\mathcal{L}^{adv}_{\text{PCM}}(\boldsymbol{\theta}, \boldsymbol{\theta}^-; \boldsymbol{\phi}, m) = D\left(\mathcal{T}^-_{s_m \to s}\mathcal{T}^\theta_{t_{n+k} \to s_m}\#\mathbb{P}_{t_{n+k}} \left\| \mathcal{T}^-_{s_m \to s}\mathcal{T}^{\theta^-}_{t_n \to s_m}\mathcal{T}^\phi_{t_{n+1} \to t_n}\#\mathbb{P}_{t_{n+1}} \right. \right), \quad (8)$$

where $\#$ is the pushforward operator, and $D$ is the distribution distance metric. To penalize the distribution distance, we apply the GAN-style training paradigm. To be specific, as shown in Fig. 4, for the sampled $\mathbf{x}_{t_{n+k}}$ and the $\mathbf{x}^\phi_{t_n}$ solved through the pre-trained diffusion model $\phi$, we first compute their predicted solution point $\tilde{\mathbf{x}}_{s_m} = \boldsymbol{f}^m_{\boldsymbol{\theta}}(\mathbf{x}_{t_{n+k}}, t_{n+k})$ and $\hat{\mathbf{x}}_{s_m} = \boldsymbol{f}_{\boldsymbol{\theta}^-}(\mathbf{x}^\phi_{t_n}, t_n)$. Then we randomly add noise to $\tilde{\mathbf{x}}_{s_m}$ and $\hat{\mathbf{x}}_{s_m}$ to obtain $\tilde{\mathbf{x}}_s$ and $\hat{\mathbf{x}}_s$ with randomly sampled $s \in [s_m, s_{m+1}]$. We optimize the adversarial loss between $\tilde{\mathbf{x}}_s$ and $\hat{\mathbf{x}}_s$. Specifically, the $\mathcal{L}^{adv}_{\text{PCM}}$ can be re-written as

$$\mathcal{L}^{adv}_{\text{PCM}}(\boldsymbol{\theta}, \boldsymbol{\theta}^-; \boldsymbol{\phi}, m) = \text{ReLU}(1 + \mathcal{D}(\tilde{\mathbf{x}}_s, s, \mathbf{c})) + \text{ReLU}(1 - \mathcal{D}(\hat{\mathbf{x}}_s, s, \mathbf{c})), \quad (9)$$

where $\text{ReLU}(x) = x$ if $x > 0$ else $\text{ReLU}(x) = 0$, $\mathcal{D}$ is the discriminator, $\mathbf{c}$ is the image conditions (e.g., text prompts), and the loss is updated in a min-max manner [12]. Therefore the eventual optimization objective is $\mathcal{L}_{\text{PCM}} + \lambda \mathcal{L}^{adv}_{\text{PCM}}$ with $\lambda$ as a hyper-parameter controlling the trade-off of distribution consistency and instance consistency. We adopt $\lambda = 0.1$ for all training settings. However, we hope to clarify that the adversarial loss has an intrinsic difference from the GAN. The GAN training aims to align the training data distribution and the generation distribution of the model. That is, $D(\mathcal{T}^\theta_{t_{n+k} \to s_m}\#\mathbb{P}_{t_{n+k}} \| \mathbb{P}_{s_m})$ In Theorem 5, we show that consistency property is enforced, our introduced adversarial loss will also coverage to zero. Yet in Theorem 6, we show that combining standard GAN with consistency distillation is a flawed design when considering the pre-trained data distribution and distillation data distribution mismatch. Its loss will be non-zero when the self-consistency property is achieved, thus corrupting the consistency distillation learning. Our experimental results also verify our statement (Fig. 7).

Table 2: One-step generation comparison on Stable Diffusion v1-5.

| METHODS | COCO-30K | | | | CC12M-30K | | | | CONSISTENCY ↑ |
|---|---|---|---|---|---|---|---|---|---|
| | FID ↓ | FID-CLIP ↓ | FID-SD ↓ | CLIP SCORE ↑ | FID ↓ | FID-CLIP ↓ | FID-SD ↓ | CLIP SCORE ↑ | |
| InstaFlow [31] | **13.59** | **9.56** | 11.51 | 29.37 | 15.06 | 6.16 | 11.90 | 24.52 | 0.61 |
| SD-Turbo [49] | 16.56 | 12.08 | 10.62 | **31.21** | 17.17 | 6.18 | 12.48 | 26.30 | 0.71 |
| CTM [21] | 67.55 | 30.39 | 63.55 | 23.98 | 56.39 | 28.47 | 56.34 | 18.81 | 0.65 |
| LCM [35] | 53.81 | 21.04 | 53.43 | 25.23 | 44.35 | 18.58 | 42.67 | 20.38 | 0.62 |
| TCD [74] | 71.69 | 31.69 | 68.04 | 23.60 | 57.97 | 30.03 | 57.21 | 18.57 | - |
| Ours | 17.91 | 11.66 | **8.27** | 29.26 | **14.79** | **5.38** | **7.91** | **26.33** | **0.81** |

# 4 Experiments

## 4.1 Experimental Setup

**Dataset. Training dataset:** For image generation, we train all models on the CC3M [5] dataset. For video generation, we train the model on WebVid-2M [2]. **Evaluation dataset:** For image generation, we evaluate the performance on the COCO-2014 [28] following the 30K split of karpathy. We also evaluate the performance on the CC12M with our randomly chosen 30K split. For video generation, we evaluate with the captions of UCF-101 [58].

**Backbones.** We verify the text-to-image generation based on Stable Diffusion v1-5 [45] and Stable Diffusion XL [41]. We verify the text-to-video generation following the design of AnimateLCM [63] with decoupled consistency distillation.

**Evaluation metrics. Image:** We report the FID [14] and CLIP score [43] of the generated images and the validation 30K-sample splits. Following [8, 47], we also compute the FID with CLIP features (FID-CLIP). Note that, all baselines and our method focus on distilling the knowledge from the pre-trained diffusion models for acceleration. Therefore, we also compute the FID of all baselines and the generated images of original pre-trained diffusion models including Stable Diffusion v1-5 and Stable Diffusion XL (FID-SD). **Video:** For video generation, we evaluate the performance from three perspectives: the CLIP Score to measure the text-video alignment, the CLIP Consistency to measure the inter-frame consistency of the generated videos, the Flow Magnitude to measure the motion magnitude of the generated videos with Raft-Large [59].

## 4.2 Comparison

**Comparison methods.** We compare PCM with Stable Diffusion v1-5 to methods including Stable Diffusion v1-5 [45], InstaFlow [31], LCM [35], CTM [21], TCD [74] and SD-Turbo [49]. We compare PCM with Stable Diffusion XL to methods including Stable Diffusion XL [41], CTM [21], SDXL-Lightning [27], SDXL-Turbo [49], and LCM [35]. We apply the 'Ours' and 'Ours*' to denote our methods trained with CFG-augmented ODE solver or not. We only report the performance of 'Ours*' with more than 4 steps which aligns with our claim that it is only possible when phasing the ODE trajectory into multiple sub-trajectories. For video generation, we compare with DDIM [54], DPM [33], and AnimateLCM [63].

**Qualitative comparison.** We evaluate our model and comparison methods with a diverse set of prompts in different inference steps. The results are listed in Fig. 5. Our method shows clearly the top performance in both image visual quality and text-image alignment across 1–16 steps.

**Quantitative comparison. One-step generation:** We show the one-step generation results comparison of methods based on Stable Diffusion v1-5 and Stable Diffusion XL in Table 2 and Table 5, respectively. Notably, PCM consistently surpasses the consistency model-based methods including LCM and CTM by a large margin. Additionally, it achieves comparable or even superior to the state-of-the-art GAN-based (SD-Turbo, SDXL-Turbo, SDXL-Lightning) or Rectified-Flow-based (InstaFlow) one-step generation methods. Note that InstaFlow applies the LIPIPS [72] loss for training and SDXL-Turbo can only generate $512 \times 512$ resolution images, therefore it is easy for them to obtain higher scores. **Multi-step generation:** We report the FID changes of different methods on COCO-30K and CC12M-30K in Table 1 and Table 3. 'Ours' and 'Ours*' achieve the best or second-best performance in most cases. It is notably the gap of performance between our methods and other baselines becoming large as the timestep increases, which indicates the phased nature of our methods supports more powerful multi-step sampling ability. **Video generation:** We show the quantitative comparison of video generation in Table 4, our model achieves consistent superior

Table 3: Comparison of FID-SD on CC12M-30K with Stable Diffusion XL.

| METHODS | 1-Step | 2-Step | 4-Step | 8-Step | 16-Step |
|---|---|---|---|---|---|
| SDXL-Lightning [27] | 8.69 | 7.29 | 5.26 | 5.83 | 6.10 |
| SDXL-Turbo (512 × 512) [49] | 6.64 | 6.53 | 7.39 | 13.88 | 26.55 |
| SDXL-LCM [35] | 57.70 | 19.64 | 11.22 | 12.89 | 23.33 |
| SDXL-CTM [21] | 72.45 | 24.06 | 20.67 | 39.89 | 39.18 |
| Ours | 8.76 | 7.02 | 6.59 | 5.19 | 4.92 |
| Ours* | - | - | 6.26 | 5.27 | 5.09 |

Table 4: Quantitative comparison for video generation.

| Methods | CLIP Score ↑ | | | Flow Magnitude ↑ | | | CLIP Consistency ↑ | | |
|---|---|---|---|---|---|---|---|---|---|
| | 1-Step | 2-Step | 4-Step | 1-Step | 2-Step | 4-Step | 1-Step | 2-Step | 4-Step |
| DDIM [54] | 4.44 | 7.09 | 23.05 | - | - | 1.47 | - | - | 0.877 |
| DPM [33] | 11.21 | 17.93 | 28.57 | - | - | 1.95 | - | - | 0.947 |
| AnimateLCM [63] | 25.41 | 29.39 | 30.62 | 1.10 | 1.81 | 2.40 | 0.967 | 0.957 | 0.965 |
| Ours | 29.88 | 30.22 | 30.72 | 4.56 | 4.38 | 4.69 | 0.956 | 0.962 | 0.968 |

Table 5: One-step and two-step generation comparison on Stable Diffusion XL.

| METHODS | COCO-30K (one-step) | | | | CC12M-30K (two-step) | | | | CONSISTENCY ↑ |
|---|---|---|---|---|---|---|---|---|---|
| | FID ↓ | FID-CLIP ↓ | FID-SD ↓ | CLIP SCORE ↑ | FID ↓ | FID-CLIP ↓ | FID-SD ↓ | CLIP SCORE ↑ | |
| SDXL-Turbo (512 × 512) [49] | 19.84 | 13.56 | 9.40 | 32.31 | 15.36 | 5.26 | 6.53 | 27.91 | 0.74 |
| SDXL-Lightning [27] | 19.73 | 13.33 | 9.11 | 30.81 | 17.99 | 7.39 | 7.29 | 26.31 | 0.76 |
| SDXL-LCM [35] | 74.65 | 31.63 | 74.46 | 27.29 | 25.88 | 10.36 | 19.64 | 25.84 | 0.66 |
| SDXL-CTM [21] | 82.14 | 37.43 | 88.20 | 26.48 | 32.05 | 12.50 | 24.06 | 24.79 | 0.66 |
| Ours | 21.23 | 13.66 | 9.32 | 31.55 | 17.87 | 5.67 | 7.02 | 27.10 | 0.83 |

performance. The 1-step and 2-step generation results of DDIM and DPM are very noisy, therefore it is meaningless to evaluate their Flow Magnitude and CLIP Consistency.

**Human evaluation metrics.** To more comprehensively reflect the performance of phased consistency models, we conduct a thorough evaluation using human aesthetic preference metrics, encompassing 1–16 steps. This assessment employs well-regarded metrics, including HPSv2 (HPS) [66], PickScore (PICKSCORE) [23], and Laion Aesthetic Score (AES) [50], to benchmark our method against all comparative baselines. As shown in Table 6 and Table 7, across all evaluated settings, our method consistently achieves either superior or comparable results, with a marked performance advantage over the consistency model baseline LCM, demonstrating its robustness and appeal across diverse human-centric evaluation criteria. We conduct a human preference ablation study on the proposed adversarial consistency loss, with the results presented in Table 8. The inclusion of adversarial consistency loss consistently enhances human evaluation metrics across different inference steps.

Table 6: Aesthetic evaluation on SD v1-5.

| Steps | Methods | HPS | AES | PICKSCORE |
|---|---|---|---|---|
| 1 | InstaFlow | 0.267 | 5.010 | 0.207 |
| | SD-Turbo | 0.276 (1) | 5.445 (1) | 0.223 (1) |
| | CTM | 0.240 | 5.155 | 0.195 |
| | LCM | 0.251 | 5.178 | 0.201 |
| | Ours | 0.276 (1) | 5.389 (2) | 0.213 (2) |
| 2 | InstaFlow | 0.249 | 5.050 | 0.196 |
| | SD-Turbo | 0.278 (1) | 5.570 (1) | 0.226 (1) |
| | CTM | 0.267 | 5.117 | 0.208 |
| | LCM | 0.266 | 5.135 | 0.210 |
| | Ours | 0.275 (2) | 5.370 (2) | 0.217 (2) |
| 4 | InstaFlow | 0.243 | 4.765 | 0.192 |
| | SD-Turbo | 0.278 (2) | 5.537 (1) | 0.224 (1) |
| | CTM | 0.274 | 5.189 | 0.213 |
| | LCM | 0.273 | 5.264 | 0.215 |
| | Ours | 0.279 (1) | 5.412 (2) | 0.217 (2) |
| 8 | InstaFlow | 0.267 | 4.548 | 0.189 |
| | SD-Turbo | 0.276 (2) | 5.390 (2) | 0.221 (1) |
| | CTM | 0.271 | 5.026 | 0.210 |
| | LCM | 0.274 | 5.366 | 0.216 |
| | Ours | 0.278 (1) | 5.398 (1) | 0.218 (2) |
| 16 | InstaFlow | 0.237 | 4.437 | 0.187 |
| | SD-Turbo | 0.277 (1) | 5.275 | 0.219 (1) |
| | CTM | 0.270 | 4.870 | 0.209 |
| | LCM | 0.274 | 5.352 (2) | 0.216 |
| | Ours | 0.277 (1) | 5.442 (1) | 0.217 (2) |

Table 7: Aesthetic evaluation on SDXL.

| Steps | Methods | HPS | AES | PICKSCORE |
|---|---|---|---|---|
| 1 | SDXL-Lightning | 0.278 | 5.65 (1) | 0.223 |
| | SDXL-Turbo | 0.279 (1) | 5.40 | 0.228 (1) |
| | SDXL-CTM | 0.239 | 4.86 | 0.201 |
| | SDXL-LCM | 0.205 | 5.04 | 0.206 |
| | Ours | 0.280 (1) | 5.62 (2) | 0.225 (2) |
| 2 | SDXL-Lightning | 0.280 | 5.72 (1) | 0.227 (1) |
| | SDXL-Turbo | 0.281 (2) | 5.46 | 0.226 (2) |
| | SDXL-CTM | 0.267 | 5.58 | 0.216 |
| | SDXL-LCM | 0.265 | 5.40 | 0.217 |
| | Ours | 0.282 (1) | 5.688 (2) | 0.225 |
| 4 | SDXL-Lightning | 0.281 | 5.76 (2) | 0.228 (1) |
| | SDXL-Turbo | 0.284 (1) | 5.49 | 0.224 |
| | SDXL-CTM | 0.278 | 5.84 (1) | 0.221 |
| | SDXL-LCM | 0.274 | 5.48 | 0.223 |
| | Ours | 0.284 (1) | 5.645 | 0.228 (2) |
| 8 | SDXL-Lightning | 0.282 | 5.75 (2) | 0.229 (1) |
| | SDXL-Turbo | 0.283 (2) | 5.59 | 0.225 |
| | SDXL-CTM | 0.276 | 5.88 (1) | 0.218 |
| | SDXL-LCM | 0.277 | 5.57 | 0.223 |
| | Ours | 0.285 (1) | 5.676 | 0.229 (2) |
| 16 | SDXL-Lightning | 0.280 (2) | 5.72 (2) | 0.225 (2) |
| | SDXL-Turbo | 0.277 | 5.56 | 0.219 |
| | SDXL-CTM | 0.274 | 5.85 (1) | 0.215 |
| | SDXL-LCM | 0.276 | 5.64 | 0.221 |
| | Ours | 0.284 (1) | 5.646 | 0.228 (1) |

Table 8: Aesthetic ablation study on the adversarial consistency loss.

| Methods | Step 1 | | | Step 2 | | | Step 4 | | | Step 8 | | | Step 16 | | |
|---|---|---|---|---|---|---|---|---|---|---|---|---|---|---|---|
| | HPS | AES | PICKSCORE | HPS | AES | PICKSCORE | HPS | AES | PICKSCORE | HPS | AES | PICKSCORE | HPS | AES | PICKSCORE |
| PCM w/ adv | 0.280 | 5.620 | 0.225 | 0.282 | 5.688 | 0.225 | 0.284 | 5.645 | 0.228 | 0.285 | 5.676 | 0.229 | 0.284 | 5.646 | 0.228 |
| PCM w/o adv | 0.251 | 4.994 | 0.206 | 0.275 | 5.502 | 0.220 | 0.281 | 5.576 | 0.225 | 0.283 | 5.637 | 0.227 | 0.283 | 5.620 | 0.227 |

## 4.3 Ablation Study

**Sensitivity to negative prompt.** To show the comparison of sensitivity to negative prompt between our model tuned without CFG-augmented ODE solver and LCM. We provide an example of a prompt

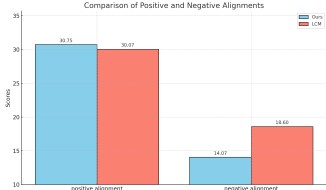

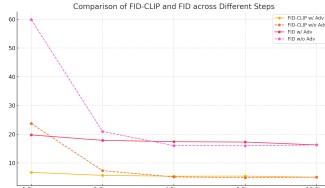

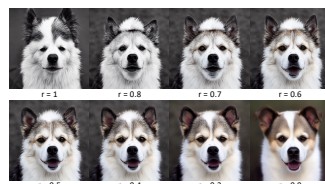

Figure 6: Sensitivity to negative prompt.

Figure 7: Effectiveness of adversarial consistency loss.

Figure 8: Randomness for sampling.

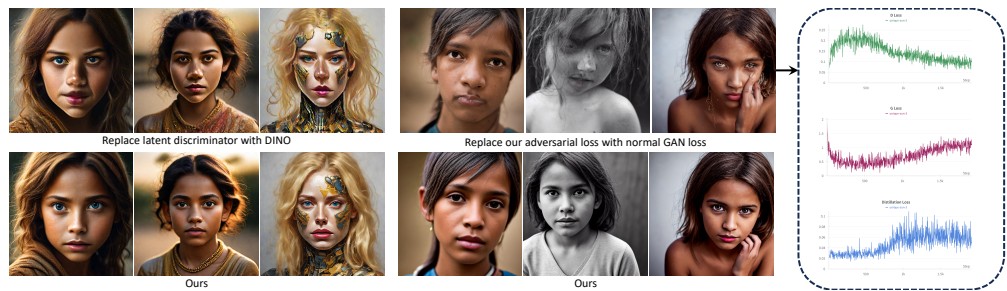

Figure 9: **Ablation study on the adversarial consistency design.** (Left) Replacing latent discriminator with DINO causes detail loss. (Right) Replacing our adversarial loss with normal GAN loss causes training conflicted objectives and instability.

and negative prompt to GPT-4o and ask it to generate 100 pairs of prompts and their corresponding negative prompts. For each prompt, we generate 10 images. We first generate images without using the negative prompt to show the positive prompt alignment comparison. Then we generate images with positive and negative prompts. We compute the CLIP score of generated images and the prompts and negative prompts. Fig. 6 shows that we not only achieve better prompt alignment but are much more sensitive to negative prompts.

**Consistent generation ability.** Consistent generation ability under different inference steps is valuable in practice for multistep refinement. We compute the average CLIP similarity between the 1-step generation and the 16-step generation for each method. As shown in the rightmost column of Table 2 and Table 5, our method achieves significantly better consistent generation ability.

**Adversarial consistency design and its effectiveness.** We show the ablation study on the adversarial consistency loss design and its effectiveness. **From the architecture level of discriminator**, we compare the latent discriminator shared from the teacher diffusion model and the pixel discriminator from pre-trained DINO [4]. Note that DINO is trained with 224 resolutions, therefore we should resize the generation results and feed them into DINO. We find this could make the generation results fail at details as shown in the left of Fig. 9. **From the adversarial loss**, we compare our adversarial loss to the normal GAN loss. We find normal GAN loss causes the training to be unstable and corrupts the generation results, which aligns with our previous analysis. **For its effectiveness**, we compare the FID-CLIP and FID scores with the adversarial loss or without the adversarial loss under different inference steps. Fig. 7 shows that it greatly improves the FID scores in the low-step regime and gradually coverage to similar performance of our model without using the adversarial loss as the step increases.

**Randomness for sampling.** Fig. 8 illustrates the influence of the randomness introduced in sampling as Eq. 44. The figure shows that introducing a certain of randomness in sampling may help to alleviate unrealistic objects or shapes.

## 5 Limitations and Conclusions

Despite being able to generate high-quality images and videos in a few steps, we find that when the number of steps is very low, especially with only one step, the generation quality is unstable. The model may produce structural errors or blurry images. Fortunately, we discover that this phenomenon can be mitigated through multi-step refinement. In conclusion, in this paper, we observe the defects in latent consistency models. We summarize these defects on three levels, analyze their causes, and generalize the design framework to address these defects.

# 6    Acknowledgements

This project is funded in part by National Key R&D Program of China Project 2022ZD0161100, by the Centre for Perceptual and Interactive Intelligence (CPII) Ltd under the Innovation and Technology Commission (ITC)'s InnoHK, by General Research Fund of Hong Kong RGC Project 14204021. Hongsheng Li is a PI of CPII under the InnoHK.

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

# Appendix

# I Related Works

## I.1 Diffusion Models

Diffusion models [15, 57, 20, 73] have gradually become the dominant foundation models in image synthesis. Many works have greatly explored the nature of diffusion models [29, 6, 57, 22] and generalize/improve the design space of diffusion models [54, 20, 22]. Some works explore the model architecture for diffusion models [7, 40, 9, 75]. Some works scale up the diffusion models for text-conditioned synthesis or real-world applications [45, 41, 52, 64, 18, 36, 69, 17, 61]. Some works explore the sampling acceleration methods, including scheduler-level [20, 33, 54] or training-level [38, 56]. The formal ones are basically to explore better approximation of the PF-ODE [33, 54].

The latter are mainly distillation methods [38, 46, 30, 65, 70, 10] or initializing diffusion weights for GAN training [49, 27, 39, 19].

## I.2 Consistency Models

The consistency model is a new family of generative models [11, 56, 55, 62, 26, 32] supporting fast and high quality generation. It can be trained either by distillation or direct training without teacher models. Improved techniques even allow consistency training excelling the performance of diffusion training [55]. Consistency trajectory model [21] proposes to learn the trajectory consistency, providing a more flexible framework. Some works combine [24] consistency training with GAN [24] for better training efficiency. Some works adopt continuous consistency models and achieve excelling performance [11, 62, 32]. Some works apply the idea of consistency model to language model [25] and policy learning [42, 34]. Some works extends the application scope of consistency models for text-conditioned image generation [35, 44, 74] and text-conditioned video generation [63, 37, 71, 60].

We notice that a recent work multistep consistency models [13] also proposes to splitting the ODE trajectory into multi-parts for consistency learning, and here we hope to credit their work for the valuable exploration. However, our work is principally different from multistep consistency models. Since they do not open-source their code and weights, we clarify the difference between our work and multistep consistency models based on the details of their technical report. Firstly, they didn't provide explicit model definitions and boundary conditions nor theoretically show the error bound to prove the soundness of their work. Yet in our work, we believe we have given an explicit definition of important components and theoretically shown the soundness of important techniques applied. For example, they claimed using DDIM for training and inference, while in our work, we have shown that although we can also optionally parameterize as the DDIM format, there's an intrinsic difference between that parameterization and DDIM (i.e., the difference between exact solution learning and first-order approximation). The aDDIM and invDDIM as highlighted in their pseudo code have no relation to PCM. Secondly, they are mainly for the unconditional or class-conditional generation, yet we aim for text-conditional generation in large models and dive into the influence of classifier-free guidance in consistency distillation. Besides, we introduce an adversarial loss that aligns well with consistency learning and improves the generation results in few-step settings. We recommend readers to read their work for better comparison.

# II Proofs

## II.1 Phased Consistency Distillation

The following is an extension of the original proof of consistency distillation for phased consistency distillation.

**Theorem 1.** *For arbitrary sub-trajectory* $[s_m, s_{m+1}]$. *let* $\Delta t_m := \max_{t_n, t_{n+1} \in [s_m, s_{m+1}]}\{|t_{n+1} - t_n|\}$, *and* $\boldsymbol{f}^m(\cdot, \cdot; \boldsymbol{\phi})$ *be the target phased consistency function induced by the pre-trained diffusion model (empirical PF-ODE). Assume the* $\boldsymbol{f}_{\boldsymbol{\theta}}^m$ *is L-Lipschitz and the ODE solver has local error uniformly bounded by* $\mathcal{O}((t_{n+1} - t_n)^{p+1})$ *with* $p \geq 1$. *Then, if* $\mathcal{L}_{PCM}(\boldsymbol{\theta}, \boldsymbol{\theta}; \boldsymbol{\phi}) = 0$, *we have*

$$\sup_{t_n, t_{n+1} \in [s_m, s_{m+1}], \mathbf{x}} \|\boldsymbol{f}_{\boldsymbol{\theta}}^m(\mathbf{x}, t_n) - \boldsymbol{f}^m(\mathbf{x}, t_n)\| = \mathcal{O}((\Delta t_m)^p).$$

*Proof.* From the condition $\mathcal{L}_{\text{PCM}}(\boldsymbol{\theta}, \boldsymbol{\theta}; \boldsymbol{\phi}) = 0$, for any $\mathbf{x}_{t_n}$ and $t_n, t_{n+1} \in [s_m, s_{m+1}]$, we have

$$\boldsymbol{f}_{\boldsymbol{\theta}}^m(\mathbf{x}_{t_{n+1}, t_{n+1}}) \equiv \boldsymbol{f}_{\boldsymbol{\theta}}^m(\hat{\boldsymbol{x}}_{t_n}^{\boldsymbol{\phi}}, t_n) \tag{10}$$

Denote $e_n^m := \boldsymbol{f}_{\boldsymbol{\theta}}^m(\mathbf{x}_{t_n}, t_n) - \boldsymbol{f}^m(\mathbf{x}_{t_n}, t_n; \boldsymbol{\phi})$, we have

$$
\begin{aligned}
e_{n+1}^m &= \boldsymbol{f}_{\boldsymbol{\theta}}^m(\mathbf{x}_{t_{n+1}}, t_{n+1}) - \boldsymbol{f}^m(\mathbf{x}_{t_{n+1}}, t_{n+1}; \boldsymbol{\phi}) \\
&= \boldsymbol{f}_{\boldsymbol{\theta}}^m(\hat{\mathbf{x}}_{t_n}^{\boldsymbol{\phi}}, t_n) - \boldsymbol{f}^m(\mathbf{x}_{t_n}, t_n; \boldsymbol{\phi}) \\
&= \boldsymbol{f}_{\boldsymbol{\theta}}^m(\hat{\mathbf{x}}_{t_n}^{\boldsymbol{\phi}}, t_n) - \boldsymbol{f}_{\boldsymbol{\theta}}^m(\mathbf{x}_{t_n}, t_n) + \boldsymbol{f}_{\boldsymbol{\theta}}^m(\mathbf{x}_{t_n}, t_n) - \boldsymbol{f}^m(\mathbf{x}_{t_n}, t_n; \boldsymbol{\phi}) \\
&= \boldsymbol{f}_{\boldsymbol{\theta}}^m(\hat{\mathbf{x}}_{t_n}^{\boldsymbol{\phi}}, t_n) - \boldsymbol{f}_{\boldsymbol{\theta}}^m(\mathbf{x}_{t_n}, t_n) + e_n^m.
\end{aligned}
\tag{11}
$$

Considering that $\boldsymbol{f}_{\boldsymbol{\theta}}^m$ is $L$-Lipschitz, we have

$$
\begin{aligned}
\|e_{n+1}^m\|_2 &= \|e_n^m + \boldsymbol{f}_{\boldsymbol{\theta}}^m(\hat{\mathbf{x}}_{t_n}^{\phi}, t_n) - \boldsymbol{f}_{\boldsymbol{\theta}}^m(\mathbf{x}_{t_n}, t_n)\|_2 \\
&\leq \|e_n^m\|_2 + \|\boldsymbol{f}_{\boldsymbol{\theta}}^m(\hat{\mathbf{x}}_{t_n}^{\phi}, t_n) - \boldsymbol{f}_{\boldsymbol{\theta}}^m(\mathbf{x}_{t_n}, t_n)\|_2 \\
&\leq \|e_n^m\|_2 + L\|\mathbf{x}_{t_n} - \mathbf{x}_{t_n}^{\phi}\|_2 \\
&= \|e_n^m\|_2 + L \cdot \mathcal{O}((t_{n+1} - t_n)^{p+1}) \\
&\leq \|e_n^m\|_2 + L(t_{n+1} - t_n) \cdot \mathcal{O}((\Delta t_m)^p) .
\end{aligned}
\tag{12}
$$

Besides, due to the boundary condition, we have

$$
\begin{aligned}
e_{s_m}^m &= \boldsymbol{f}_{\boldsymbol{\theta}}^m(\mathbf{x}_{s_m}, s_m) - \boldsymbol{f}^m(\mathbf{x}_{s_m}, s_m; \phi) \\
&= \mathbf{x}_{s_m} - \mathbf{x}_{s_m} \\
&= 0
\end{aligned}
\tag{13}
$$

Hence, we have

$$
\begin{aligned}
\|e_{n+1}^m\|_2 &\leq \|e_{s_m}^m\|_2 + L \cdot \mathcal{O}((\Delta t)^p) \sum_{t_i, t_{i+1} \in [s_m, s_{m+1}]} t_{i+1} - t_i \\
&= 0 + L \cdot \mathcal{O}((\Delta t)^p) \cdot (s_{m+1} - s_m) \\
&= \mathcal{O}((\Delta t_m)^p)
\end{aligned}
\tag{14}
$$

$\square$

**Theorem 2.** *For arbitrary set of sub-trajectories $\{[s_i, s_{i+1}]\}_{i=m}^{m'}$. Let $\Delta t_m := \max_{t_n, t_{n+1} \in [s_m, s_{m+1}]}\{|t_{n+1} - t_n|\}$, $\Delta t_{m,m'} := \max_{i \in [m', m]}\{\Delta t_i\}$, and $\boldsymbol{f}^{m,m'}(\cdot, \cdot; \phi)$ be the target phased consistency function induced by the pre-trained diffusion model (empirical PF-ODE). Assume the $\boldsymbol{f}_{\boldsymbol{\theta}}^{m,m'}$ is $L$-Lipschitz and the ODE solver has local error uniformly bounded by $\mathcal{O}((t_{n+1} - t_n)^{p+1})$ with $p \geq 1$. Then, if $\mathcal{L}_{PCM}(\boldsymbol{\theta}, \boldsymbol{\theta}; \phi) = 0$, we have*

$$
\sup_{t_n \in [s_m, s_{m+1}), \mathbf{x}} \|\boldsymbol{f}_{\boldsymbol{\theta}}^{m,m'}(\mathbf{x}, t_n) - \boldsymbol{f}^{m,m'}(\mathbf{x}, t_n)\| = \mathcal{O}((\Delta t_{m,m'})^p).
$$

*Proof.* From the condition $\mathcal{L}_{\mathrm{PCM}}(\boldsymbol{\theta}, \boldsymbol{\theta}; \phi) = 0$, for any $\mathbf{x}_{t_n}$ and $t_n, t_{n+1} \in [s_m, s_{m+1}]$, we have

$$
\boldsymbol{f}_{\boldsymbol{\theta}}^m(\mathbf{x}_{t_{n+1}, t_{n+1}}) \equiv \boldsymbol{f}_{\boldsymbol{\theta}}^m(\hat{\boldsymbol{x}}_{t_n}^{\phi}, t_n)
\tag{15}
$$

Denote $e_n^{m,m'} := \boldsymbol{f}_{\boldsymbol{\theta}}^{m,m'}(\mathbf{x}_{t_n}, t_n) - \boldsymbol{f}^{m,m'}(\mathbf{x}_{t_n}, t_n; \phi)$, we have

$$
\begin{aligned}
e_{n+1}^{m,m'} &= \boldsymbol{f}_{\boldsymbol{\theta}}^{m,m'}(\mathbf{x}_{t_{n+1}}, t_{n+1}) - \boldsymbol{f}^{m,m'}(\mathbf{x}_{t_{n+1}}, t_{n+1}; \phi) \\
&= \boldsymbol{f}_{\boldsymbol{\theta}}^{m,m'}(\hat{\mathbf{x}}_{t_n}^{\phi}, t_n) - \boldsymbol{f}^{m,m'}(\mathbf{x}_{t_n}, t_n; \phi) \\
&= \boldsymbol{f}_{\boldsymbol{\theta}}^{m,m'}(\hat{\mathbf{x}}_{t_n}^{\phi}, t_n) - \boldsymbol{f}_{\boldsymbol{\theta}}^{m,m'}(\mathbf{x}_{t_n}, t_n) + \boldsymbol{f}_{\boldsymbol{\theta}}^{m,m'}(\mathbf{x}_{t_n}, t_n) - \boldsymbol{f}^{m,m'}(\mathbf{x}_{t_n}, t_n; \phi) \\
&= \boldsymbol{f}_{\boldsymbol{\theta}}^{m,m'}(\hat{\mathbf{x}}_{t_n}^{\phi}, t_n) - \boldsymbol{f}_{\boldsymbol{\theta}}^{m,m'}(\mathbf{x}_{t_n}, t_n) + e_n^{m,m'} .
\end{aligned}
\tag{16}
$$

Considering that $\boldsymbol{f}_{\boldsymbol{\theta}}^m$ is $L$-Lipschitz, we have

$$
\begin{aligned}
\|e_{n+1}^{m,m'}\|_2 &= \|e_n^{m,m'} + \boldsymbol{f}_{\boldsymbol{\theta}}^{m,m'}(\hat{\mathbf{x}}_{t_n}^{\phi}, t_n) - \boldsymbol{f}_{\boldsymbol{\theta}}^{m,m'}(\mathbf{x}_{t_n}, t_n)\|_2 \\
&\leq \|e_n^{m,m'}\|_2 + \|\boldsymbol{f}_{\boldsymbol{\theta}}^{m,m'}(\hat{\mathbf{x}}_{t_n}^{\phi}, t_n) - \boldsymbol{f}_{\boldsymbol{\theta}}^{m,m'}(\mathbf{x}_{t_n}, t_n)\|_2 \\
&\leq \|e_n^{m,m'}\|_2 + L\|\boldsymbol{f}_{\boldsymbol{\theta}}^{m,m'+1}(\hat{\mathbf{x}}_{t_n}^{\phi}, t_n) - \boldsymbol{f}_{\boldsymbol{\theta}}^{m,m'+1}(\mathbf{x}_{t_n}, t_n)\|_2 \\
&\leq \|e_n^{m,m'}\|_2 + L^2\|\boldsymbol{f}_{\boldsymbol{\theta}}^{m,m'+2}(\hat{\mathbf{x}}_{t_n}^{\phi}, t_n) - \boldsymbol{f}_{\boldsymbol{\theta}}^{m,m'+2}(\mathbf{x}_{t_n}, t_n)\|_2 \\
&\leq \quad \vdots \\
&\leq \|e_n^{m,m'}\|_2 + L^{m-m'}\|\boldsymbol{f}_{\boldsymbol{\theta}}^m(\hat{\mathbf{x}}_{t_n}^{\phi}, t_n) - \boldsymbol{f}_{\boldsymbol{\theta}}^m(\mathbf{x}_{t_n}, t_n)\|_2 \\
&\leq \|e_n^{m,m'}\|_2 + L^{m-m'+1}\|\mathbf{x}_{t_n} - \mathbf{x}_{t_n}^{\phi}\|_2 \\
&= \|e_n^{m,m'}\|_2 + L^{m-m'+1} \cdot \mathcal{O}((t_{n+1} - t_n)^{p+1}) \\
&\leq \|e_n^{m,m'}\|_2 + L^{m-m'+1}(t_{n+1} - t_n) \cdot \mathcal{O}((\Delta t_m)^p) .
\end{aligned}
\tag{17}
$$

Hence, we have

$$\|e^{m,m'}_{n+1}\|_2 \leq \|e^{m,m'}_{s_m}\|_2 + L^{m-m'+1} \cdot \mathcal{O}((\Delta t)^p) \sum_{t_i,t_{i+1}\in[s_m,s_{m+1}]} t_{i+1} - t_i$$

$$= \|e^{m,m'}_{s_m}\|_2 + L^{m-m'+1} \cdot \mathcal{O}((\Delta t)^p) \cdot (s_{m+1} - s_m) \quad , \quad (18)$$

$$= \|e^{m,m'}_{s_m}\|_2 + \mathcal{O}((\Delta t_m)^p)$$

$$= \|e^{m-1,m'}_{s_m}\|_2 + \mathcal{O}((\Delta t_m)^p)$$

Where the last equation is due to the boundary condition $\boldsymbol{f}^{m,m'}_{\boldsymbol{\theta}}(\mathbf{x}_{s_m}, s_m) = \boldsymbol{f}^{m-1,m'}_{\boldsymbol{\theta}}(\boldsymbol{f}^{m}_{\boldsymbol{\theta}}(\mathbf{x}_{s_m}, s_m), s_m) = \boldsymbol{f}^{m-1,m'}_{\boldsymbol{\theta}}(\mathbf{x}_{s_m}, s_m)$.

Thereby, we have

$$\|e^{m,m'}_{n+1}\|_2 \leq \|e^{m-1,m'}_{s_m}\|_2 + \mathcal{O}((\Delta t_m)^p)$$

$$\leq \|e^{m-2,m'}_{s_{m-1}}\|_2 + \mathcal{O}((\Delta t_m)^p) + \mathcal{O}((\Delta t_{m-1})^p)$$

$$\leq \quad \vdots$$

$$\leq \|e^{m'}_{s_{m'+1}}\| + \sum_{i=m}^{m'+1} \mathcal{O}((\Delta t_i)^p) \quad (19)$$

$$\leq \sum_{i=m}^{m'} \mathcal{O}((\Delta t_i)^p)$$

$$\leq (m - m' + 1)\mathcal{O}((\Delta t_{m,m'})^p)$$

$$= \mathcal{O}((\Delta t_{m,m'})^p)$$

$$\square$$

## II.2 Parameterization Equivalence

We show that the parameterization of Equation 3 is equal to the DDIM inference format [33, 54].

**Theorem 3.** *Define* $F_{\boldsymbol{\theta}}(\mathbf{x}_t, t, s) = \frac{\alpha_s}{\alpha_t}\mathbf{x}_t - \alpha_s\hat{\epsilon}_{\boldsymbol{\theta}}(\mathbf{x}_t, t) \int_{\lambda_t}^{\lambda_s} e^{-\lambda}d\lambda$, *then the parameterization has the same format of DDIM* $\mathbf{x}_s = \alpha_s \left(\frac{\mathbf{x}_t - \sigma_t\hat{\boldsymbol{\epsilon}}_{\boldsymbol{\theta}}(\mathbf{x}_t,t)}{\alpha_t}\right) + \sigma_s\hat{\boldsymbol{\epsilon}}_{\boldsymbol{\theta}}(\mathbf{x}_t,t)$.

*Proof.*

$$F_{\boldsymbol{\theta}}(\mathbf{x}_t, t, s) = \frac{\alpha_s}{\alpha_t}\mathbf{x}_t - \alpha_s\hat{\epsilon}_{\boldsymbol{\theta}}(\mathbf{x}_t, t) \int_{\lambda_t}^{\lambda_s} e^{-\lambda}d\lambda$$

$$= \frac{\alpha_s}{\alpha_t}\mathbf{x}_t - \alpha_s\hat{\epsilon}_{\boldsymbol{\theta}}(\mathbf{x}_t, t)\left(e^{-\lambda_t} - e^{-\lambda_s}\right)$$

$$= \frac{\alpha_s}{\alpha_t}\mathbf{x}_t - \alpha_s\hat{\epsilon}_{\boldsymbol{\theta}}(\mathbf{x}_t, t)(\frac{\sigma_t}{\alpha_t} - \frac{\sigma_s}{\alpha_s}) \quad (20)$$

$$= \frac{\alpha_s}{\alpha_t}\mathbf{x}_t - \frac{\alpha_s\sigma_t}{\alpha_t}\hat{\boldsymbol{\epsilon}}_{\boldsymbol{\theta}}(\mathbf{x}_t, t) + \sigma_s\hat{\boldsymbol{\epsilon}}_{\boldsymbol{\theta}}(\mathbf{x}_t, t)$$

$$= \alpha_s \left(\frac{\mathbf{x}_t - \sigma_t\hat{\boldsymbol{\epsilon}}_{\boldsymbol{\theta}}(\mathbf{x}_t, t)}{\alpha_t}\right) + \sigma_s\hat{\boldsymbol{\epsilon}}_{\boldsymbol{\theta}}(\mathbf{x}_t, t)$$

$$\square$$

## II.3 Guided Distillation

We show the relationship of epsilon prediction of consistency models trained with guided distillation and diffusion models when considering the classifier-free guidance.

**Theorem 4.** *Assume the consistency model $\epsilon_{\theta}$ is trained by consistency distillation with the teacher diffusion model $\epsilon_{\phi}$, and the ODE solver is augmented with CFG value $w$ and null text embedding $\varnothing$. Let $c$ and $c_{neg}$ be the prompt and negative prompt applied for the inference of consistency model. Then, if the ODE solver is perfect and the $\mathcal{L}_{PCM}(\theta, \theta) = 0$, we have*

$$\epsilon_{\theta}(\mathbf{x}, t, \boldsymbol{c}, \boldsymbol{c_{neg}}; w') \propto ww' \left[ \epsilon_{\phi}(\mathbf{x}, t, \boldsymbol{c}) - ((1 - \frac{w-1}{ww'})\epsilon_{\phi}(\mathbf{x}, t, \boldsymbol{c_{neg}}) + \frac{w-1}{ww'}\epsilon_{\phi}(\mathbf{x}, t, \varnothing)) \right]$$
$$+ \epsilon_{\phi}(\mathbf{x}, t, \boldsymbol{c_{neg}})$$

*Proof.* If the ODE solver is perfect, that means the empirical PF-ODE is exactly the PF-ODE of the training data. Then considering Theorem 1 and Theorem 2, it is apparent that the consistency model will fit the PF-ODE. To show that, considering the case in Theorem 1, we have

$$
\begin{aligned}
e_{n+1}^m &= \boldsymbol{f}_{\boldsymbol{\theta}}^m(\mathbf{x}_{t_{n+1}}, t_{n+1}) - \boldsymbol{f}^m(\mathbf{x}_{t_{n+1}}, t_{n+1}; \boldsymbol{\phi}) \\
&= \boldsymbol{f}_{\boldsymbol{\theta}}^m(\hat{\mathbf{x}}_{t_n}^{\phi}, t_n) - \boldsymbol{f}^m(\mathbf{x}_{t_n}, t_n; \boldsymbol{\phi}) \\
&= \boldsymbol{f}_{\boldsymbol{\theta}}^m(\hat{\mathbf{x}}_{t_n}^{\phi}, t_n) - \boldsymbol{f}_{\boldsymbol{\theta}}^m(\mathbf{x}_{t_n}, t_n) + \boldsymbol{f}_{\boldsymbol{\theta}}^m(\mathbf{x}_{t_n}, t_n) - \boldsymbol{f}^m(\mathbf{x}_{t_n}, t_n; \boldsymbol{\phi}) \\
&= \boldsymbol{f}_{\boldsymbol{\theta}}^m(\hat{\mathbf{x}}_{t_n}^{\phi}, t_n) - \boldsymbol{f}_{\boldsymbol{\theta}}^m(\mathbf{x}_{t_n}, t_n) + e_n^m \\
&\overset{(i)}{=} \boldsymbol{f}_{\boldsymbol{\theta}}^m(\mathbf{x}_{t_n}, t_n) - \boldsymbol{f}_{\boldsymbol{\theta}}^m(\mathbf{x}_{t_n}, t_n) + e_n^m \\
&= e_n^m \\
&= \vdots \\
&= e_{s_m}^m \\
&= 0\,,
\end{aligned}
\tag{21}
$$

where $(i)$ is because the ODE solver is perfect. Considering our parameterization in Equation 3, then it should be equal to the exact solution in Equation 2. That is,

$$
\frac{\alpha_s}{\alpha_t}\mathbf{x}_t - \alpha_s \hat{\epsilon}_{\theta}(\mathbf{x}_t, t) \int_{\lambda_t}^{\lambda_s} e^{-\lambda} \mathrm{d}\lambda = \frac{\alpha_s}{\alpha_t}\mathbf{x}_t + \alpha_s \int_{\lambda_t}^{\lambda_s} e^{-\lambda} \sigma_{t_\lambda(\lambda)} \nabla \log \mathbb{P}_{t_\lambda(\lambda)}(\mathbf{x}_{t_\lambda(\lambda)}) \mathrm{d}\lambda
$$
$$
\hat{\epsilon}_{\theta}(\mathbf{x}_t, t) \int_{\lambda_t}^{\lambda_s} e^{-\lambda} \mathrm{d}\lambda = \int_{\lambda_t}^{\lambda_s} e^{-\lambda} \epsilon_{\phi}(\mathbf{x}_{t_\lambda(\lambda)}, t_\lambda(\lambda)) \mathrm{d}\lambda
\tag{22}
$$
$$
\hat{\epsilon}_{\theta}(\mathbf{x}_t, t) = \frac{\int_{\lambda_t}^{\lambda_s} e^{-\lambda} \epsilon_{\phi}(\mathbf{x}_{t_\lambda(\lambda)}, t_\lambda(\lambda)) \mathrm{d}\lambda}{\int_{\lambda_t}^{\lambda_s} e^{-\lambda} \mathrm{d}\lambda}\,.
$$

Then we have the epsilon prediction is weighted integral of diffusion-based epsilon prediction on the trajectory. Therefore, it is apparent that, for any $t' \le t$ and $t'$ on the same sub-trajectory with $t$, we have

$$\hat{\epsilon}_{\theta}(\mathbf{x}_t, t) \propto \epsilon_{\phi}(\mathbf{x}_{t'}, t') \tag{23}$$

When considering the text-conditioned generation and the ODE solver being augmented with the CFG value $w$, we have

$$\hat{\epsilon}_{\theta}(\mathbf{x}_t, t, \boldsymbol{c}) \propto \epsilon_{\phi}(\mathbf{x}_{t'}, t', \varnothing) + w(\epsilon_{\phi}(\mathbf{x}_{t'}, t', \boldsymbol{c}) - \epsilon_{\phi}(\mathbf{x}_{t'}, t', \varnothing))\,. \tag{24}$$

If we additionally apply the classifier-free guidance to the consistency models with negative prompt embedding $c_{\mathrm{neg}}$ and CFG value $w'$, we have

$$
\begin{aligned}
&\hat{\epsilon}_{\theta}(\mathbf{x}_t, t, \boldsymbol{c}, \boldsymbol{c}_{\mathrm{neg}}; w') \\
&= \hat{\epsilon}_{\theta}(\mathbf{x}_t, t, \boldsymbol{c}_{\mathrm{neg}}) + w'(\hat{\epsilon}_{\theta}(\mathbf{x}_t, t, \boldsymbol{c}) - \hat{\epsilon}_{\theta}(\mathbf{x}_t, t, \boldsymbol{c}_{\mathrm{neg}})) \\
&\propto \epsilon_{\phi}(\mathbf{x}_{t'}, t', \varnothing) + w(\epsilon_{\phi}(\mathbf{x}_{t'}, t', \boldsymbol{c}) - \epsilon_{\phi}(\mathbf{x}_{t'}, t', \varnothing)) \\
&\quad + w'\{[\epsilon_{\phi}(\mathbf{x}_{t'}, t', \varnothing) + w(\epsilon_{\phi}(\mathbf{x}_{t'}, t', \boldsymbol{c}) - \epsilon_{\phi}(\mathbf{x}_{t'}, t', \varnothing))] \\
&\quad - [\epsilon_{\phi}(\mathbf{x}_{t'}, t', \varnothing) + w(\epsilon_{\phi}(\mathbf{x}_{t'}, t', \boldsymbol{c}) - \epsilon_{\phi}(\mathbf{x}_{t'}, t', \varnothing))]\} \\
&= \epsilon_{\phi}(\mathbf{x}_{t'}, t', \varnothing) + w(\epsilon_{\phi}(\mathbf{x}_{t'}, t', \boldsymbol{c}_{\mathrm{neg}}) - \epsilon_{\phi}(\mathbf{x}_{t'}, t', \varnothing)) + ww'(\epsilon_{\phi}(\mathbf{x}_{t'}, t', \boldsymbol{c}) - \epsilon_{\phi}(\mathbf{x}_{t'}, t', \boldsymbol{c}_{\mathrm{neg}})) \\
&= ww'(\epsilon_{\phi}(\mathbf{x}_{t'}, t', \boldsymbol{c}) - ((1-\alpha)\epsilon_{\phi}(\mathbf{x}_{t'}, t', \boldsymbol{c}_{\mathrm{neg}}) + \alpha\epsilon_{\phi}(\mathbf{x}_{t'}, t', \varnothing))) + \epsilon_{\phi}(\mathbf{x}_{t'}, t', \boldsymbol{c}_{\mathrm{neg}})
\end{aligned}
\tag{25}
$$

$\square$

## II.4 Distribution Consistency Convergence

We firstly show that when $\mathcal{L}_{\text{PCM}} = 0$ is achieved, our distribution consistency loss will also converge to zero. Then, we additionally show that, when considering the pre-train data distribution and distillation data distribution mismatch, combining GAN is a flawed design. The loss is still non-zero even when the self-consistency is achieved, thus corrupting the training.

**Theorem 5.** *Denote the data distribution applied for consistency distillation phased is $\mathbb{P}_0$. And considering that the forward conditional probability path is defined by $\alpha_t \mathbf{x}_0 + \sigma_t \epsilon$, we further define the intermediate distribution $\mathbb{P}_t(\mathbf{x}) = (\mathbb{P}_0(\frac{\mathbf{x}}{\alpha_t}) \cdot \frac{1}{\alpha_t}) * \mathcal{N}(0, \sigma_t)$. Similarly, we denote the data distribution applied for pretraining the diffusion model is $\mathbb{P}_0^{pretrain}(\mathbf{x})$ and the intermediate distribution following forward process are $\mathbb{P}_t^{pretrain}(\mathbf{x}) = (\mathbb{P}_t^{pretrain}(\mathbf{x}) = (\mathbb{P}_0^{pretrain}(\frac{\mathbf{x}}{\alpha_t}) \cdot \frac{1}{\alpha_t}) * \mathcal{N}(0, \sigma_t))$. This is reasonable since current large diffusion models are typically trained with much more resources on much larger datasets compared to those of consistency distillation. And, we denote the flow $\mathcal{T}_{t \to s}^{\phi}$, $\mathcal{T}_{t \to s}^{\theta}$, and $\mathcal{T}_{t \to s}^{\phi'}$ correspond to our consistency model, pre-trained diffusion model, and the PF-ODE of the data distribution used for consistency distillation, respectively. Additionally, let $\mathcal{T}_{s \to t}^{-}$ be the distribution transition following the forward process SDE (adding noise). Then, if the $\mathcal{L}_{\text{PCM}} = 0$, for arbitrary sub-trajectory $[s_m, s_{m+1}]$, we have,*

$$\mathcal{L}_{PCM}^{adv}(\boldsymbol{\theta}, \boldsymbol{\theta}; \boldsymbol{\phi}, m) = D\left(\mathcal{T}_{s_m \to s}^{-} \mathcal{T}_{t_{n+1} \to s_m}^{\boldsymbol{\theta}} \# \mathbb{P}_{t_{n+1}} \middle\| \mathcal{T}_{s_m \to s}^{-} \mathcal{T}_{t_n \to s_m}^{\boldsymbol{\theta}} \mathcal{T}_{t_{n+1} \to t_n}^{\boldsymbol{\phi}} \# \mathbb{P}_{t_{n+1}}\right) = 0.$$

*Proof.* Firstly, considering that the forward process $\mathcal{T}_{s \to t}^{-}$ is equivalent to scaling the original variables and then performing convolution operations with $\mathcal{N}(\mathbf{0}, \sigma_{s \to t}^2 \mathbf{I})$. Therefore, as long as

$$D\left(\mathcal{T}_{t_{n+1} \to s_m}^{\boldsymbol{\theta}} \# \mathbb{P}_{t_{n+1}} \middle\| \mathcal{T}_{t_n \to s_m}^{\boldsymbol{\theta}} \mathcal{T}_{t_{n+1} \to t_n}^{\boldsymbol{\phi}} \# \mathbb{P}_{t_{n+1}}\right) = 0, \tag{26}$$

then we have

$$D\left(\mathcal{T}_{s_m \to s}^{-} \mathcal{T}_{t_{n+1} \to s_m}^{\boldsymbol{\theta}} \# \mathbb{P}_{t_{n+1}} \middle\| \mathcal{T}_{s_m \to s}^{-} \mathcal{T}_{t_n \to s_m}^{\boldsymbol{\theta}} \mathcal{T}_{t_{n+1} \to t_n}^{\boldsymbol{\phi}} \# \mathbb{P}_{t_{n+1}}\right) = 0. \tag{27}$$

From the condition $\mathcal{L}_{\text{PCM}}(\boldsymbol{\theta}, \boldsymbol{\theta}; \boldsymbol{\phi}) = 0$, for any $\mathbf{x}_{t_n} \in \mathbb{P}_{t_n}$ and $\mathbf{x}_{t_{n+1}} \in \mathbb{P}_{t_{n+1}}$ and $t_n, t_{n+1} \in [s_m, s_{m+1}]$, we have

$$\boldsymbol{f}_{\boldsymbol{\theta}}^m(\mathbf{x}_{t_{n+1}}, t_{n+1}) \equiv \boldsymbol{f}_{\boldsymbol{\theta}}^m(\hat{\boldsymbol{x}}_{t_n}^{\boldsymbol{\phi}}, t_n), \tag{28}$$

which induces that

$$\mathcal{T}_{t_{n+1} \to s_m}^{\boldsymbol{\theta}} \# \mathbb{P}_{t_{n+1}} \equiv \mathcal{T}_{t_n \to s_m}^{\boldsymbol{\theta}} \mathcal{T}_{t_{n+1} \to t_n}^{\boldsymbol{\phi}} \# \mathbb{P}_{t_{n+1}}. \tag{29}$$

Therefore, we show that if $\mathcal{L}_{PCM}(\boldsymbol{\theta}, \boldsymbol{\theta}; \boldsymbol{\phi}) = 0$, then

$$\mathcal{L}_{\text{PCM}}^{adv}(\boldsymbol{\theta}, \boldsymbol{\theta}; \boldsymbol{\phi}) = 0 \tag{30}$$

$\square$

**Theorem 6.** *Denote the data distribution applied for consistency distillation phased is $\mathbb{P}_0$. And considering that the forward conditional probability path is defined by $\alpha_t \mathbf{x}_0 + \sigma_t \epsilon$, we further define the intermediate distribution $\mathbb{P}_t(\mathbf{x}) = (\mathbb{P}_0(\frac{\mathbf{x}}{\alpha_t}) \cdot \frac{1}{\alpha_t}) * \mathcal{N}(0, \sigma_t)$. Similarly, we denote the data distribution applied for pretraining the diffusion model is $\mathbb{P}_0^{pretrain}(\mathbf{x})$ and the intermediate distribution following forward process are $\mathbb{P}_t^{pretrain}(\mathbf{x}) = (\mathbb{P}_t^{pretrain}(\mathbf{x}) = (\mathbb{P}_0^{pretrain}(\frac{\mathbf{x}}{\alpha_t}) \cdot \frac{1}{\alpha_t}) * \mathcal{N}(0, \sigma_t))$. This is reasonable since current large diffusion models are typically trained with much more resources on much larger datasets compared to those of consistency distillation. And, we denote the flow $\mathcal{T}_{t \to s}^{\phi}$, $\mathcal{T}_{t \to s}^{\theta}$, and $\mathcal{T}_{t \to s}^{\phi'}$ correspond to our consistency model, pre-trained diffusion model, and the PF-ODE of the data distribution used for consistency distillation, respectively. Then, if the $\mathcal{L}_{PCM} = 0$, for arbitrary sub-trajectory $[s_m, s_{m+1}]$, we have,*

$$\mathcal{L}_{PCM}^{adv}(\boldsymbol{\theta}, \boldsymbol{\theta}; \boldsymbol{\phi}, m) = D\left(\mathcal{T}_{t_{n+1} \to s_m}^{\boldsymbol{\theta}} \# \mathbb{P}_{t_{n+1}} \middle\| \mathbb{P}_{s_m}\right) \geq 0.$$

*Proof.* From the condition $\mathcal{L}_{\text{PCM}}(\boldsymbol{\theta}, \boldsymbol{\theta}; \boldsymbol{\phi}) = 0$, for any $\mathbf{x}_{t_n} \in \mathbb{P}_{t_n}$ and $\mathbf{x}_{t_{n+1}} \in \mathbb{P}_{t_{n+1}}$ and $t_n, t_{n+1} \in [s_m, s_{m+1}]$, we have

$$\boldsymbol{f}_{\boldsymbol{\theta}}^m(\mathbf{x}_{t_{n+1}}, t_{n+1}) \equiv \boldsymbol{f}_{\boldsymbol{\theta}}^m(\hat{\boldsymbol{x}}_{t_n}^{\boldsymbol{\phi}}, t_n), \tag{31}$$

which induces that

$$\mathcal{T}_{t_{n+1} \rightarrow s_m}^{\boldsymbol{\theta}} \# \mathbb{P}_{t_{n+1}} \equiv \mathcal{T}_{t_{n+1} \rightarrow s_m}^{\boldsymbol{\phi}} \# \mathbb{P}_{t_{n+1}} . \tag{32}$$

Besides, since $\mathcal{T}_{t \rightarrow s}^{\boldsymbol{\phi}'}$ corresponds to the PF-ODE of the data distribution used for consistency distillation, we can rewrite

$$\mathbb{P}_{s_m} \equiv \mathcal{T}_{t_{n+1} \rightarrow s_m}^{\boldsymbol{\phi}'} \# \mathbb{P}_{t_{n+1}} . \tag{33}$$

Therefore, we have

$$D\left(\mathcal{T}_{t_{n+1} \rightarrow s_m}^{\boldsymbol{\theta}} \# \mathbb{P}_{t_{n+1}} \middle\| \mathbb{P}_{s_m}\right) = D\left(\mathcal{T}_{t_{n+1} \rightarrow s_m}^{\boldsymbol{\phi}} \# \mathbb{P}_{t_{n+1}} \middle\| T_{t_{n+1} \rightarrow s_m}^{\boldsymbol{\phi}'} \# \mathbb{P}_{t_{n+1}}\right) . \tag{34}$$

Since we know that $\mathbb{P}_0^{pretrain} \neq \mathbb{P}_0$. Therefore, there exists $m$ and $n$ achieving the strict inequality. Specifically, if we set $s_m = \epsilon$ and $t_{n+1} = T$, we have

$$D\left(\mathcal{T}_{T \rightarrow \epsilon}^{\boldsymbol{\phi}} \# \mathbb{P}_T \middle\| T_{T \rightarrow \epsilon}^{\boldsymbol{\phi}'} \# \mathbb{P}_T\right) = D\left(\mathbb{P}_0^{pretrain} \middle\| \mathbb{P}_0\right) > 0 . \tag{35}$$

$\square$

## III    Discussions

### III.1    Numerical Issue of Parameterization

Even though our above-discussed parameterization is theoretically sound, it poses a numerical issue when applied to the epsilon-prediction-based models, especially for the one-step generation. To be specific, for the one-step generation, we are required to predict the solution point $\mathbf{x}_0$ with $\mathbf{x}_t$ from the self-consistency property of consistency models. With epsilon-prediction models, the formula can represented as the following

$$\mathbf{x}_0 = \frac{\mathbf{x}_t - \sigma_t \boldsymbol{\epsilon}_{\boldsymbol{\theta}}(\mathbf{x}_t, t)}{\alpha_t} . \tag{36}$$

However, when inference, we should choose $t = T$ since it is the noisy timestep that is closest to the normal distribution. For the DDPM framework, there should be $\sigma_T \approx 1$ and $\alpha_T \approx 0$ to make the noisy timestep $T$ as close to the normal distribution as possible. For instance, the noise schedulers applied by Stable Diffusion v1-5 and Stable Diffusion XL all have the $\alpha_T = 0.068265$. Therefore, we have

$$\mathbf{x}_0 = \frac{\mathbf{x}_T - \sigma_T \boldsymbol{\epsilon}_{\boldsymbol{\theta}}(\mathbf{x}_t, t)}{\alpha_T} \approx 14.64(\mathbf{x}_T - \sigma_T \boldsymbol{\epsilon}_{\boldsymbol{\theta}}(\mathbf{x}_t, t)) . \tag{37}$$

This indicates that we will multiply over $10\times$ to the model prediction. In our experiments, we find the SDXL is more influenced by this issue, tending to produce artifact points or lines.

Generally speaking, using the x0-prediction and v-prediction can well solve these issues. It is obvious for the x0-prediction. For the v-prediction, we have

$$\mathbf{x}_0 = \alpha_t \mathbf{x}_t - \sigma_t \boldsymbol{v}_{\boldsymbol{\theta}}(\mathbf{x}_t, t) . \tag{38}$$

However, since the diffusion models are trained with the epsilon-prediction, transforming them into the other prediction formats takes additional computation resources and might harm the performance due to the relatively large gaps among different prediction formats.

We solve this issue through a simple way that we set a clip boundary to the value of $\alpha_t$. Specifically, we apply

$$\mathbf{x}_0 = \frac{\mathbf{x}_t - \sigma_t \boldsymbol{\epsilon}_{\boldsymbol{\theta}}(\mathbf{x}_t, t)}{\max\{\alpha_t, 0.5\}} . \tag{39}$$

### III.2 Why CTM Needs Target Timestep Embeddings?

Consistency trajectory model (CTM), as we discussed, proposes to learn how 'move' from arbitrary points on the ODE trajectory to arbitrary other points. Thereby, the model should be aware of 'where they are' and 'where they target' for precise prediction.

Basically, they can still follow our parameterization with an additional target timestep embedding.

$$\mathbf{x}_s = \frac{\alpha_s}{\alpha_t}\mathbf{x}_t - \alpha_s\hat{\boldsymbol{\epsilon}}_{\boldsymbol{\theta}}(\mathbf{x}_t, t, s) \int_{\lambda_t}^{\lambda_s} e^{-\lambda}\mathrm{d}\lambda \tag{40}$$

In this design, we have

$$\hat{\boldsymbol{\epsilon}}_{\boldsymbol{\theta}}(\mathbf{x}_t, t, s) = \frac{\int_{\lambda_t}^{\lambda_s} e^{-\lambda}\boldsymbol{\epsilon}_{\boldsymbol{\phi}}(\mathbf{x}_{t_\lambda(\lambda)}, t_\lambda(\lambda))\mathrm{d}\lambda}{\int_{\lambda_t}^{\lambda_s} e^{-\lambda}\mathrm{d}\lambda} . \tag{41}$$

Here, we show that dropping the target timestep embedding $s$ is a **flawed** design.

Assume we hope to predict the results at timestep $s'$ and timestep $s''$ from timestep $t$ and sample $\mathbf{x}_t$, where $s' < s''$.

If we hope to learn both $\mathbf{x}_{s'}$ and $\mathbf{x}_{s''}$ correctly, without the target timestep embedding, we must have

$$\hat{\boldsymbol{\epsilon}}_{\boldsymbol{\theta}}(\mathbf{x}_t, t) = \frac{\int_{\lambda_t}^{\lambda_{s'}} e^{-\lambda}\boldsymbol{\epsilon}_{\boldsymbol{\phi}}(\mathbf{x}_{t_\lambda(\lambda)}, t_\lambda(\lambda))\mathrm{d}\lambda}{\int_{\lambda_t}^{\lambda_{s'}} e^{-\lambda}\mathrm{d}\lambda} \tag{42}$$

$$\hat{\boldsymbol{\epsilon}}_{\boldsymbol{\theta}}(\mathbf{x}_t, t) = \frac{\int_{\lambda_t}^{\lambda_{s''}} e^{-\lambda}\boldsymbol{\epsilon}_{\boldsymbol{\phi}}(\mathbf{x}_{t_\lambda(\lambda)}, t_\lambda(\lambda))\mathrm{d}\lambda}{\int_{\lambda_t}^{\lambda_{s''}} e^{-\lambda}\mathrm{d}\lambda} . \tag{43}$$

This will lead to the conclusion that all the segments $[t, s]$ on the ODE trajectory have the same weighted epsilon prediction, which violates the truth, ignoring the dynamic variations and dependencies specific to each segments.

### III.3 Contributions

Here we re-emphasize the key components of PCM and summarize the contributions of our work.

The motivation of our work is to accelerate the sampling of high-resolution text-to-image and text-to-video generation with consistency models training paradigm. Previous work, latent consistency model (LCM), tried to replicate the power of the consistency model in this challenging setting but did not achieve satisfactory results. We observe and analyze the limitations of LCM from three perspectives and propose PCM, generalizing the design space and tackling all these limitations.

At the heart of PCM is to phase the whole ODE trajectory into multiple phases. Each phase corresponding to a sub-trajectory is treated as an independent consistency model learning objective. We provide a standard definition of PCM and show that the optimal error bounds between the prediction of trained PCM and PF-ODE are upper-bounded by $\mathcal{O}((\Delta t)^p)$. The phasing technique allows for deterministic multi-step sampling, ensuring consistent generation results under different inference steps.

Additionally, we provide a deep analysis of the parameterization when converting pre-trained diffusion models into consistency models. From the exact solution format of PF-ODE, we propose a simple yet effective parameterization and show that it has the same formula as first-order ODE solver DDIM. However, we show that the parameterization has a natural difference from the DDIM ODE solver. That is the difference between exact solution learning and score estimation. Besides, we point out that even though the parameterization is theoretically sound, it poses numerical issues when building the one-step generator. We propose a simple yet effective threshold clip strategy for the parameterization.

We introduce an innovative adversarial loss, which greatly boosts the generation quality at a low-step regime. We implement the loss in a GAN style. However, we show that our method has an intrinsic

difference from traditional GAN. We show that our introduced adversarial loss will converge to zero when the self-consistency property is achieved. However, the GAN loss will still be non-zero when considering the distribution distance between the pre-trained dataset for diffusion training and the distillation dataset for consistency distillation. We investigate the structures of discriminators and compare the latent-based discriminator and pixel-based discriminator. Our finding is that applying the latent-based visual backbone of pre-trained diffusion U-Net makes the discriminator design simple and can produce better visual results compared to pixel-based discriminators. We also implement a similar discriminator structure for the text-to-video generation with a temporal inflated U-Net.

For the setting of text-to-image generation and text-to-video generation, classifier-free guidance (CFG) has become an important technique for achieving better controllability and generation quality. We investigate the influence of CFG on consistency distillation. And, to our knowledge, we, for the first time, point out the relations of consistency model prediction and diffusion model prediction when considering the CFG-augmented ODE solver (guided distillation). We show that this technique causes the trained consistency models unable to use large CFG values and be less sensitive to the negative prompt.

We achieve state-of-the-art few-step text-to-image generation and text-to-video generation with only 8 A 800 GPUs, indicating the advancements of our method.

## IV  Sampling

### IV.1  Introducing Randomness for Sampling

For a given initial sample at timestep $t$ belonging to the sub-trajectory $[s_m, s_{m+1}]$, we can support deterministic sampling according to the definition of $\boldsymbol{f}^{m,0}$.

However, previous work [21, 67] reveals that introducing a certain degree of stochasticity can lead to better generation results. We also observe a similar trade-off phenomenon in our practice. Therefore, we reparameterize the $F_{\boldsymbol{\theta}}(\mathbf{x}, t, s)$ as

$$\alpha_s\big(\frac{\mathbf{x}_t - \sigma_t\boldsymbol{\epsilon}_{\boldsymbol{\theta}}(\mathbf{x}_t, t)}{\alpha_t}\big) + \sigma_s\big(\sqrt{r}\boldsymbol{\epsilon}_{\boldsymbol{\theta}}(\mathbf{x}_t, t) + \sqrt{(1-r)}\boldsymbol{\epsilon}\big), \quad \boldsymbol{\epsilon} \sim \mathcal{N}(\mathbf{0}, \mathbf{I}). \tag{44}$$

By controlling the value of $r \in [0, 1]$, we can determine the stochasticity for generation. With $r = 1$, it is pure deterministic sampling. With $r = 0$, it degrades to pure stochastic sampling. Note that, introducing the random $\boldsymbol{\epsilon}$ will make the generation results be away from the target induced by the diffusion models. However, since $\boldsymbol{\epsilon}_{\boldsymbol{\theta}}$ and $\boldsymbol{\epsilon}$ all follows the normal distribution, we have $\sqrt{r}\boldsymbol{\epsilon}_{\boldsymbol{\theta}}(\mathbf{x}_t, t) + \sqrt{(1-r)}\boldsymbol{\epsilon} \sim \mathcal{N}(\mathbf{0}, \mathbf{I})$ and thereby the predicted $\mathbf{x}_s$ should still follow the same distribution $\mathbb{P}_s$ approximately.

### IV.2  Improving Diversity for Generation

Our another observation is the generation diversity with guided distillation is limited compared to original diffusion models. This is due to that the consistency models are distilled with a relatively large CFG value for guided distillation. The large CFG value, though known for enhancing the generation quality and text-image alignment, will degrade the generation diversity. We explore a simple yet effective strategy by adjusting the CFG values. $w'\boldsymbol{\epsilon}_{\boldsymbol{\theta}}(\mathbf{x}, t, \boldsymbol{c}) + (1 - w')\boldsymbol{\epsilon}_{\boldsymbol{\theta}}(\mathbf{x}, t, \varnothing)$, where $w' \in (0.5, 1]$. The epsilon prediction is the convex combination of the conditional prediction and the unconditional prediction.

## V  Societal Impact

### V.1  Positive Societal Impact

We firmly believe that our work has a profound positive impact on society. While diffusion techniques excel in producing high-quality images and videos, their iterative inference process incurs significant computational and power costs. Our approach accelerates the inference of general diffusion models by up to 20 times or more, thus substantially reducing computational and power consumption. Moreover, it fosters enthusiasm among creators engaged in AI-generated content creation and lowers entry barriers.

## V.2 Negative Societal Impact

Addressing potential negative consequences, given the generative nature of the model, there's a risk of generating false or harmful content.

## V.3 Safeguards

To mitigate this, we conduct harmful information detection on user-provided text inputs. If harmful prompts are detected, the generation process is halted. Additionally, our method, fortunately, builds upon the open-source Stable Diffusion model, which includes an internal safety checker for detecting harmful content. This feature significantly enhances our ability to prevent the generation of harmful content.

# VI Implementation Details

## VI.1 Training Details

For the comparison of baselines, the training code for InstaFlow [31], SDXL-Turbo [49], SD-Turbo [49], TCD [74], and SDXL-Lightning [27] are not open-sourced yet, so we only compare their open-source weights.

For LCM [35], CTM [21], and our method, they are all implemented by us and trained with the same configuration.

Specifically, for multi-step models, we trained LoRA with a rank of 64. For models based on SD v1-5, we used a learning rate of 5e-6, a batch size of 160, and trained for 5k iterations. For models based on SDXL, we used a learning rate of 5e-6, a batch size of 80, and trained for 10k iterations. We did not use EMA for LoRA training.

For single-step models, we followed the approach of SD-Turbo and SDXL-Lightning, training all parameters. For models based on SD v1-5, we used a learning rate of 5e-6, a batch size of 160, and trained for 10k iterations. For models based on SDXL, we used a learning rate of 1e-6, a batch size of 16, and trained for 50k iterations. We used EMA=0.99 for training.

For CTM, we additionally learned a timestep embedding to indicate the target timestep.

For all training settings, we uniformly sample 50 timesteps from the 1000 timesteps of StableDiffusion and apply DDIM as the ODE solver.

## VI.2 Pseudo Training Code

---

**Algorithm 1** Phased Consistency Distillation with CFG-augmented ODE solver (PCD)

---

**Input:** dataset $\mathcal{D}$, initial model parameter $\boldsymbol{\theta}$, learning rate $\eta$, ODE solver $\Psi(\cdot,\cdot,\cdot,\cdot)$, distance metric $d(\cdot,\cdot)$, EMA rate $\mu$, noise schedule $\alpha_t, \sigma_t$, guidance scale $[w_{\min}, w_{\max}]$, number of ODE step $k$, discretized timesteps $t_0 = \epsilon < t_1 < t_2 < \cdots < t_N = T$, edge timesteps $s_0 = t_0 < s_1 < s_2 < \cdots < s_M = t_N \in \{t_i\}_{i=0}^N$ to split the ODE trajectory into $M$ sub-trajectories.
Training data : $\mathcal{D}_{\mathbf{x}} = \{(\mathbf{x}, \boldsymbol{c})\}$
$\boldsymbol{\theta}^- \leftarrow \boldsymbol{\theta}$
**repeat**
 Sample $(\boldsymbol{z}, \boldsymbol{c}) \sim \mathcal{D}_z, n \sim \mathcal{U}[0, N-k]$ and $\omega \sim [\omega_{\min}, \omega_{\max}]$
 Sample $\mathbf{x}_{t_{n+k}} \sim \mathcal{N}(\alpha_{t_{n+k}} \boldsymbol{z}; \sigma_{t_{n+k}}^2 \mathbf{I})$
 Determine $[s_m, s_{m+1}]$ given $n$
 $\mathbf{x}_{t_n}^{\phi} \leftarrow (1+\omega)\Psi(\mathbf{x}_{t_{n+k}}, t_{n+k}, t_n, \boldsymbol{c}) - \omega\Psi(\mathbf{x}_{t_{n+k}}, t_{n+k}, t_n, \varnothing)$
 $\tilde{\mathbf{x}}_{s_m} = \boldsymbol{f}_{\boldsymbol{\theta}}^m(\mathbf{x}_{t_{n+k}}, t_{n+k}, \boldsymbol{c})$ and $\hat{\mathbf{x}}_{s_m} = \boldsymbol{f}_{\boldsymbol{\theta}^-}(\mathbf{x}_{t_n}^{\phi}, t_n, \boldsymbol{c})$
 Obtain $\tilde{\mathbf{x}}_s$ and $\hat{\mathbf{x}}_s$ through adding noise to $\tilde{\mathbf{x}}_{s_m}$ and $\hat{\mathbf{x}}_{s_m}$
 $\mathcal{L}(\boldsymbol{\theta}, \boldsymbol{\theta}^-) = d(\tilde{\mathbf{x}}_{s_m}, \hat{\mathbf{x}}_{s_m}) + \lambda(\text{ReLU}(1 + \tilde{\mathbf{x}}_s) + \text{ReLU}(1 - \hat{\mathbf{x}}_s))$
 $\boldsymbol{\theta} \leftarrow \boldsymbol{\theta} - \eta\nabla_{\boldsymbol{\theta}}\mathcal{L}(\boldsymbol{\theta}, \boldsymbol{\theta}^-)$
 $\boldsymbol{\theta}^- \leftarrow \text{stopgrad}(\mu\boldsymbol{\theta}^- + (1-\mu)\boldsymbol{\theta})$
**until** convergence

---

---

**Algorithm 2** Phased Consistency Distillation with normal ODE solver (PCD*)

---

**Input:** dataset $\mathcal{D}$, initial model parameter $\boldsymbol{\theta}$, learning rate $\eta$, ODE solver $\Psi(\cdot,\cdot,\cdot,\cdot)$, distance metric $d(\cdot,\cdot)$, EMA rate $\mu$, noise schedule $\alpha_t, \sigma_t$, drop ratio $\eta$, number of ODE step $k$, discretized timesteps $t_0 = \epsilon < t_1 < t_2 < \cdots < t_N = T$, edge timesteps $s_0 = t_0 < s_1 < s_2 < \cdots < s_M = t_N \in \{t_i\}_{i=0}^{N}$ to split the ODE trajectory into $M$ sub-trajectories.
Training data : $\mathcal{D}_{\mathbf{x}} = \{(\mathbf{x}, \boldsymbol{c})\}$
$\boldsymbol{\theta}^- \leftarrow \boldsymbol{\theta}$
**repeat**
    Sample $(\boldsymbol{z}, \boldsymbol{c}) \sim \mathcal{D}_z$, $n \sim \mathcal{U}[0, N-k]$
    Sample $\mathbf{x}_{t_{n+k}} \sim \mathcal{N}(\alpha_{t_{n+k}} \boldsymbol{z}; \sigma_{t_{n+k}}^2 \mathbf{I})$
    Determine $[s_m, s_{m+1}]$ given $n$
    $r \sim \mathcal{U}[0, 1]$
    **if** $r < \eta$ **then**
    $\boldsymbol{c} = \varnothing$
    **end if**
    $\mathbf{x}_{t_n}^{\phi} \leftarrow \Psi(\mathbf{x}_{t_{n+k}}, t_{n+k}, t_n, \boldsymbol{c})$
    $\tilde{\mathbf{x}}_{s_m} = \boldsymbol{f}_{\boldsymbol{\theta}}^m(\mathbf{x}_{t_{n+k}}, t_{n+k}, \boldsymbol{c})$ and $\hat{\mathbf{x}}_{s_m} = \boldsymbol{f}_{\boldsymbol{\theta}^-}(\mathbf{x}_{t_n}^{\phi}, t_n, \boldsymbol{c})$
    Obtain $\tilde{\mathbf{x}}_s$ and $\hat{\mathbf{x}}_s$ through adding noise to $\tilde{\mathbf{x}}_{s_m}$ and $\hat{\mathbf{x}}_{s_m}$
    $\mathcal{L}(\boldsymbol{\theta}, \boldsymbol{\theta}^-) = d(\tilde{\mathbf{x}}_{s_m}, \hat{\mathbf{x}}_{s_m}) + \lambda(\text{ReLU}(1 + \tilde{\mathbf{x}}_s) + \text{ReLU}(1 - \hat{\mathbf{x}}_s))$
    $\boldsymbol{\theta} \leftarrow \boldsymbol{\theta} - \eta \nabla_{\boldsymbol{\theta}} \mathcal{L}(\boldsymbol{\theta}, \boldsymbol{\theta}^-)$
    $\boldsymbol{\theta}^- \leftarrow \text{stopgrad}(\mu \boldsymbol{\theta}^- + (1 - \mu)\boldsymbol{\theta})$
**until** convergence

---

## VI.3  Decoupled Consistency Distillation For Video Generation

Our video generation model follows the design space of most current text-to-video generation models, viewing videos as temporal stacks of images and inserting temporal blocks to a pre-trained text-to-image generation U-Net to accommodate 3D features of noisy video inputs. We term this process temporal inflation.

Video generation models are typically much more resource-consuming than image generation models. Also, the overall caption and visual quality of video datasets are generally inferior to those of image datasets. Therefore, we apply the decoupled consistency learning to ease the training burden of text-to-video generation, which was first proposed by the previous work AnimateLCM. To be specific, we first conduct phased consistency distillation on stable diffusion v1-5 to obtain the one-step text-to-image generation models. Then we apply the temporal inflation to adapt the text-to-image generation model for video generation. Eventually, we conduct phased consistency distillation on the video dataset. We observe an obvious training speed-up with this decoupled strategy. This allows us to train the state-of-the-art fast video generation models with only 8 A 800 GPUs.

## VI.4  Discriminator Design of Image Generation and Video Generation

### VI.4.1  Image Generation

The overall discriminator design was greatly inspired by previous work StyleGAN-T [48], which showed that a pre-trained visual backbone can work as a great discriminator. For training the discriminator, we freeze the original weight of pre-trained visual backbones and insert light-weight Convolution-based discriminator heads for training.

**Discriminator backbones.** We consider two types of visual backbones as the discriminator backbone: pixel-based and latent-based.

We apply DINO [4] as the pixel-based backbone. To be specific, once obtaining the denoised latent code, we decode it through the VAE decoder to obtain the generated image. Then, we resize the generated image into the resolution that DINO trained with (e.g., $224 \times 224$) and feed it as the inputs of the DINO backbone. We extract the hidden features of the DINO backbone of different layers and feed them to the corresponding discriminator heads. Since DINO is trained in a self-supervised manner and no texts are used for training. Therefore, it is necessary to incorporate the text embeddings in the discriminator heads for text-to-image generation. To achieve that, we use the embedding of [CLS] token in the last layer of the pre-trained CLIP text encoder, linearly map it to the same dimension of discriminator output, and conduct affine transformation to the discriminator output. However, the pixel-based backbones have inevitable drawbacks. Firstly, it requires mapping

the latent code to a very high-dimensional image through the VAE decoder, which is much more costly compared to traditional diffusion training. Secondly, it can only accept the inputs of clean images, which poses challenges to PCM training, since we require the discrimination of inputs of intermediate noisy inputs. Thirdly, it requires resizing the images into the relatively low resolution it trained with. When applying this backbone design with high-dimensional image generation with Stable-Diffusion XL, we observe unsatisfactory results.

We apply the U-Net of the pre-trained diffusion model as the latent-based backbone. It has several advantages compared to the pixel-based backbones. Firstly, trained on the latent space, the U-Net possesses rich knowledge about latent space and can directly work as the discriminator at the latent space, avoiding costly decoding processes. Besides, it can accept the inputs of latent code at different timesteps, which aligns well with the design of PCM and can be applied for regularizing the consistency of all the intermediate distributions instead (i.e., $\mathbb{P}_t(\mathbf{x})$) of distribution at the edge points (i.e., $\mathbb{P}_{s_m}(\mathbf{x})$). Additionally, since the text information is already encoded in the U-Net backbone, we do not need to incorporate the text information in the discriminator head, which simplifies the discriminator design. We insert several randomly initialized lightweight simple discriminator heads after each block of the pre-trained U-Net. Each discriminator head consists of two lightweight convolution blocks connected with residuals. Each convolution block is composed of a convolution 2D layer, Group Normalization, and GeLU non-linear activation function. Then we apply a 2D point-wise convolution layer and set the output channel to 1, thus mapping the input latent feature to a 2D scalar value output map with the same spatial size.

We train our one-step text-to-image model on Stable Diffusion XL using these two choices of discriminator respectively, while keeping the other settings unchanged. Our experiments reveal that the latent-based discriminator not only reduces the training cost but also provides more visually compelling generation results. Additionally, note that the teacher diffusion model applied for phased consistency distillation is actually a pre-trained U-Net. And we freeze the parameters of U-Net for the training discriminator. Therefore, we can apply the same U-Net, which simultaneously works as the teacher diffusion model for numerical ODE solver computation and discriminator visual backbone.

### VI.4.2 Video Generation

For the video discriminator, we mainly follow our design on the image discriminator. We use the same U-Net with temporal inflation as the visual backbone for video latent code as well as the teacher video diffusion model for phased consistency distillation. We still apply the 2D convolution layers as discriminator heads for each frame of hidden features extracted from the temporal inflated U-Net since we do not observe performance gain when using 3D convolutions. Note that the temporal relationships are already processed by the pre-trained temporal inflated U-Net, therefore using simple 2D convolution layers as discriminator heads is enough for supervision of the distribution of video inputs.

## VII   More Generation Results.

Prompt: "a **happy** white man in **black** suit, sky, **red** tie, **river**, **mountain**, colourful, **clouds**, best view, **fall**"

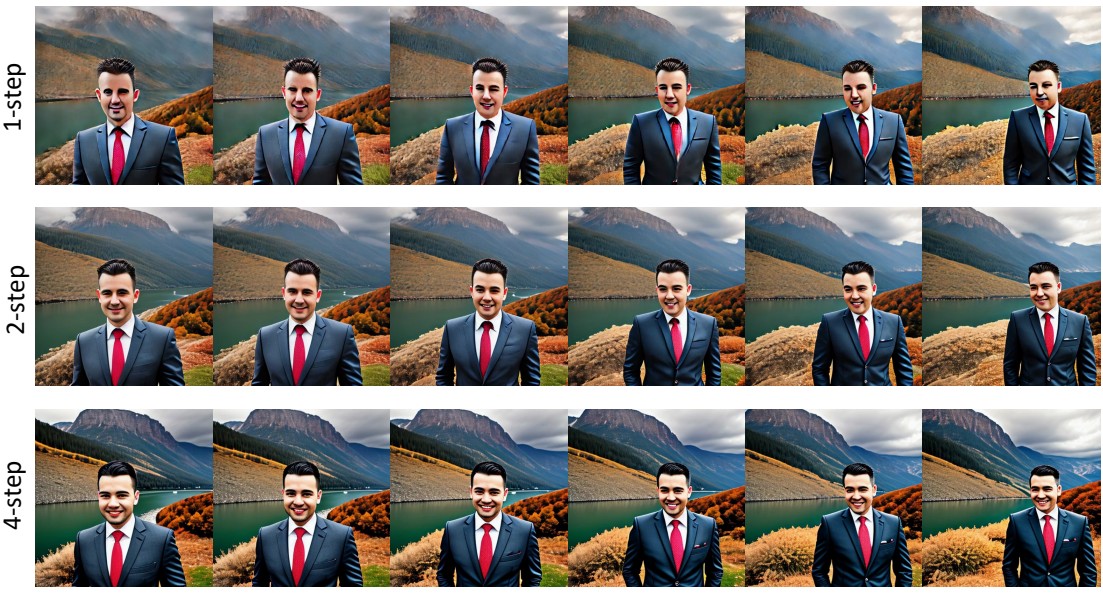

Prompt: "a astronaut walking on the moon"

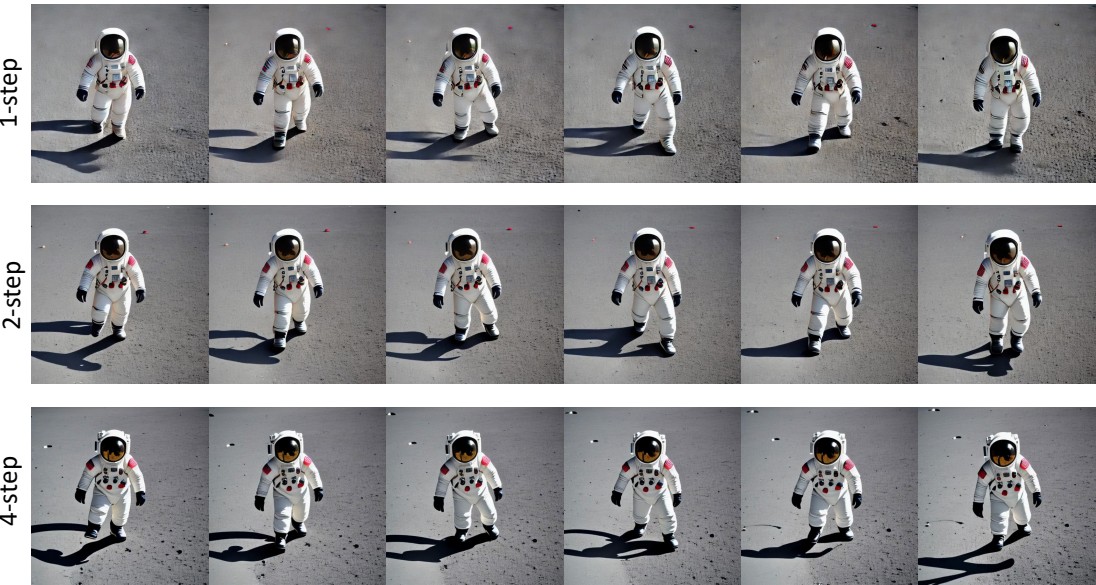

Figure 10: PCM video generation results with Stable-Diffusion v1-5 under $1 \sim 4$ inference steps.

Prompt: "Photo of a dramatic cliffside lighthouse in a storm, waves crashing, symbol of guidance and resilience"

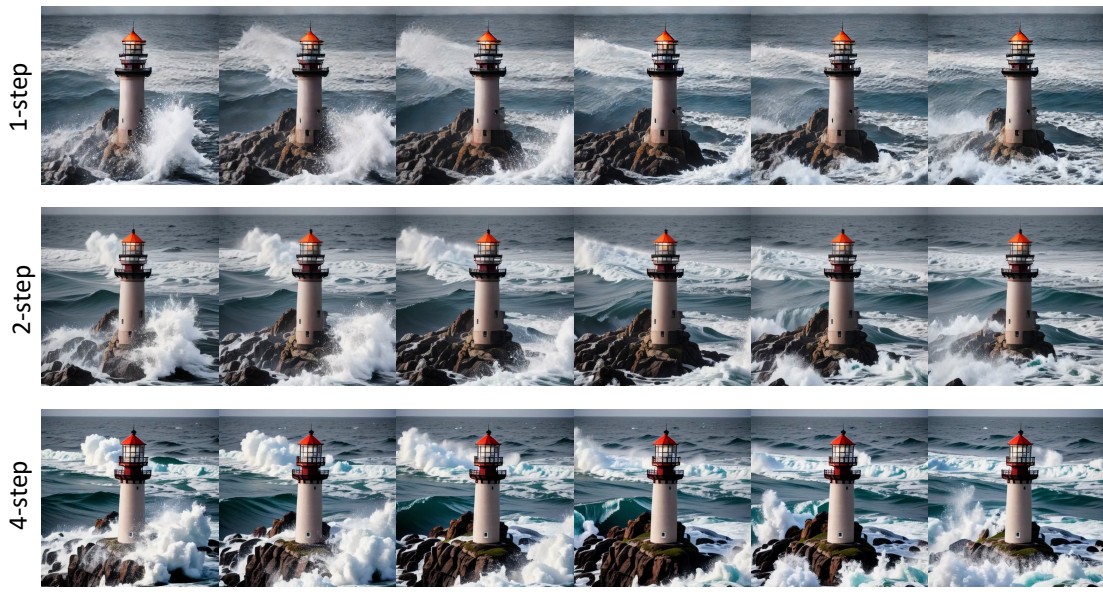

Prompt: "RAW photo, face portrait photo of beautiful 26 y.o woman, cute face, wearing black dress, happy face, hard shadows, cinematic shot, dramatic lighting"

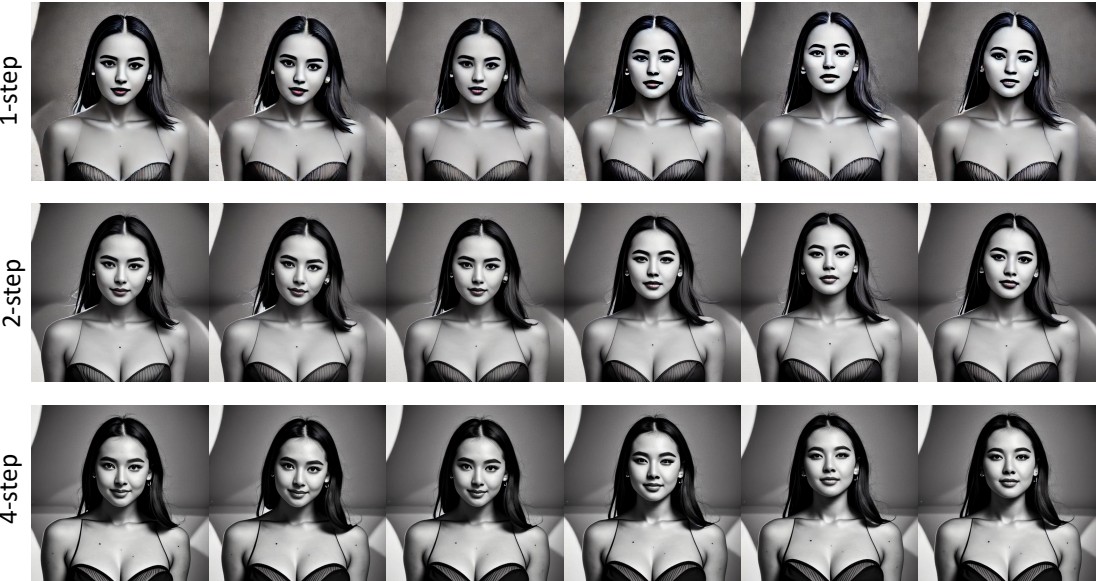

Figure 11: PCM video generation results with Stable-Diffusion v1-5 under $1 \sim 4$ inference steps.

Prompt: "a car running on the snowy road"

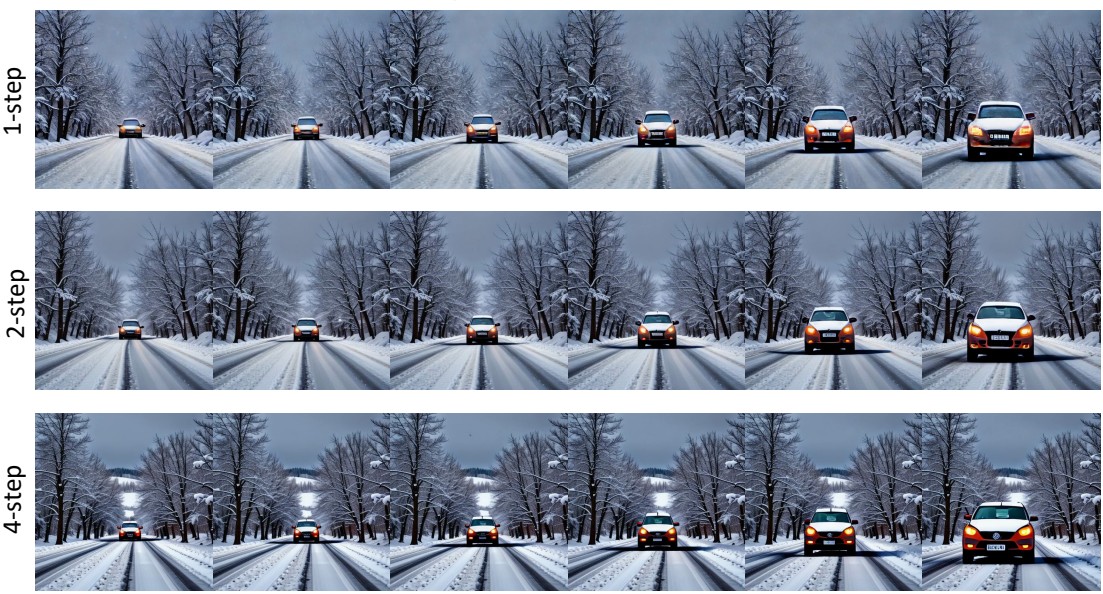

Prompt: "a monkey eating apple"

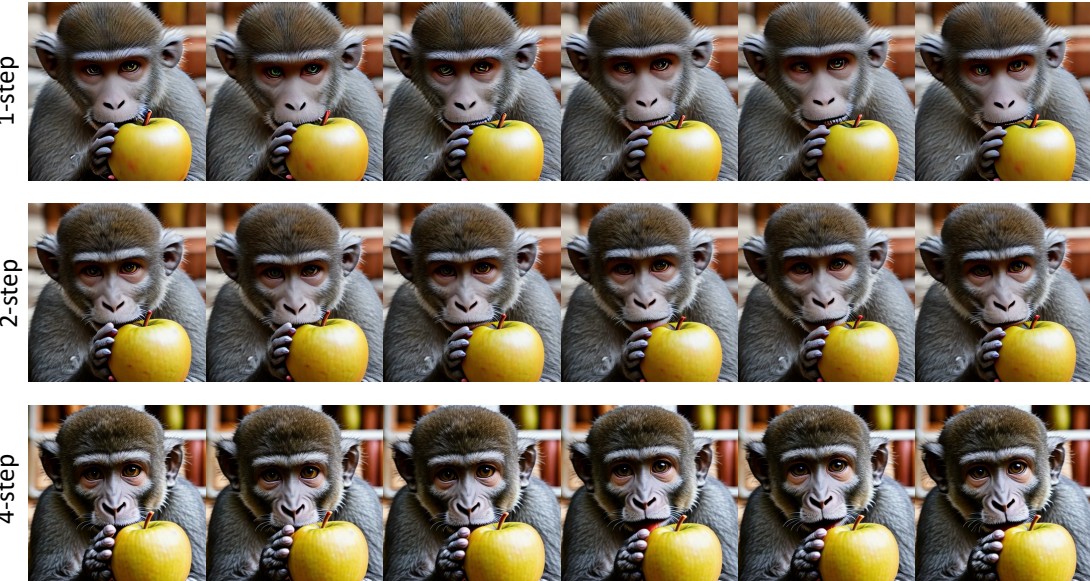

Figure 12: PCM video generation results with Stable-Diffusion v1-5 under $1 \sim 4$ inference steps.

Prompt: "Vincent vangogh style, painting, a boy, clouds in the sky"

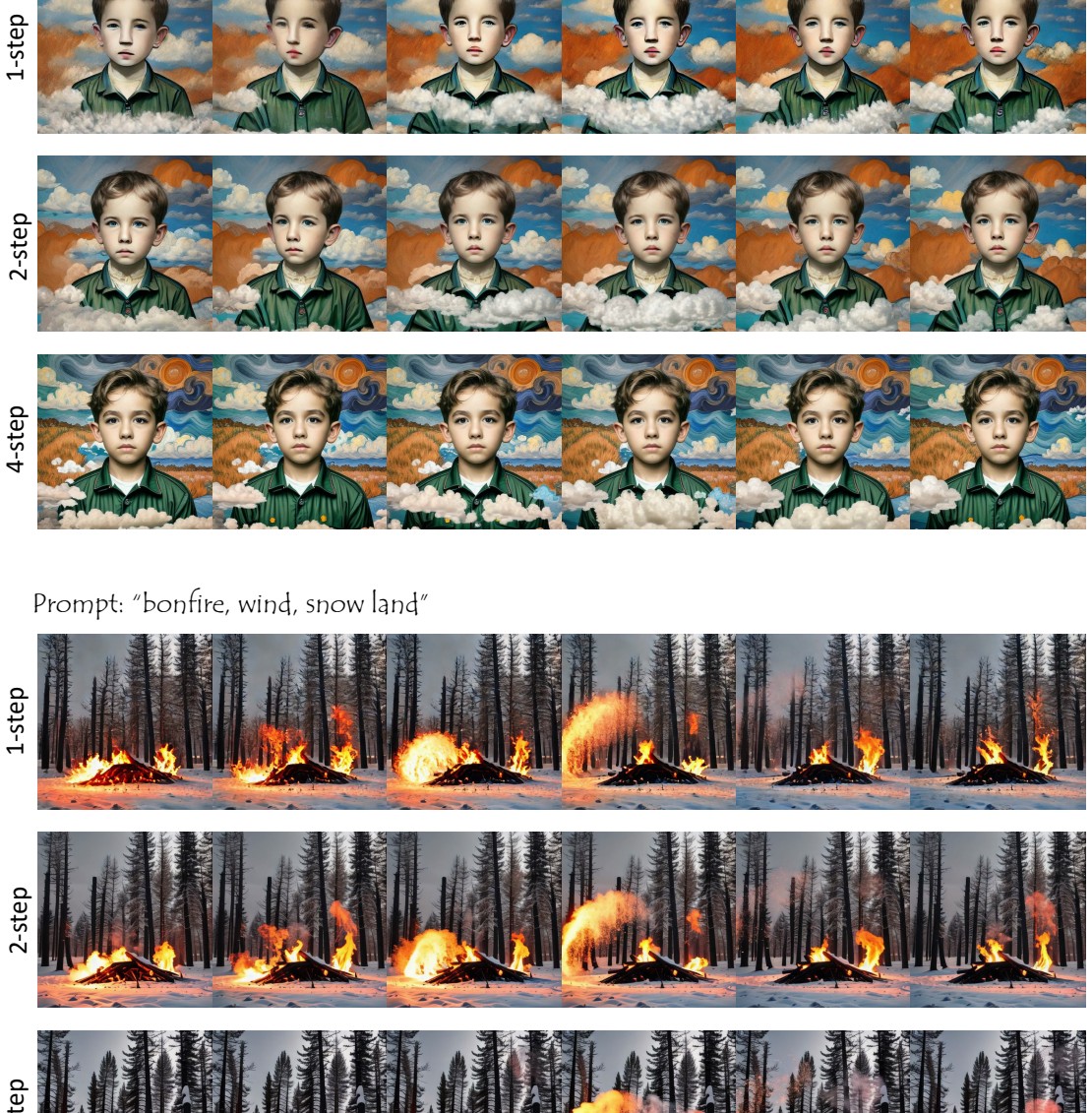

Prompt: "bonfire, wind, snow land"

Figure 13: PCM video generation results with Stable-Diffusion v1-5 under $1 \sim 4$ inference steps.

Prompt: "a lion on the grass land"

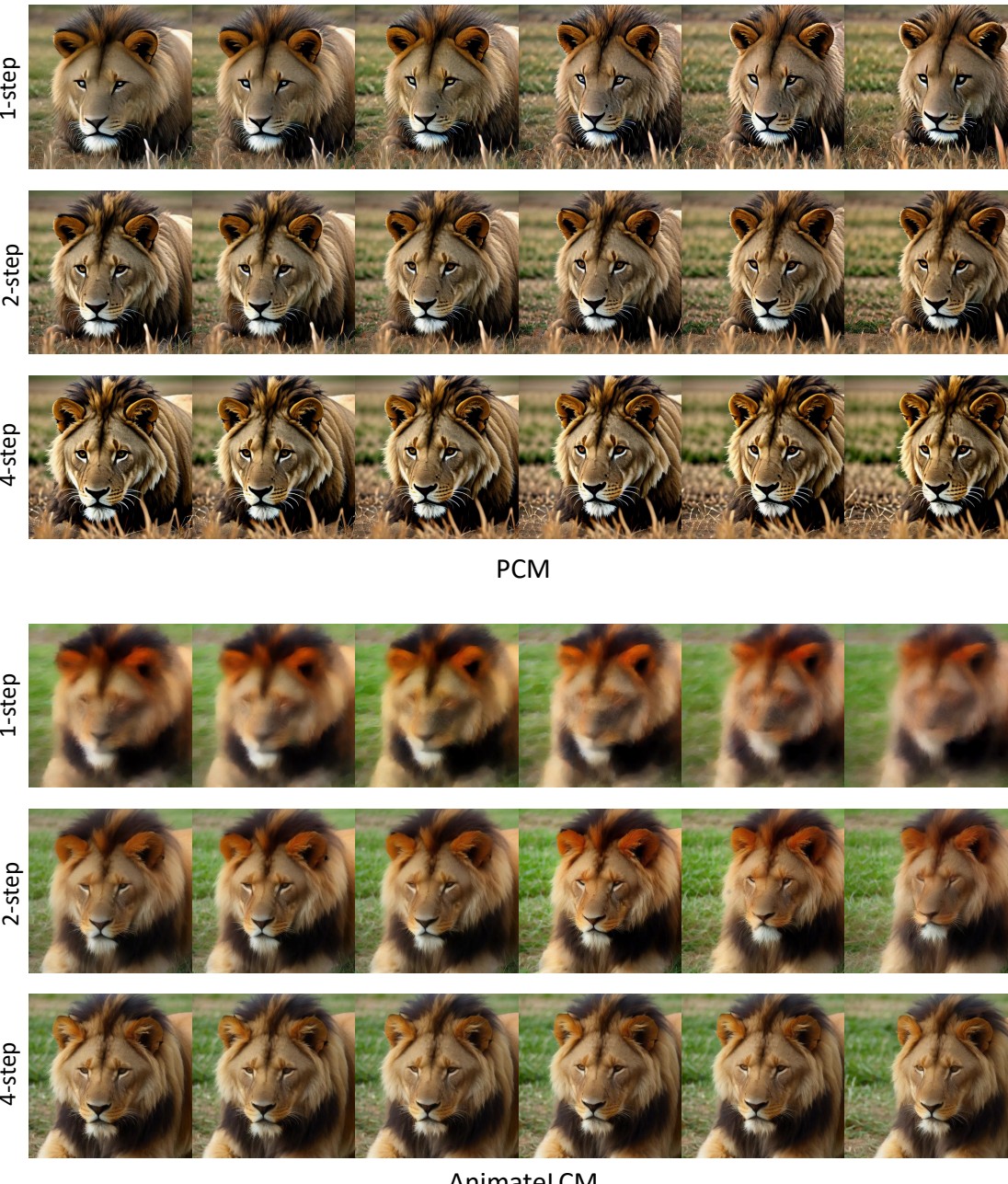

PCM

AnimateLCM

Figure 14: PCM video generation results comparison with AnimateLCM under $1 \sim 4$ inference steps.

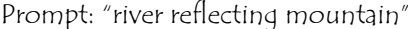

Prompt: "river reflecting mountain"

PCM

AnimateLCM

Figure 15: PCM video generation results comparison with AnimateLCM under $1 \sim 4$ inference steps.

1-step      2-step      4-step      8-step      16-step

Prompt: "a monkey living on the tree"

Prompt: "a lion"

Prompt: "heart-like pink cloud in the sky"

Prompt: "a cat"

Prompt: "a red car"

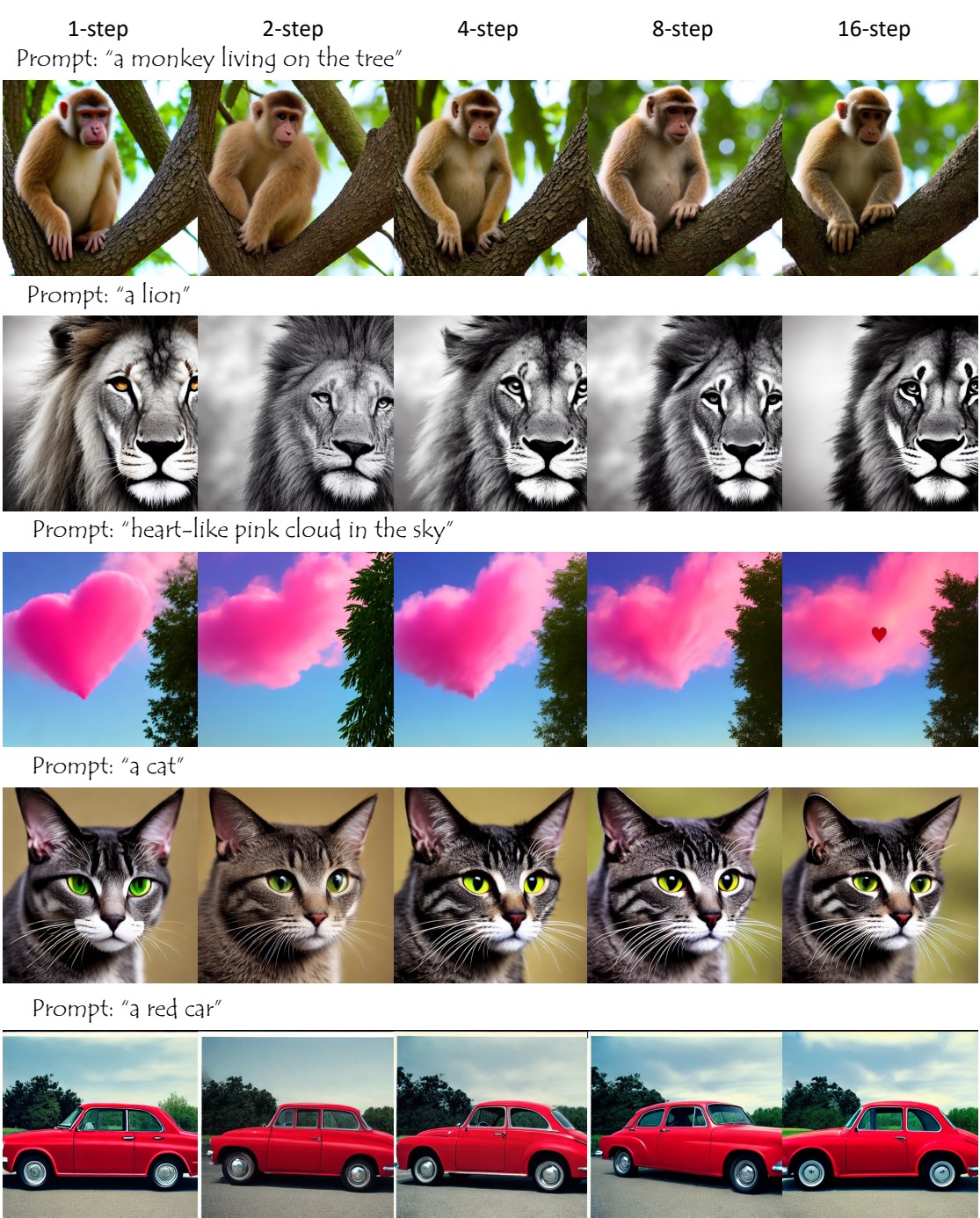

Figure 16: PCM generation results with Stable-Diffusion v1-5 under different inference steps.

1-step        2-step        4-step        8-step        16-step

Prompt: "a boy walking by the sea"

Prompt: "a dog"

Prompt: "a swimming pool"

Prompt: "a bike"

Prompt: "bedroom"

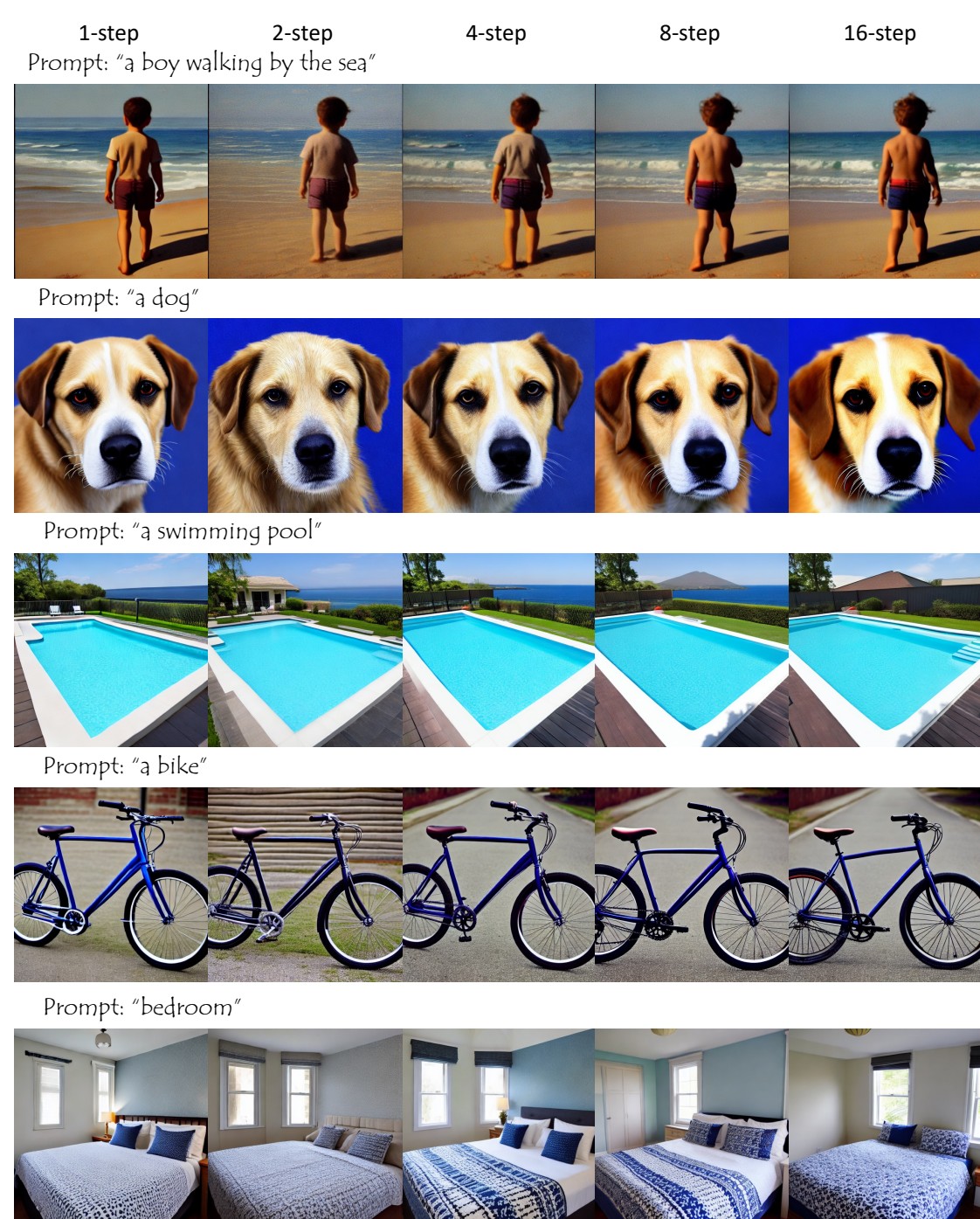

Figure 17: PCM generation results with Stable-Diffusion v1-5 under different inference steps.

| 1-step | 2-step | 4-step | 8-step | 16-step |
|--------|--------|--------|--------|---------|

Prompt: "a blue bed in the bottle, surrounded by clouds"

Prompt: "a cat near the sea"

Prompt: "a girl with white dress"

Prompt: "an ice made lion"

Prompt: "Son Goku made of marble"

Figure 18: PCM generation results with Stable-Diffusion XL under different inference steps.

|          1-step          |          2-step          |          4-step          |          8-step          |          16-step          |

Prompt: "a pink dog wearing blue sunglasses"

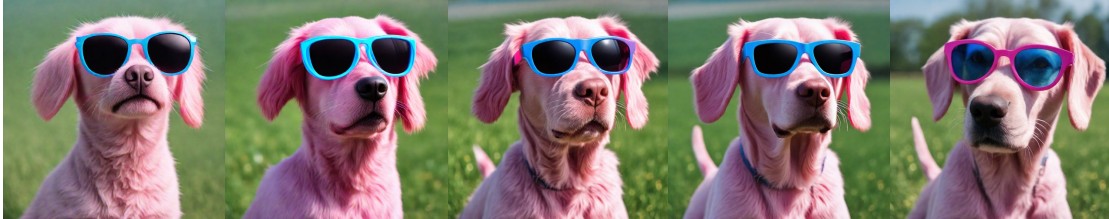

Prompt: "a dream island, surrounded by sea, with bird flying on the sky"

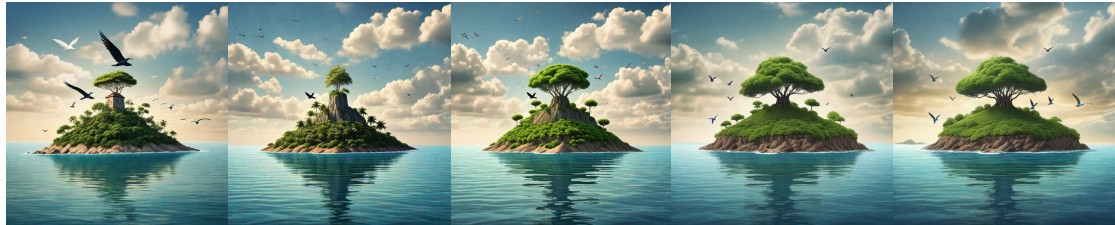

Prompt: "photography of a man kissing woman, vangogh style"

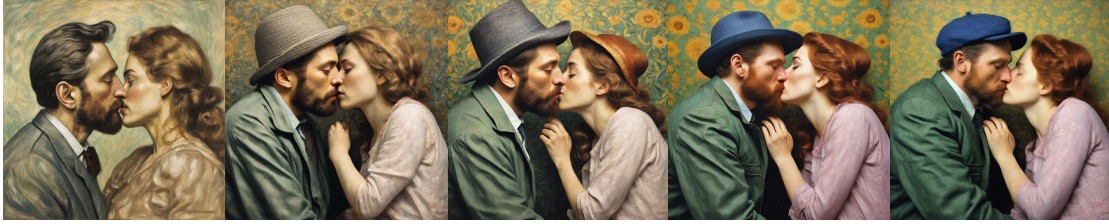

Prompt: "a pikachu made of wood"

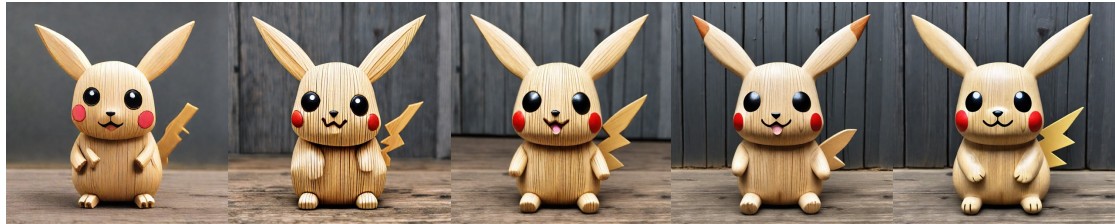

Prompt: "firework in the night, best quality, modern city, river reflecting"

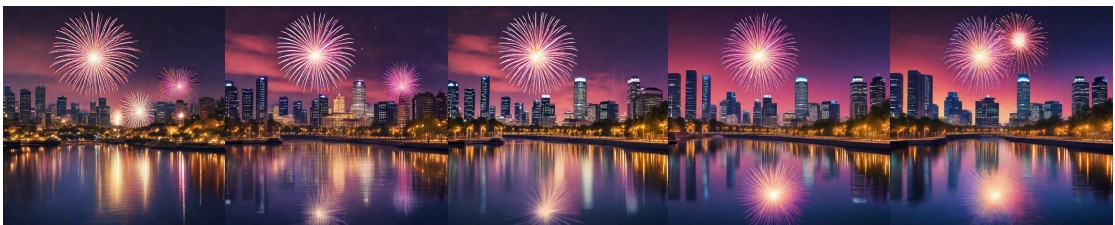

Figure 19: PCM generation results with Stable-Diffusion XL under different inference steps.

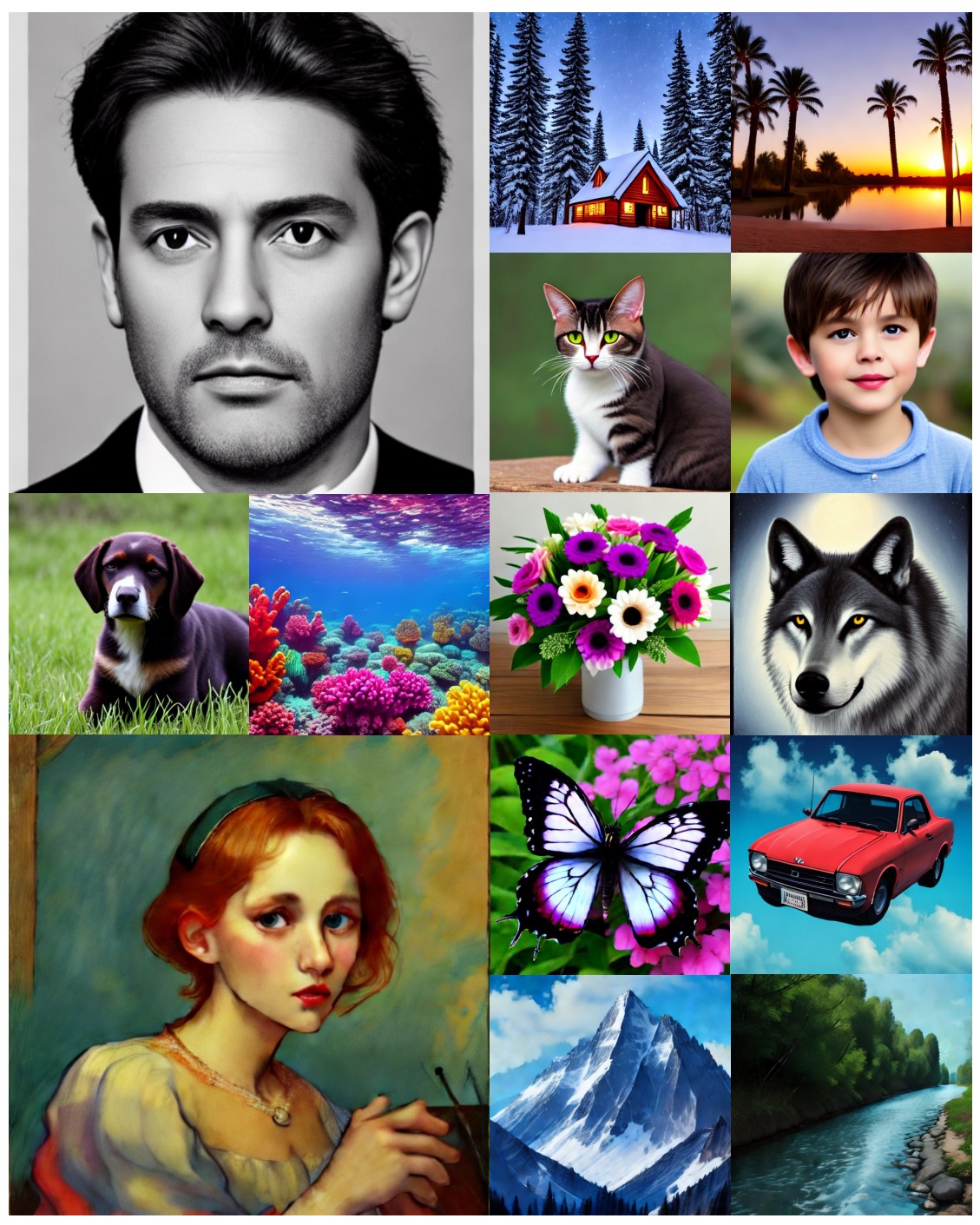

Figure 20: PCM generation 1-step results with Stable-Diffusion v1-5.

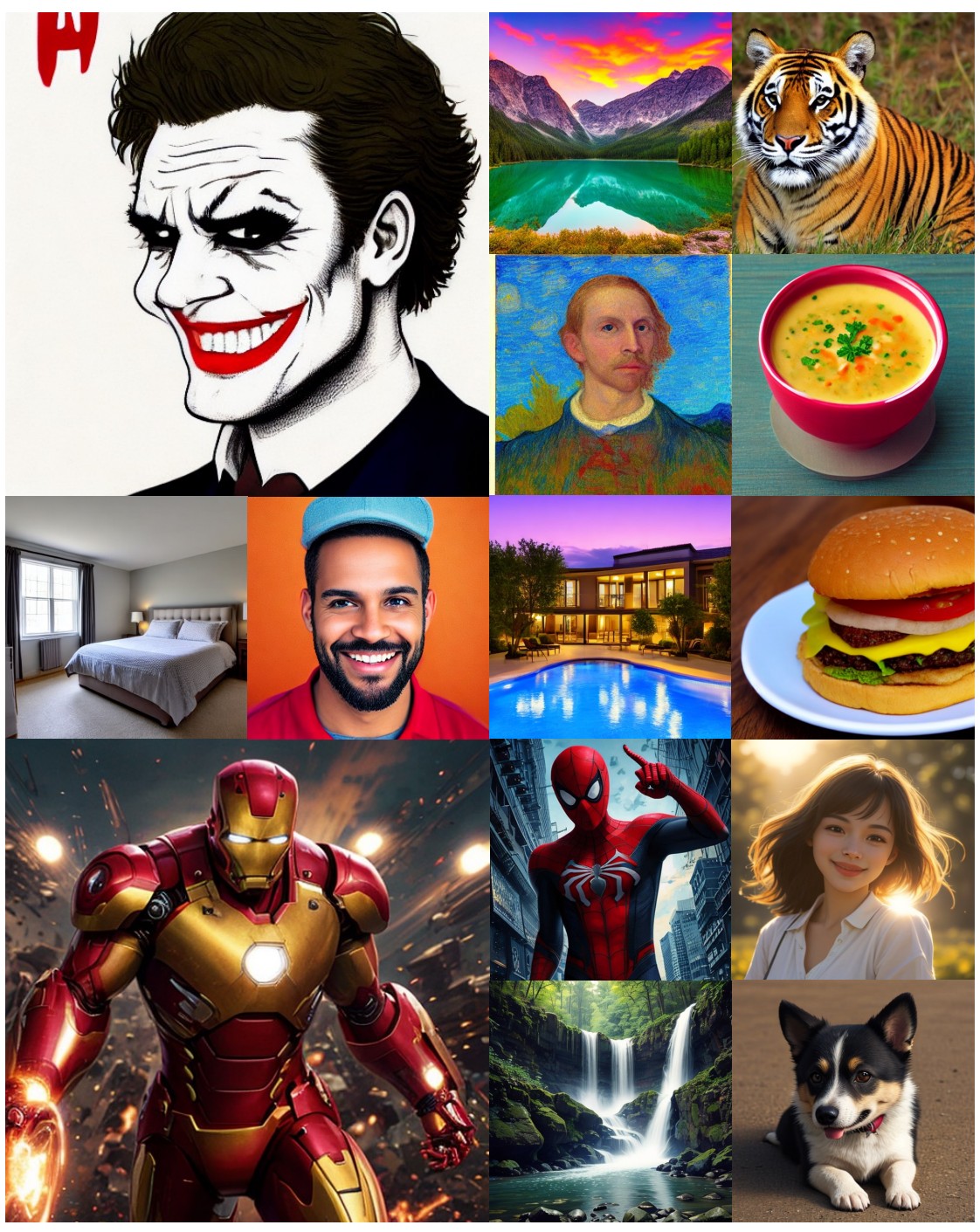

Figure 21: PCM generation 2-step results with Stable-Diffusion v1-5.

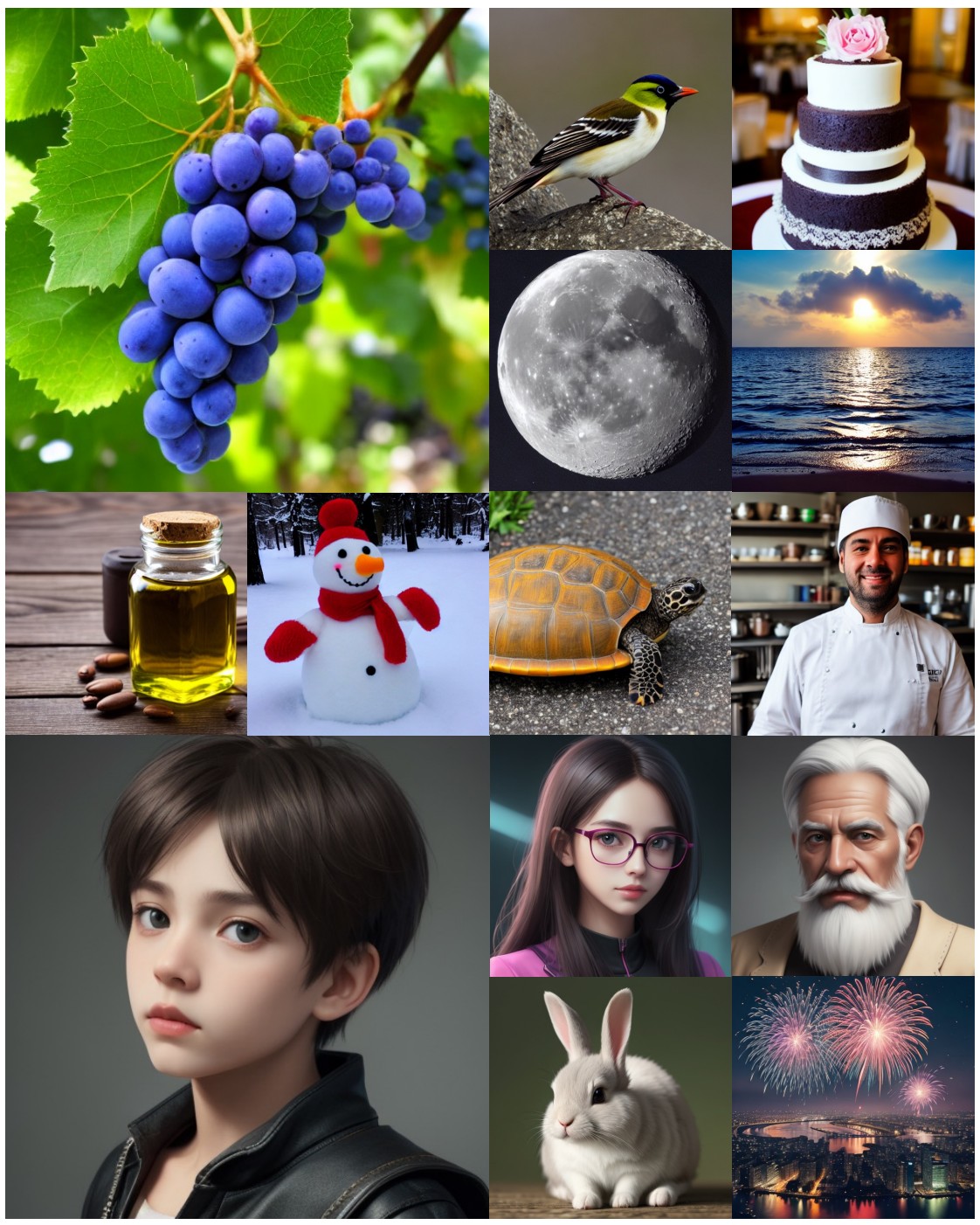

Figure 22: PCM generation 4-step results with Stable-Diffusion v1-5.

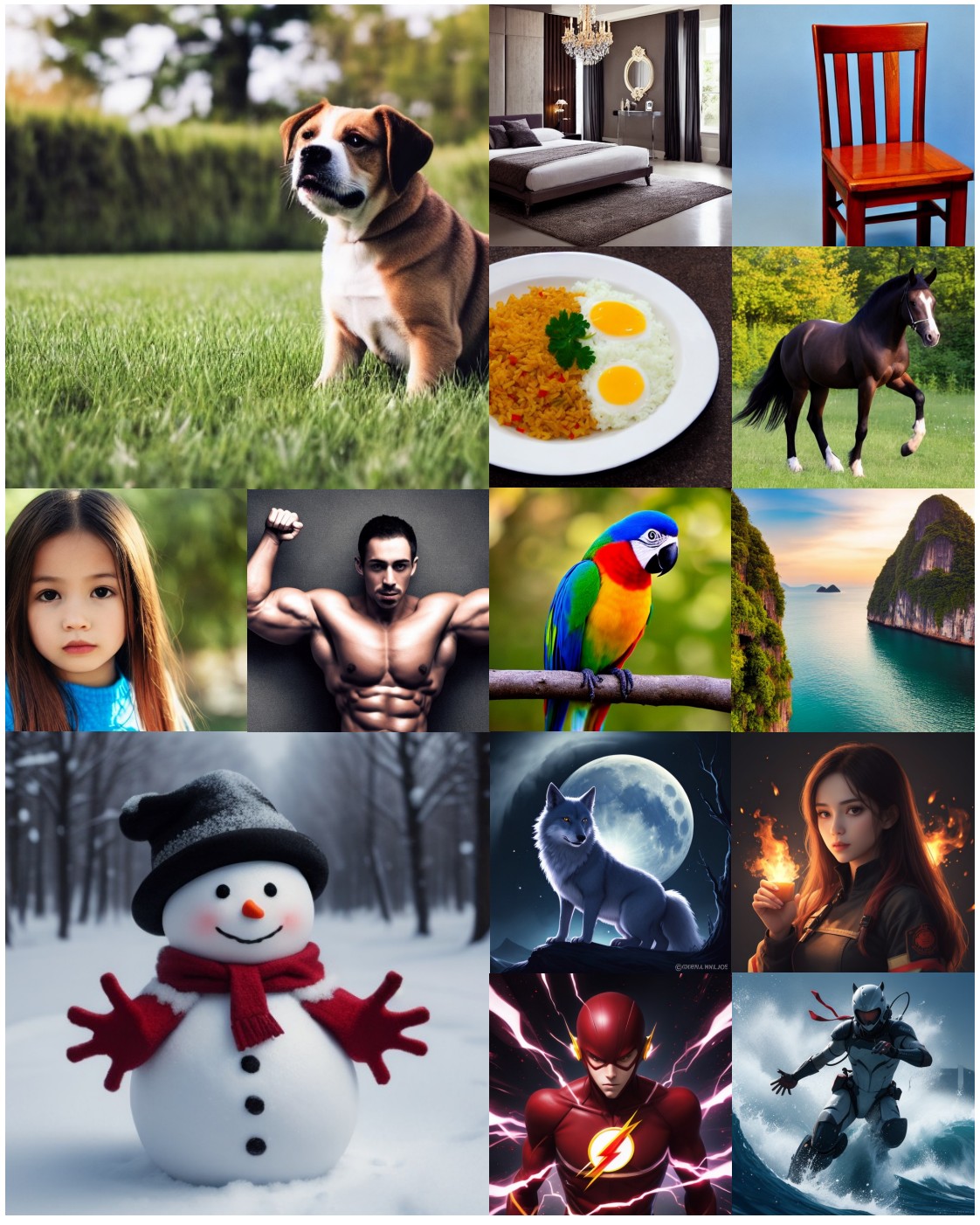

Figure 23: PCM generation 8-step results with Stable-Diffusion v1-5.

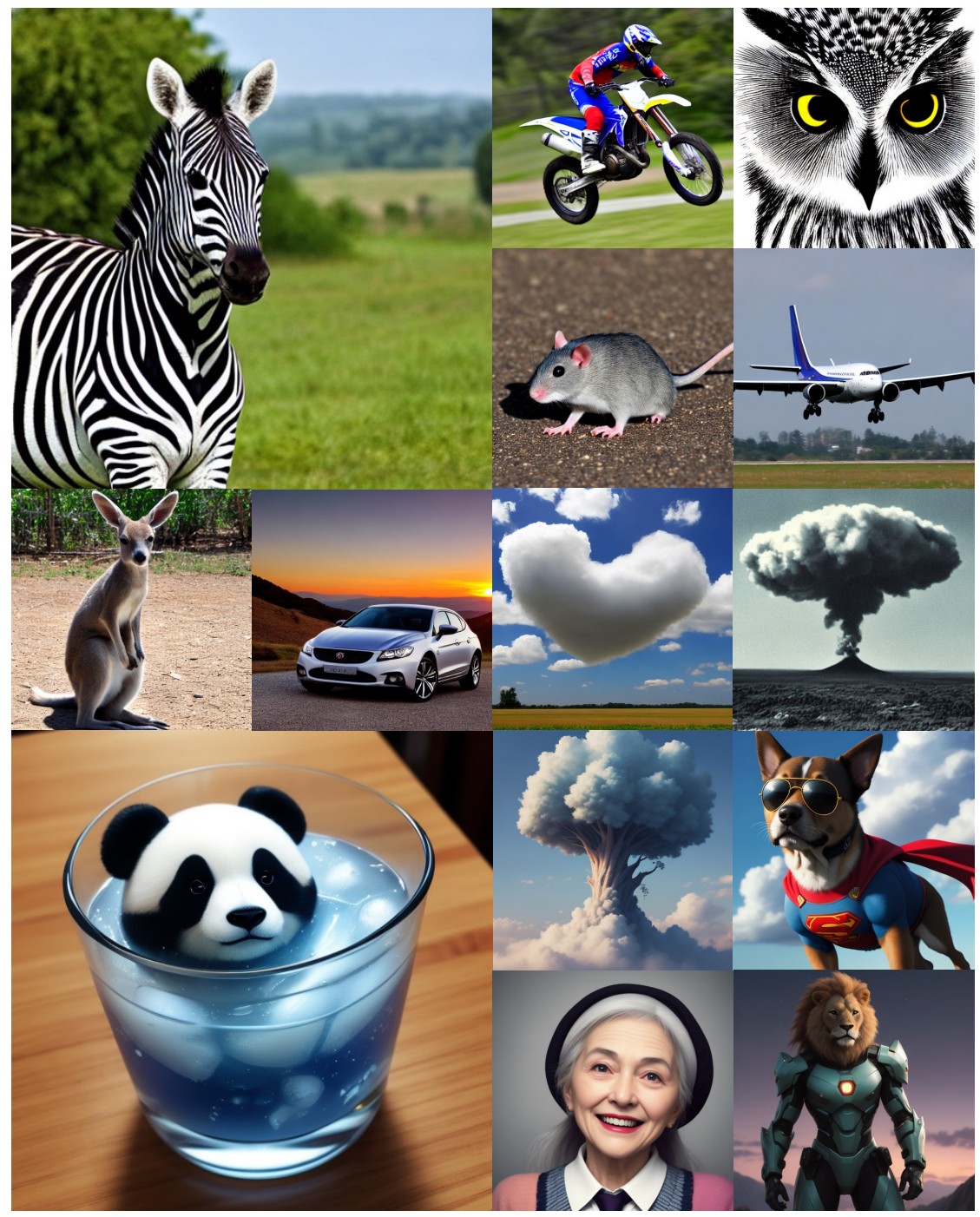

Figure 24: PCM generation 16-step results with Stable-Diffusion v1-5.

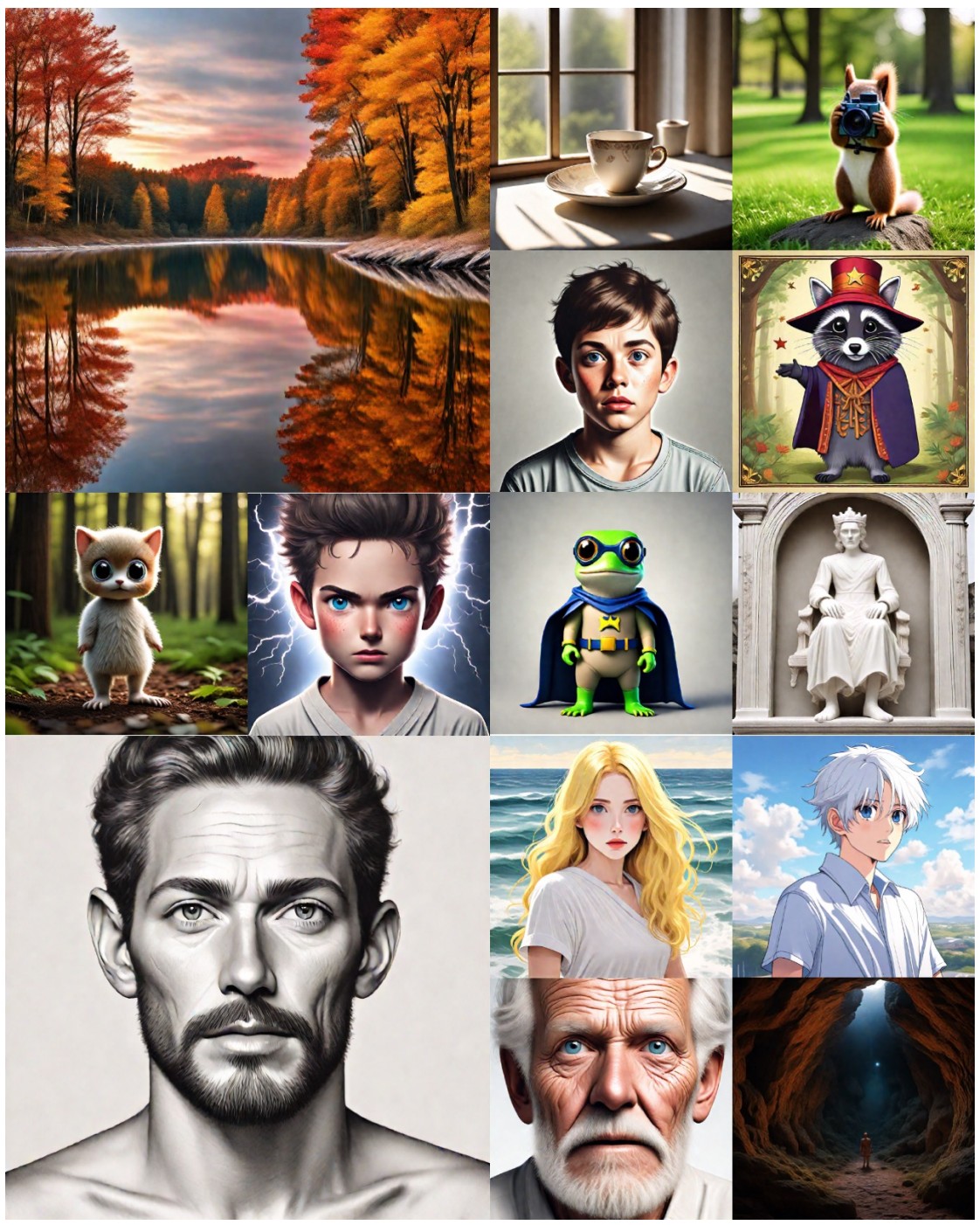

Figure 25: PCM generation 1-step results with Stable-Diffusion XL.

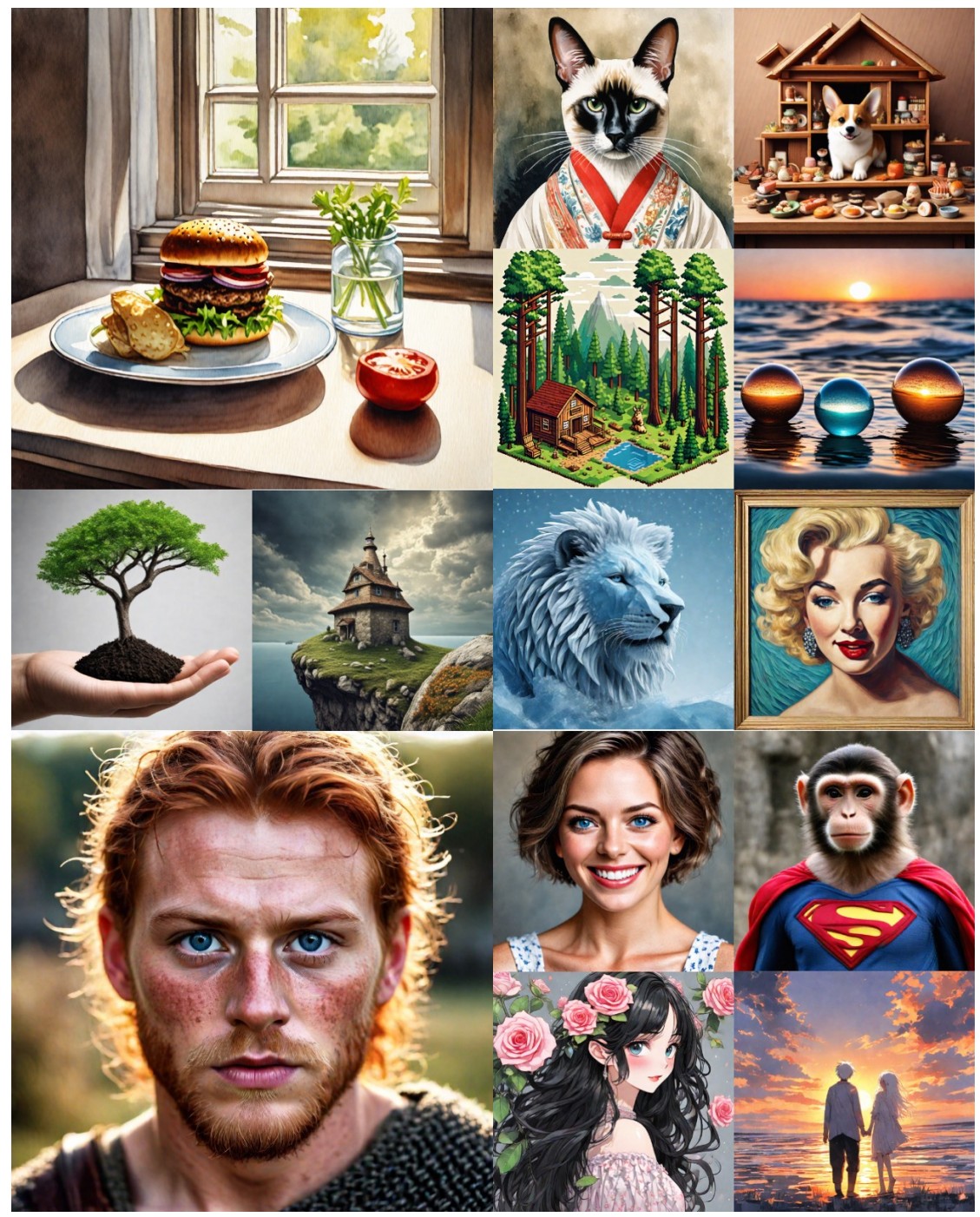

Figure 26: PCM generation 2-step results with Stable-Diffusion XL.

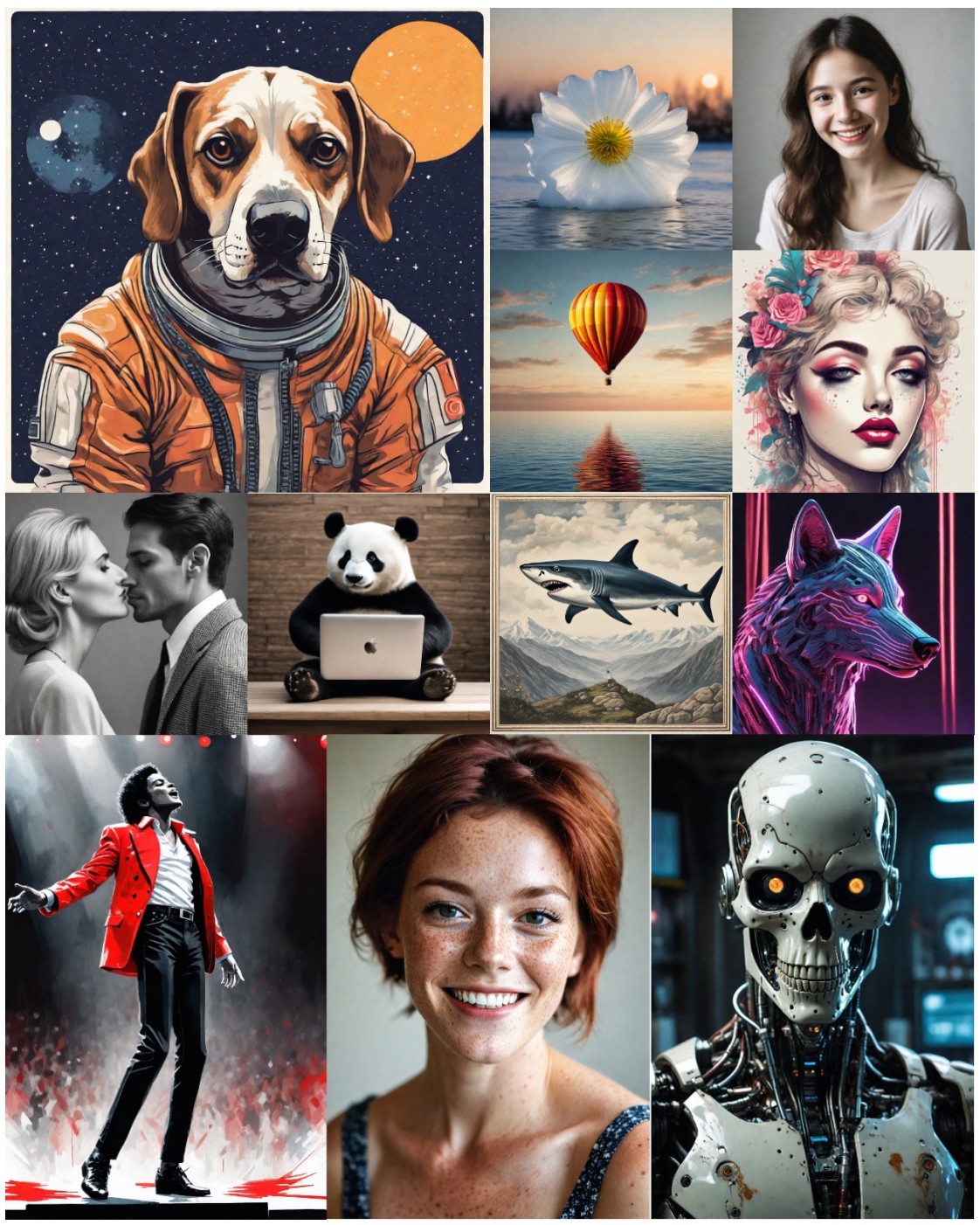

Figure 27: PCM generation 4-step results with Stable-Diffusion XL.

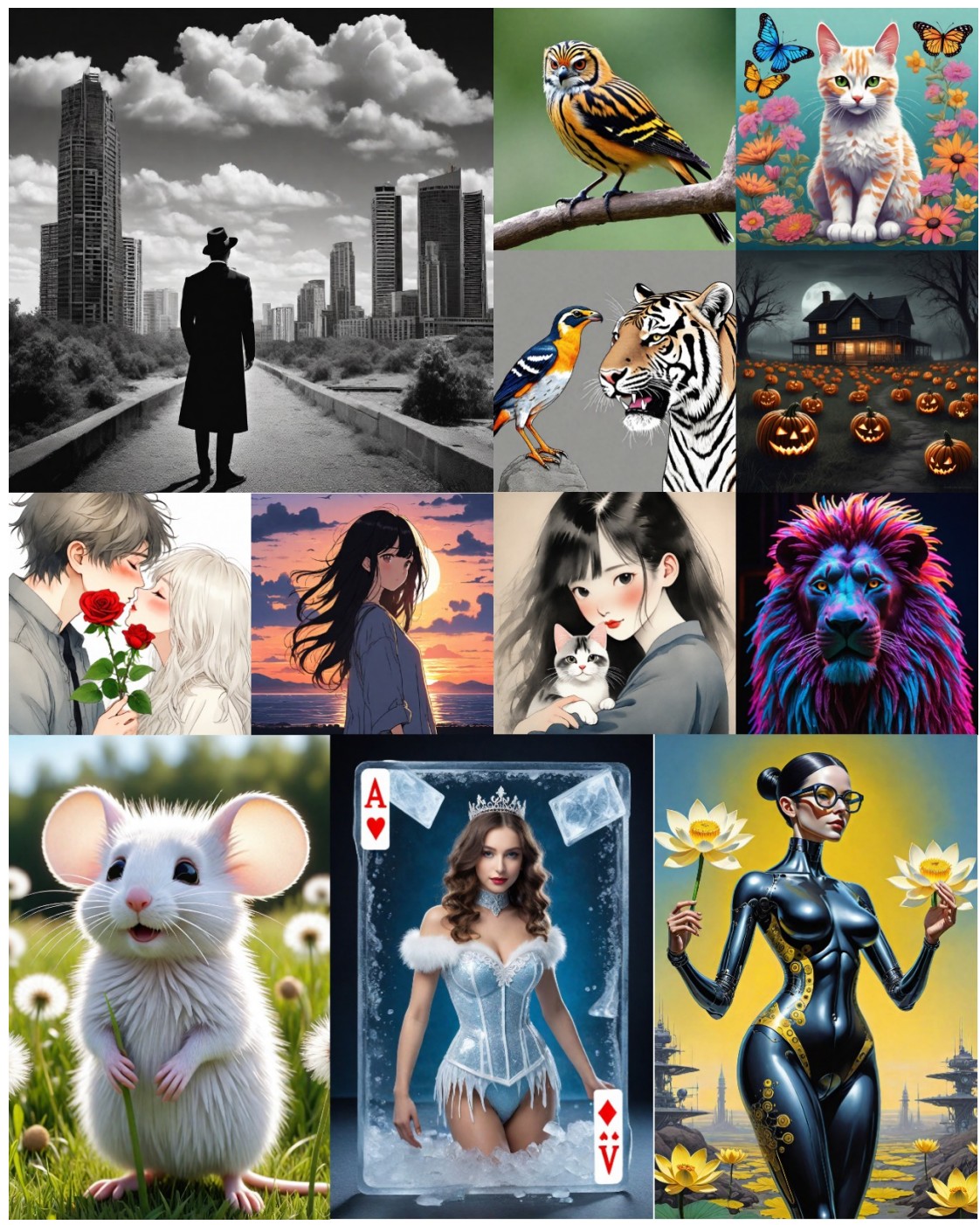

Figure 28: PCM generation 8-step results with Stable-Diffusion XL.

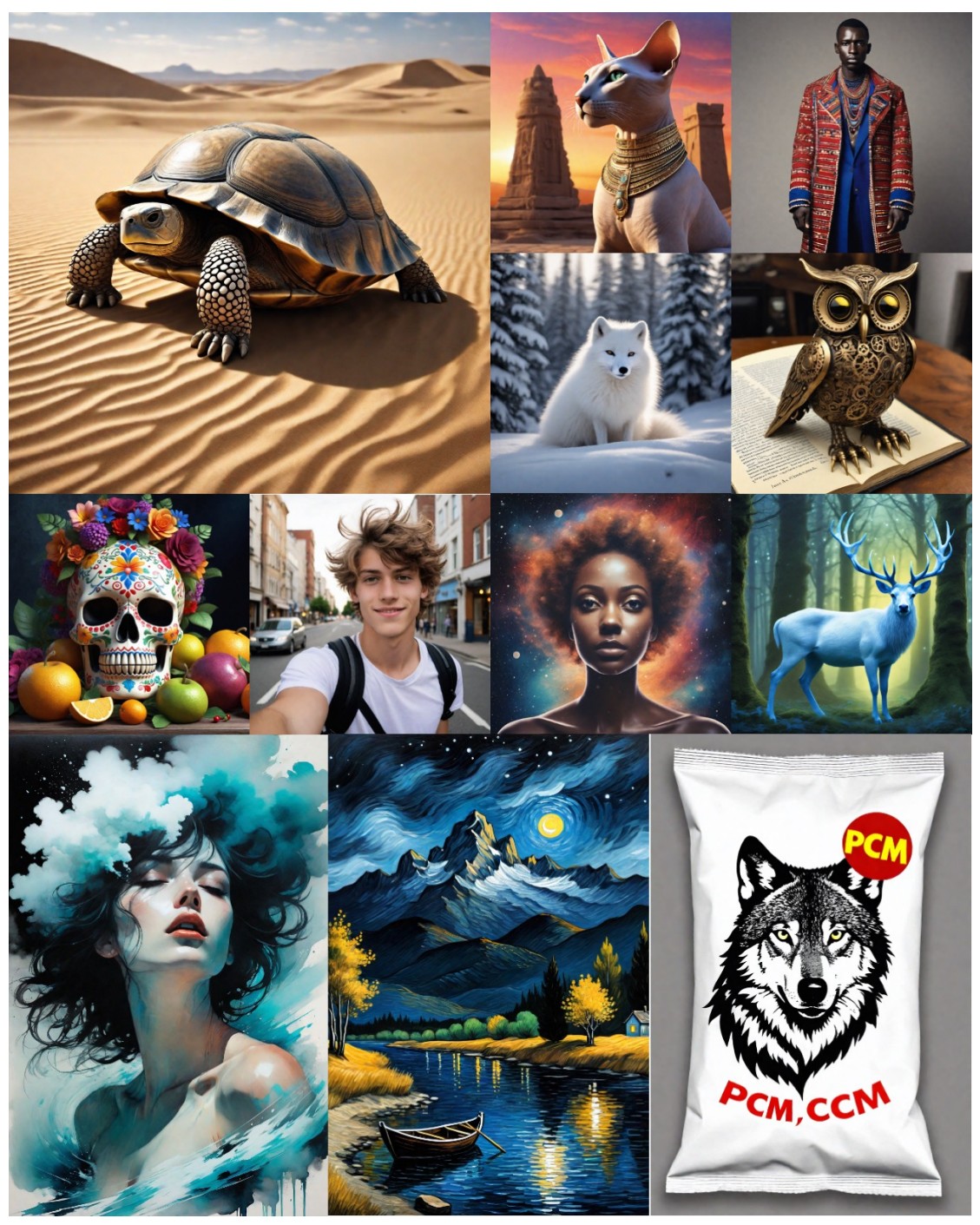

Figure 29: PCM generation 16-step results with Stable-Diffusion XL.

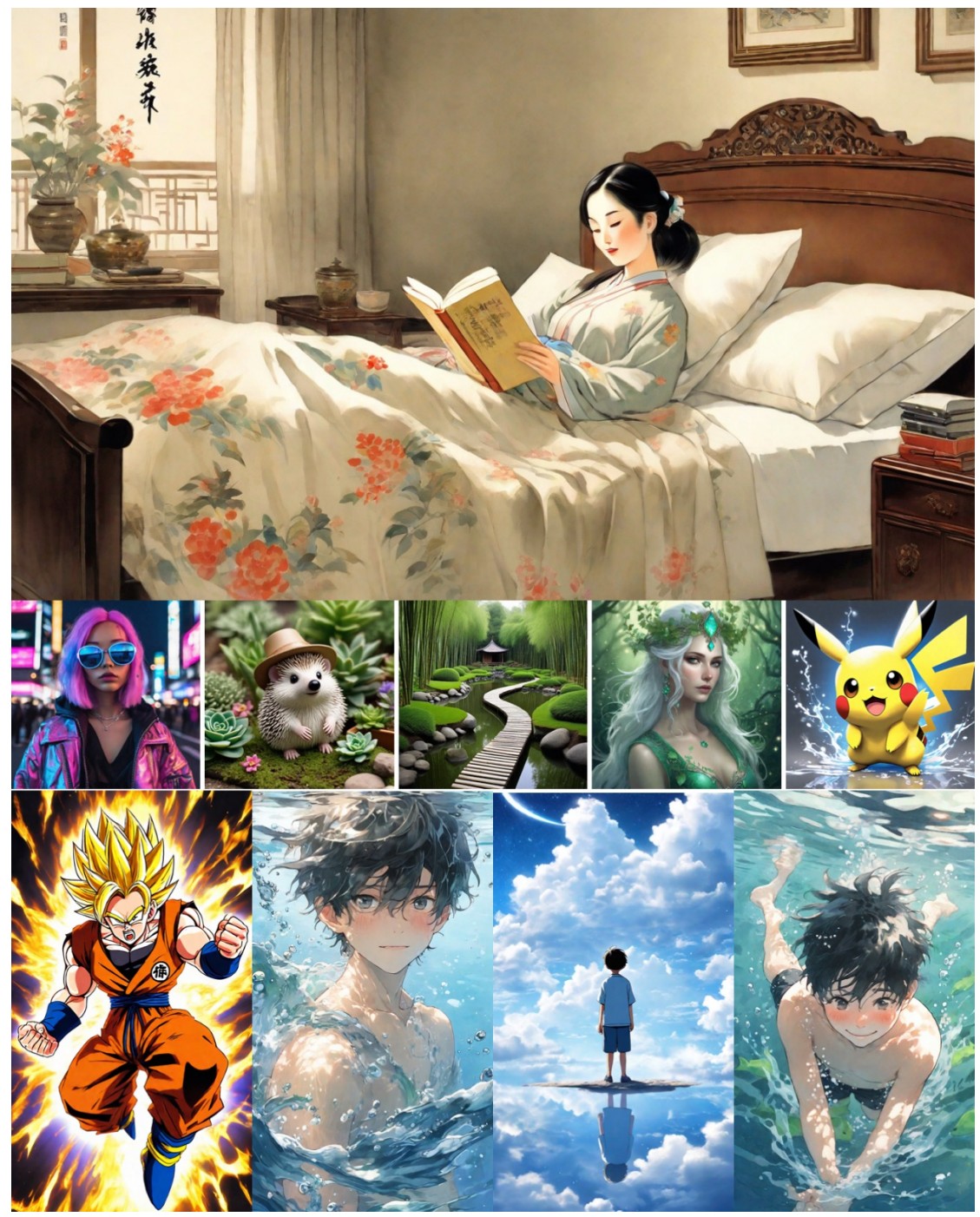

Figure 30: PCM generation 16-step results with Stable-Diffusion XL.

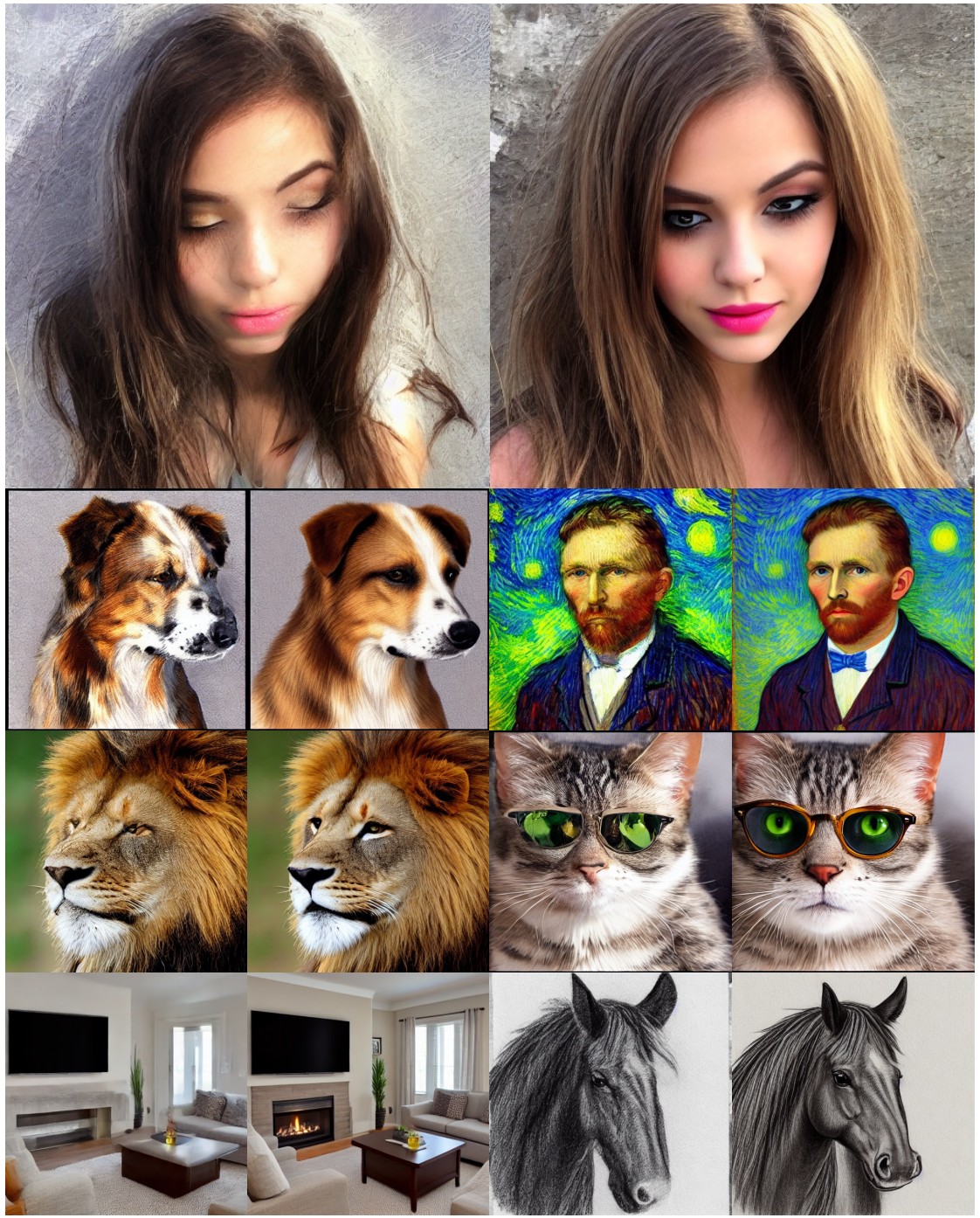

Figure 31: Visual examples of ablation study on the proposed distribution consistency loss. Left: Results generated without the distribution consistency loss. Right: Results generated with the distribution consistency loss.

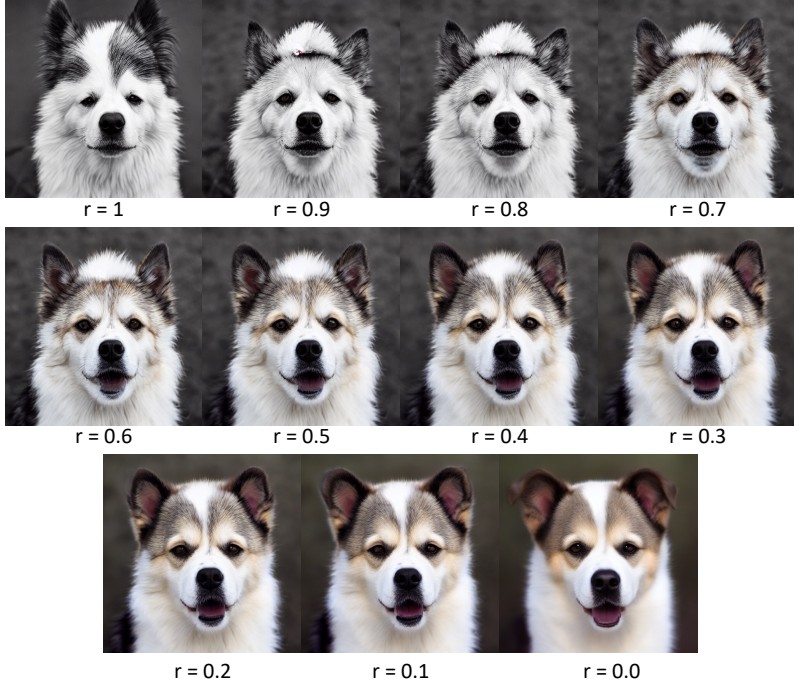

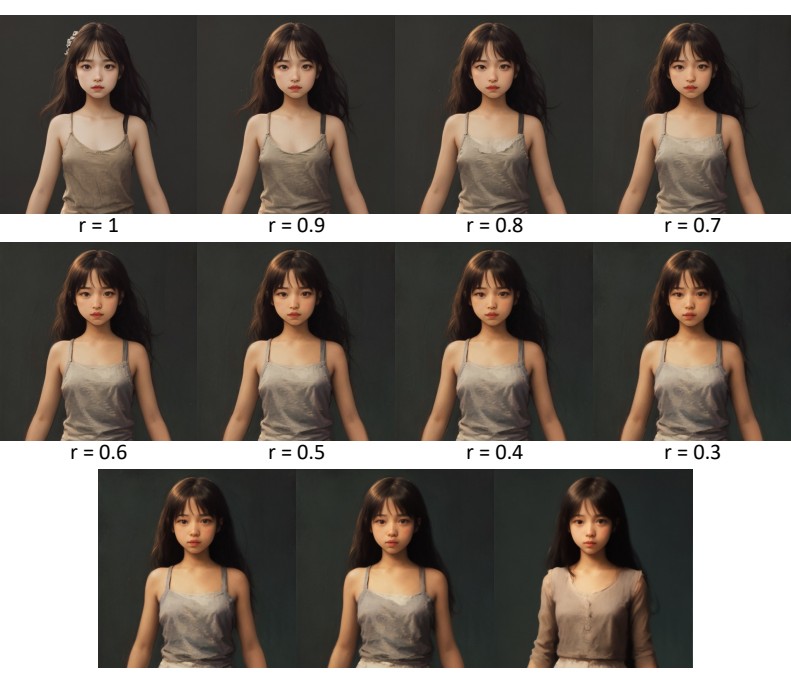

Figure 32: Visual examples of ablation study on the proposed way for stochastic sampling. Pure deterministic sampling algorithms sometimes bring artifacts in the generated results. Adding stochasticity to a certain degree can alleviate those artifacts.

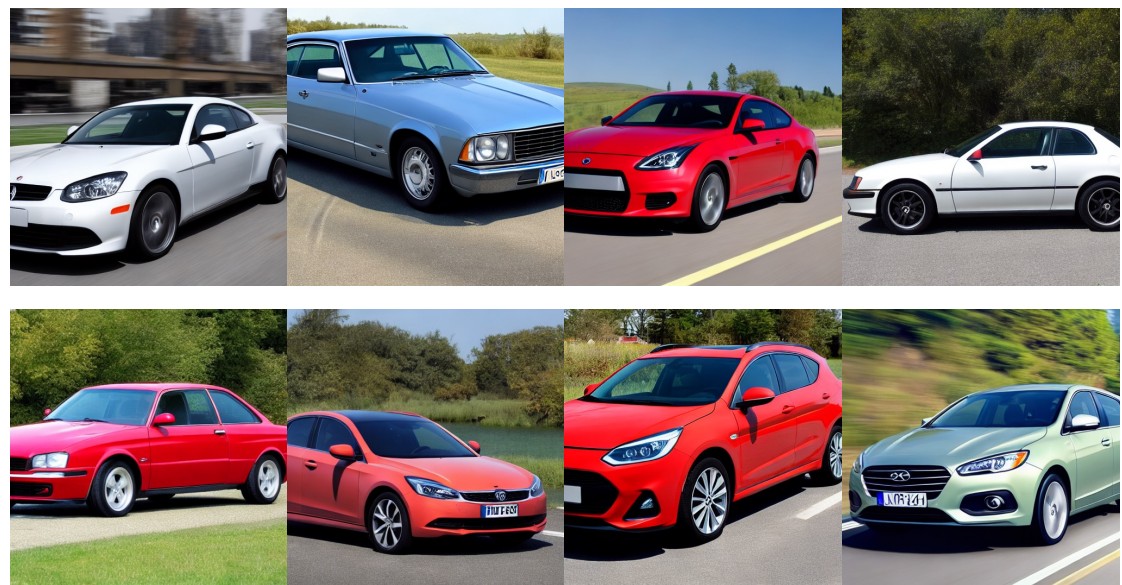

CFG value = 1.0. The entropy of color: 1.75

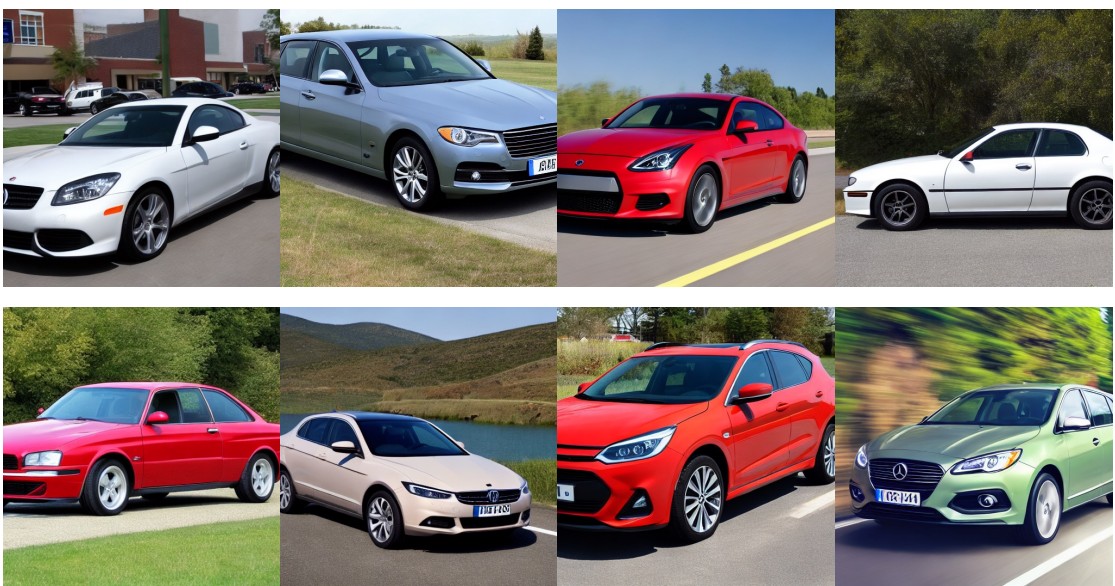

CFG value = 0.6. The entropy of color: 2.16

Figure 33: Visual examples of ablation study on the proposed strategy for promoting diversity. Upper: Batch samples generated with prompt "a car" with normal CFG=1.0 value in 4-step. Lower: Batch sample generated with prompt "a car" with our proposed strategy with CFG=0.6.

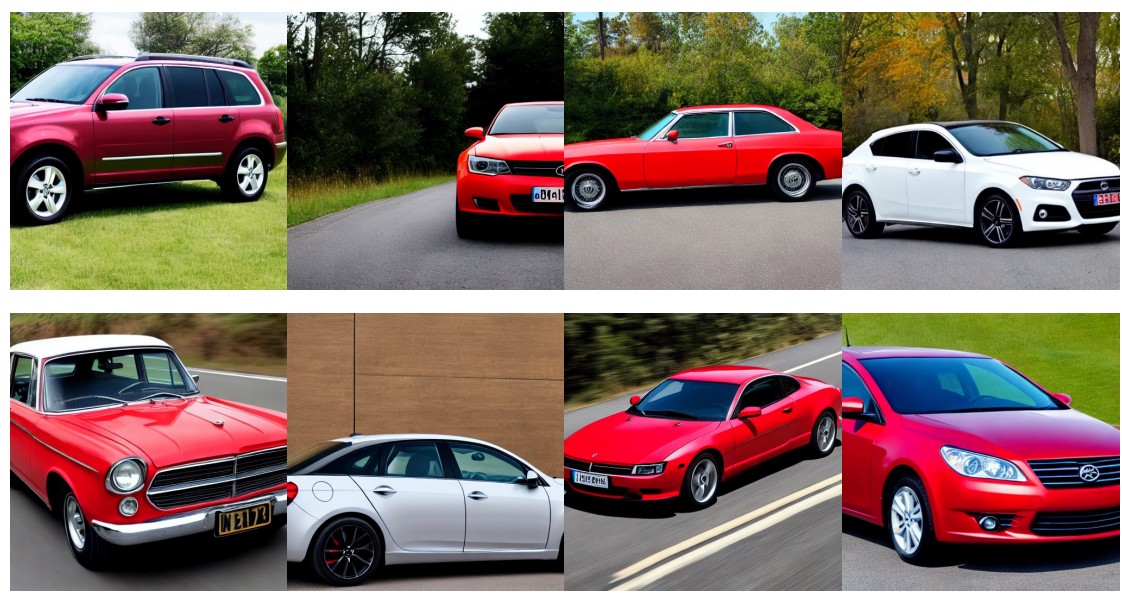

CFG value = 1.0. The entropy of color: 1.30

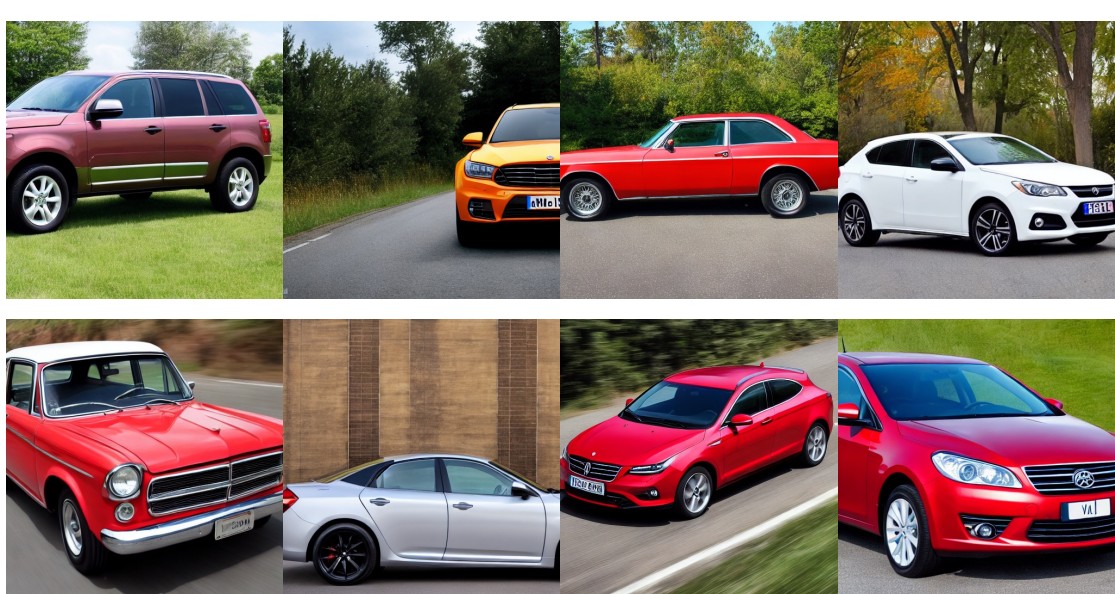

CFG value = 0.6. The entropy of color: 1.75

Figure 34: Visual examples of ablation study on the proposed strategy for promoting diversity. Upper: Batch samples generated with prompt "a car" with normal CFG=1.0 value in 4-step. Lower: Batch sample generated with prompt "a car" with our proposed strategy with CFG=0.6.

