# OpenReview forum: "Phased Consistency Models"
_NeurIPS.cc/2024/Conference — NeurIPS 2024 poster_

### Official Review · Reviewer_ukQK · 2024-07-03

**Soundness:** 2
**Presentation:** 2
**Contribution:** 1
**Rating:** 3
**Confidence:** 5

**Summary:**

The paper introduces the Phased Consistency Model (PCM), which enhances stability and speed in high-resolution image and video generation by improving the design of consistency (trajectory) models.
The main improvement of PCM is a new Parameterization of consistency function and the phase trajectory.  There is also a new method to improve the utilization of CFG.
Experimental results show that PCM performs better across 1 to 16-step generation settings.

**Strengths:**

1. The architecture of the method is straight-forward and clear.

2. The insights gained from observing the multi-step process are commendable.

3. The results are solid and demonstrate the effectiveness of the proposed approach.

4. Detailed proofs have been provided for all the hypotheses.

**Weaknesses:**

Despite the strong experimental results, I still have many reservations about this work.
The primary concern stems from the inaccurate description and the incomplete experiments.

Here's my MAJOR concerns:

1. The author also asserts that the findings are derived based on CTM. However, the author does not provide any comparisons with the official CTM, e.g., ImageNet or CIFAR.

2. Another concern is that the author claims to have compared the results with SD15 CTM and SDXL CTM. However, I could not find any related descriptions or implementations in the official CTM publications or GitHub repository.  The authors claimed such models are implemented by theirs but there's no detail. Specifically, the author stated they *additionally learned a timestep embedding to indicate the target timestep* but without any details and reasons.

3. A further question about implement detail: 1-step PCM and multi-step PCM are trained separately? If so, only the range of training parameters different? Are there any further differences?

4. The author appears to misunderstand the concept of the negative prompt, e.g. Controllability in the LCM drawbacks. Furthermore, the provided results do not demonstrate that PCM can mitigate the impact of the negative prompt.

5. The author does not clearly explain the differences between CTM and PCM, especially without a clear description of implementation of CTM in SD15 and SDXL.
Based on CTM, the novelty of this work on my side is 4-fold: a re-parameterization method, an improved GAN structure, an edged timestep and a CFG guidance method.


In conclusion,
the main problem is the authors **did not** conduct an objective comparison and the work lacks sufficient quantitative ablation analysis with official CTM on ImageNet or CIFAR. From the existing data, it is difficult for me to determine whether the proposed solutions by the authors are truly effective.


Here's some minor questions:

1. The author relied heavily on FID and CLIP metrics for comparison, but these have known limitations, particularly in large-scale text-to-image applications. More contemporary metrics, such as HPS or PICK-SCORE, should be used to demonstrate the effectiveness.

2. Additionally, I suggest author to use recall to demonstrate the diversity of the model. The introduction of additional GAN modules can lead to a lack of diversity in the model.

**Questions:**

please see Weaknesses

**Limitations:**

The author has already discussed limitations.

---

> ### Author Rebuttal · Authors · 2024-08-06
>
> Thank you for the in-depth review. We appreciate the chance to more comprehensively compare PCM with CTM.
>
> Q1:
>
> **CIFAR-10**
>
> We additionally implement the core idea "phasing" technique into CIFAR (unconditional generation) and ImageNet (conditional generation). For CIFAR, we train our models with a batch-size 256 for 200, 000 iterations (800, 000 iterations used in official CM), without any additional GAN training. We listed the FID and inception score comparison in the following table.
>
> | Methods       | NFE  | FID  | IS |
> | - | - | - | - |
> | EDM           | 35   | 2.04 | 9.84            |
> | Score SDE     | 2000 | 2.20 | 9.89            |
> | CD            | 1    | 3.55 | 9.48            |
> | CD            | 2    | 2.93 | 9.75            |
> | CTM w/ GAN ++ | 1    | 1.98 | -               |
> | CTM w/ GAN ++ | 2    | 1.87 | -               |
> | CTM w/ GAN    | 1    | 2.28 | -               |
> | CTM w/ GAN    | 18   | 2.23 | -               |
> | CTM w/o GAN   | 1    | 5.19 | -               |
> | CTM w/o GAN   | 18   | 3.00 | -               |
> | Ours w/o GAN  | 1    | 2.95 | 9.72            |
> | Ours w/o GAN  | 2    | 2.14 | 9.76            |
> | Ours w/o GAN  | 4    | 2.04 | 9.90            |
> | Ours w/o GAN  | 8    | 1.95 | 9.92            |
>
> In the above table, "CTM w/ GAN" and "CTM w/o GAN" are obtained from Table 3 of supplementary of CTM paper, tagged with "**fair comparison**" in the title. "CTM w/ GAN ++" is obtained from Table 1 of the paper, with more advanced techniques including classifier rejection sampling. We did not apply these techniques to our method. We can observe that the success of CTM greatly relies on GAN training. Yet, we show that our method, without any GAN training, can achieve comparable performance through multistep refinement.
>
> **ImageNet**
>
> | Methods                              | NFE  | FID  | Recall |
> | -| - | - | - |
> | EDM                                  | 79   | 2.44 | 0.67   |
> | ADM                                  | 250  | 2.07 | 0.64   |
> | BigGAN-deep                          | 1    | 4.06 | 0.48   |
> | StyleGAN-XL                          | 1    | 2.09 | 0.52   |
> | CD (L2)                         | 1    | 12.5 | -      |
> | CD (L2)                         | 2    | 6.8  | -      |
> | CD (L2)                         | 4    | 5.2  | -      |
> | CD (LPIPS)                           | 1    | 6.2  | 0.63   |
> | CD (LPIPS)                           | 2    | 4.7  | 0.64   |
> | CD (LPIPS)                           | 4    | 4.3  | 0.64   |
> | CTM w/ GAN ++                        | 1    | 1.92 | 0.57   |
> | CTM w/ GAN ++                        | 2    | 1.73 | 0.57   |
> | Ours w/o GAN (3.5 % Training budget) | 4    | 3.7  | 0.66   |
>
> For ImageNet, due to the limited time and GPU resources, we can only afford to run a tentative experiment. Specifically, we run a 4-phased PCM by adapting the official repo of CM with a normal L2 loss. For the last phase, we apply LPIPS loss. We first train the model for 100,000 iterations with a batch size of 64 and further train the model for 50, 000 iterations with a batch size of 1024. Note that the official CM trains the model with batch size 2048 for 800, 000 iterations. Therefore the current training budget for our method is 3.5% of the official CM. But importantly, we achieve even superior performance than the official CD. We believe this is clear evidence of the effectiveness of the "Phasing" ODE trajectory.
>
> Q2 & Q5:  The implementation details of CTM are the same as LCM and PCM, with the only difference on the loss design. We use the following table to show the difference.
>
> | Methods | timestep sampled                                      | previous timestep | target timestep                                    |
> | - | - | -| - |
> | CM      | $t_{n+1}\sim\rm{Uniform}(\{t_1, t_2, \dots\})$     | $t_{n}$            | $s:=t_1:=0$                                         |
> | CTM     | $t_{n+1}\sim\rm{Uniform}(\{t_1, t_2, \dots\})$     | $t_{n}$            | $s\sim\rm{Uniform}(\{t_1, t_2, \dots, t_{n}\})$ |
> | PCM     | $t_{n+1}\sim\rm{Uniform}(\{s_m, \dots, s_{m+1}\})$ | $t_{n}$            | $s := s_{m}$                                        |
>
> where $s_m, m=0,1,\dots$  are pre-defined edge timesteps. Note that for both LCM and PCM, the target timesteps are fixed. While in CTM, the target timesteps are randomly chosen. Therefore it requires target timesteps embeddings. Besides, in the official CTM code base, the model can also optionally receive the target timestep embeddings. We also provide more theoretical proof in our submission, please see "III.2 Why CTM Needs Target Timestep Embeddings?".  To put it straightforwardly, CTM does not require a target timestep embedding if and only if the ODE trajectory is first-order ( i.e., it is straight), which violates the truth since ODE trajectories are typically higher orders.
>
> Q3: Training. No further difference. The LoRA has limited capacity, and therefore it is not enough for stable one-step generation.
>
> Q4: Negative prompt. Please refer to Ln 289-296 ablation study of our submission and Figure 6. We show that our model has a higher CLIP score on positive prompts lower CLIP  score on negative prompts compared with LCM.
>
> Q6: Human metrics. Please refer to our rebuttal to Reviewer tuze.
>
> Q7: Regarding recall, we have the following interesting findings. When computing the recall with coco, we find the introduced adversarial loss reduces the recall slightly. But when computing the recall with generation results of the teacher diffusion models, we find the adversarial loss increases the recall at a low-step regime, which we attributed to the quality improvements.
>
> | Dataset |    | 1step | 2step | 4step | 8step | 16step |
> |--|-|-|-|-|-|-|
> | COCO    | adv         | 0.535  | 0.679 | 0.626 | 0.611 | 0.609   |
> |    | w/o adv | 0.585  | 0.690 | 0.686 | 0.647 | 0.637   |
> | SD      | adv         | 0.819 | 0.824 | 0.813  | 0.821  | 0.825   |
> | | w/o adv | 0.608 | 0.707 | 0.793  | 0.849  | 0.861   |

---

> > ### Comment · Reviewer_ukQK · 2024-08-10
> >
> > I don’t believe the author’s response has adequately addressed my concerns. The paper still requires substantial experimental validation before it can be considered for publication. The author repeatedly compares their work with CTM throughout the paper, yet I found no clear advantages in the data provided in the rebuttal. This raises serious doubts about the paper’s contribution. As a result, I will be lowering my score.

---

> ### Author Response · Authors · 2024-08-10
> **Thank you for your response!**
>
> Thank you for your response!  Could you specify `which parts are not resolved properly`? We are willing for further reply and discussion.  For the data, we already showed that our model without applying any adversarial loss consistently surpassing the CTM w/ GAN on CIFAR through multistep refinement.  Additionally, we show that our model trained with only 3.5% training budget of the official CM achieved even superior performance.

---

> ### Author Response · Authors · 2024-08-10
> **We are willing for further discussion!**
>
> Besides, when comparing each baseline for fair, the principle design of CTM without GAN loss is even inferior to naive CM.
>
> The core difference of CM, CTM and PCM are listed as the following table.
>
> | Methods | timesteps sampled                                      | adjacent timesteps | target timesteps                                    |
> | ------- | ------------------------------------------------------ | ------------------ | --------------------------------------------------- |
> | CM      | $t_{n+1}\sim\textrm{Uniform}(\{t_1, t_2, \dots\})$     | $t_{n}$            | $s:=t_1:=0$                                         |
> | CTM     | $t_{n+1}\sim\textrm{Uniform}(\{t_1, t_2, \dots\})$     | $t_{n}$            | $s\sim\textrm{Uniform}(\{t_1, t_2, \dots, t_{n}\})$ |
> | PCM     | $t_{n+1}\sim\textrm{Uniform}(\{s_m, \dots, s_{m+1}\})$ | $t_{n}$            | $s := s_{m}$                                        |
>
> In our paper, we have provided `thorough analysis and discussion with CTM`. The random choice design of target timestep make the design of CTM :
> - more complex (additionally target timestep embedding)
> - need more capacity ($\mathcal O (N^2)$ learning objectives)
> - and harder to train (inferior performance without GAN loss)
>
> Please let us know which part do you think is not convincing enough, we are willing for further discussion!

---

> ### Author Response · Authors · 2024-08-10
> **We are willing for further discussion! CTM metrics needs to generate much more samples than compared methods, which is unfair to compare directly.**
>
> **The main metrics listed in the Table 1 of original CTM paper is augmented with more advanced sampling techniques while all the compared methods do not use them**. CTM applies a pretrained image classification model to score the generated samples, and only retain the sample with higher confidence and regenerate new samples if the confidence of the classification model is low. That means, `It needs to generate much more samples than all the compared methods`. It is noting that FID will generally become smaller when the number of samples become larger.

---

> ### Author Response · Authors · 2024-08-10
> **We are willing for further discussion! We validate the design of PCM on much more baselines than all compared papers!**
>
> Except for the validation on CIFAR and ImageNet. We hope to emphasize that we validate the design of PCM on much more baselines for all compared methods.
>
> PCM (Ours): 1. `CIFAR-10`, 2. `ImageNet`, 3. `SD-v15`, 4. `SDXL`, 5.`Text-to-Video`
>
> Instaflow: SD-v15
>
> SDXL-Lightning: SDXL
>
> CTM: CIFAR, ImageNet
>
> CM: CIFAR, ImageNet
>
> SDXL-Turbo: SDXL
>
> LCM: SD v1-5, SDXL
>
> AnimateLCM: Text-to-Video
>
> We sincerely hope any further chance for discussion! Looking forward to your reply!

---

> ### Comment · Reviewer_ukQK · 2024-08-10
> **A suspicion of potential plagiarism.**
>
> I respectfully request that the other reviewers and the Area Chair take note of the potential plagiarism concerns regarding this paper.
>
> I suspect that this paper may have plagiarized the work of [1] Zheng, Jianbin, et al. "Trajectory Consistency Distillation: Improved Latent Consistency Distillation by Semi-Linear Consistency Function with Trajectory Mapping." arXiv preprint arXiv:2402.19159 (2024). https://arxiv.org/abs/2402.19159v2
>
> In accordance with Harvard’s Plagiarism Policy(https://usingsources.fas.harvard.edu/what-constitutes-plagiarism-0), this paper may be engaging in ***Inadequate Paraphrase*** and ***Uncited Paraphrasing***.
>
>
> The reasons for my concerns are as follows:
>
>    1. In the manuscript submitted by the author, from lines L134 to L161, the author asserts that their proposed parameterization is one of their core contributions. However, the equations presented, specifically Eq. 2 and 3, are identical to Eqs. 19 and 20 in [1]. Moreover, [1] was the first to apply Eq. 19 to the task of latent consistency distillation in text to image diffusion models. The description L134 to L161 also significantly overlaps with the parameterization discussion in Sec 4.2 of [1].
>    2. The objective function detailed in lines L62 to L178 of the manuscript are consistent with Eq. 29 in [1], and the content and explanation in Sec 4.2 “training” of [1] closely resemble the author’s manuscript.
>    3.	The author frequently references [2] CTM, which conducted extensive experiments on CIFAR and ImageNet datasets during the experimental phase. However, the author’s manuscript does not mention these tasks in experiment at all. In their rebuttal, the author mentioned lacking sufficient computational resources to conduct experiments related to [2] CTM, which is particularly confusing to me. Furthermore, the tasks in the author’s manuscript are identical to those in [1].
>
> And obviously, given the knowledge of [1], the innovation of this paper is significantly weakened both in task and experimental results.
>
> ***I am not able to make a definitive judgment at this stage. However, based on the current evidence, I believe this paper may be suspected of plagiarism.***
>
> ------
> Here we paste the definition of Uncited Paraphrasing and Inadequate Paraphrase here:
>
> 1. Inadequate Paraphrase: When you paraphrase, your task is to distill the source's ideas in your own words. It's not enough to change a few words here and there and leave the rest; instead, you must completely restate the ideas in the passage in your own words. If your own language is too close to the original, then you are plagiarizing, even if you do provide a citation.
>
> In order to make sure that you are using your own words, it's a good idea to put away the source material while you write your paraphrase of it. This way, you will force yourself to distill the point you think the author is making and articulate it in a new way. Once you have done this, you should look back at the original and make sure that you have represented the source’s ideas accurately and that you have not used the same words or sentence structure. If you do want to use some of the author's words for emphasis or clarity, you must put those words in quotation marks and provide a citation.
>
> 2. Uncited Paraphrasing: When you use your own language to describe someone else's idea, that idea still belongs to the author of the original material. Therefore, it's not enough to paraphrase the source material responsibly; you also need to cite the source, even if you have changed the wording significantly. As with quoting, when you paraphrase you are offering your reader a glimpse of someone else's work on your chosen topic, and you should also provide enough information for your reader to trace that work back to its original form. The rule of thumb here is simple: Whenever you use ideas that you did not think up yourself, you need to give credit to the source in which you found them, whether you quote directly from that material or provide a responsible paraphrase.
>
>
> [1] Zheng, Jianbin, et al. "Trajectory consistency distillation: Improved Latent Consistency Distillation by Semi-Linear Consistency Function with Trajectory Mapping" arXiv preprint arXiv:2402.19159 (2024). https://arxiv.org/abs/2402.19159v2
>
> [2] Kim, Dongjun, et al. "Consistency trajectory models: Learning probability flow ode trajectory of diffusion." arXiv preprint arXiv:2310.02279 (2023). https://arxiv.org/abs/2310.02279v3

---

> ### Author Response · Authors · 2024-08-10
> **Thank you for your further response!**
>
> Thank you for your reply!
>
> We understand your concern on the paper: Trajectory Consistency Distillation: Improved Latent Consistency
> Distillation by Semi-Linear Consistency Function with Trajectory Mapping (TCD) [1].
>
>
>
> **Parameterization**
>
> The parameterization of PCM was induced from the analysis form of the exact solution provided in the paper DPM-Solver [2].  All the characters used in our paper and first-order approximation proof follow DPM-Solver (including Eq. 2 - 3).  We carefully checked the paper TCD and found it provided a similar discussion from DPM-Solver. It appeared on the arXiv earlier. We are willing to add citations of TCD in the relevant discussion.  Both PCM and TCD cite DPM-Solver.
>
> Additionally,  we hope to clarify that's just a initially basic part for our further discussion. Our further discussion on the parameterization including the analysis of $\boldsymbol \epsilon_{\theta} (\mathbf x_t, t) = \frac{\int_{\lambda_t}^{\lambda_s}e^{-\lambda} \boldsymbol \epsilon_{\boldsymbol \phi}(\mathbf x_{t_{\lambda}(\lambda)}, t_{\lambda}(\lambda))\mathrm d \lambda}{\int_{\lambda_t}^{\lambda_s} e^{-\lambda} \mathrm d\lambda} $ and relevant proofs including, `error bound`, `CFG`, `adversarial loss`, have no intersections with TCD.
>
> **Training objective**
>
> The objective design of PCM is `totally different` from TCD.
>
> For equation 29 in TCD, please note that the loss of TCD accepts an additional timestep  `$t_m$`  for training, which is equivalent to the `$s$`  in the third row of following table. That is, they have to `randomly choose` the target timestep. While for PCM, we fixed the target timestep (i.e., $s:=s_{m}$) before training and `set them to pre-defined`. `This is exactly the core difference between PCM and TCD, CTM [3]`. Considering your concern, we are also willing to run additional comparison experiments with the official TCD and have a specific discussion on it in our paper.
>
> We list the loss design comparison in the following table
>
> | Methods | timesteps sampled                                            | adjacent timesteps | target timesteps                                             |
> | ------- | ------------------------------------------------------------ | ------------------ | ------------------------------------------------------------ |
> | CM      | $t_{n+1}\sim\textrm{Uniform}(\{t_1, t_2, \dots\})$           | $t_{n}$            | $s:=t_1:=0$ (`Fixed`)                                        |
> | CTM     | $t_{n+1}\sim\textrm{Uniform}(\{t_1, t_2, \dots\})$           | $t_{n}$            | $s\sim\textrm{Uniform}(\{t_1, t_2, \dots, t_{n}\})$ (`Random`) |
> | TCD     | $t_{n+1}\sim\textrm{Uniform}(\{t_1, t_2, \dots\})$ (Sampled from all possible timesteps) | $t_{n}$            | $s\sim\textrm{Uniform}(\{t_1, t_2, \dots, t_{n}\})$ (`Random`) |
> | PCM     | $t_{n+1}\sim\textrm{Uniform}(\{s_m, \dots, s_{m+1}\})$ (Sampled within pre-defined phases) | $t_{n}$            | $s := s_{m}$ (`Fixed`,  no target timestep embedding)        |
>
> From a more high-level perspective, as the sentence  "enabling `seamless transitions at any point` along the trajectory governed by the PF ODE"  stated in TCD,  CTM and TCD all try to learn arbitrary pairs.
>
> While our method PCM `only learns transition within each sub-phase`, which not only easy to train but also does not require any target timestep embeddings.
>
>
> **New Experiments**
>
> Our paper is originally for high-resolution text-conditioned image and video generation, where we try to tackle the limitations of LCM in text-to-image generation and text-to-video generation, as stated in our paper. We did not test on the CIFAR and ImageNet since they are for unconditional or class-conditional low-resolution generation. For your request, we have tried our best to run as many experiments as possible on the CIFAR and ImageNet. Following the official CM,  the ImageNet experiments require training on 64 A100 GPUs for a week, which we can not afford in the limited rebuttal period.
>
> [1] Trajectory consistency distillation
>
> [2] Dpm-solver: A fast ode solver for diffusion probabilistic model sampling in around 10 steps
>
> [3] Consistency trajectory models: Learning probability flow ode trajectory of diffusion

---

### Official Review · Reviewer_tuze · 2024-07-14

**Soundness:** 4
**Presentation:** 3
**Contribution:** 3
**Rating:** 8
**Confidence:** 4

**Summary:**

The paper titled "Phased Consistency Model" (PCM) introduces a novel model designed to address the limitations of Latent Consistency Models (LCMs) in high-resolution, text-conditioned image generation. The authors identify three primary flaws in LCMs: inconsistency, controllability, and efficiency. PCM is proposed to generalize the design space of LCMs and overcome these limitations by segmenting the ODE trajectory into multiple sub-trajectories and enforcing self-consistency within each sub-trajectory. This approach allows for more stable and deterministic sampling, better control over image generation with larger CFG values, and improved quality in low-step settings through an adversarial loss in the latent space. Extensive experiments demonstrate PCM’s superior performance in both image and video generation tasks compared to existing methods.

**Strengths:**

1. **Technical contribution**: PCM introduces a novel approach of segmenting the ODE trajectory, which is a significant advancement over traditional LCMs. This phasing approach effectively tackles the identified limitations of LCMs.

2. **Comprehensive Evaluation**: The paper provides extensive experimental results on widely recognized benchmarks, showing PCM's superiority in both image and video generation tasks across various settings.

3. **Versatility**: PCM’s methodology is versatile, supporting multi-step refinement and enabling state-of-the-art few-step text-to-video generation.

4. **Clear Illustrations and Comparisons**: Figures and tables in the paper clearly illustrate the performance improvements and qualitative differences between PCM and other models, aiding in understanding the advantages of PCM.

**Weaknesses:**

No specific weakness to mention.

**Questions:**

- Could the author provide some other human related evaluation metrics, e.g., ImageReward, Aesthetic scores, etc, to compare the generation quality within different efficient diffusion models?
- Given the introduction of an adversarial loss, how stable is the training process of PCM? Are there any observed issues with training stability or convergence, and how are they addressed?

**Limitations:**

The authors have demonstrated the limitations, e.g., inability in 1-step generation, in their paper.

---

> ### Author Rebuttal · Authors · 2024-08-06
>
> Thank you for your very positive review, and constructive suggestions.
>
> Q1: In terms of human evaluation metrics, we have re-evaluated the generation results of our method alongside all the comparative baselines mentioned in the paper. This re-evaluation was conducted over varying steps: 1, 2, 4, 8, and 16, using a range of well-established human evaluation metrics such as HPS-v2 [1], PickScore [2], and Laion Aesthetic Score [3]. Our method consistently achieved the best or at least comparable performance across all tested settings, notably outperforming the consistency model baseline LCM.
>
> Experiments on SD v1-5
>
> | Steps | Methods           | HPS       | AES       | PICKSCORE |
> | ----- | ----------------- | --------- | --------- | --------- |
> | 50    | SD v1-5 (Teacher) | 0.277     | 5.389     | 0.218     |
> | 1     | INSTAFLOW         | 0.267     | 5.010     | 0.207     |
> | 1     | SD TURBO          | 0.276 (1) | 5.445 (1) | 0.223 (1) |
> | 1     | CTM               | 0.240     | 5.155     | 0.195     |
> | 1     | LCM               | 0.251     | 5.178     | 0.201     |
> | 1     | PCM               | 0.276 (1) | 5.389 (2) | 0.213 (2) |
> | 2     | INSTAFLOW         | 0.249     | 5.050     | 0.196     |
> | 2     | SD TURBO          | 0.278 (1) | 5.570 (1) | 0.226 (1) |
> | 2     | CTM               | 0.267     | 5.117     | 0.208     |
> | 2     | LCM               | 0.266     | 5.135     | 0.210     |
> | 2     | PCM               | 0.275 (2) | 5.370 (2) | 0.217 (2) |
> | 4     | INSTAFLOW         | 0.243     | 4.765     | 0.192     |
> | 4     | SD TURBO          | 0.278 (2) | 5.537 (1) | 0.224 (1) |
> | 4     | CTM               | 0.274     | 5.189     | 0.213     |
> | 4     | LCM               | 0.273     | 5.264     | 0.215     |
> | 4     | PCM               | 0.279 (1) | 5.412 (2) | 0.217 (2) |
> | 8     | INSTAFLOW         | 0.267     | 4.548     | 0.189     |
> | 8     | SD TURBO          | 0.276 (2) | 5.390 (2) | 0.221 (1) |
> | 8     | CTM               | 0.271     | 5.026     | 0.210     |
> | 8     | LCM               | 0.274     | 5.366     | 0.216     |
> | 8     | PCM               | 0.278 (1) | 5.398 (1) | 0.218 (2) |
> | 16    | INSTAFLOW         | 0.237     | 4.437     | 0.187     |
> | 16    | SD TURBO          | 0.277 (1) | 5.275     | 0.219 (1) |
> | 16    | CTM               | 0.270     | 4.870     | 0.209     |
> | 16    | LCM               | 0.274     | 5.352 (2) | 0.216     |
> | 16    | PCM               | 0.277 (1) | 5.442 (1) | 0.217 (2) |
>
> Experiments on SD XL
>
> | Steps   | Methods        | HPS       | AES       | PICKSCORE |
> | ------- | -------------- | --------- | --------- | --------- |
> | 1-step  | SDXL Lightning | 0.278     | 5.65 (1)  | 0.223     |
> | 1-step  | SDXL TURBO     | 0.279 (1) | 5.40      | 0.228 (1) |
> | 1-step  | CTM            | 0.239     | 4.86      | 0.201     |
> | 1-step  | LCM            | 0.205     | 5.04      | 0.2006    |
> | 1-step  | PCM            | 0.28 (1)  | 5.62 (2)  | 0.225 (2) |
> | 2-step  | SDXL Lightning | 0.28      | 5.72 (1)  | 0.227 (1) |
> | 2-step  | SDXL TURBO     | 0.281 (2) | 5.46      | 0.226 (2) |
> | 2-step  | CTM            | 0.267     | 5.58      | 0.216     |
> | 2-step  | LCM            | 0.265     | 5.40      | 0.217     |
> | 2-step  | PCM            | 0.282 (1) | 5.688 (2) | 0.225     |
> | 4-step  | SDXL Lightning | 0.281     | 5.76 (2)  | 0.228 (1) |
> | 4-step  | SDXL TURBO     | 0.284 (1) | 5.49      | 0.224     |
> | 4-step  | CTM            | 0.278     | 5.84 (1)  | 0.221     |
> | 4-step  | LCM            | 0.274     | 5.48      | 0.223     |
> | 4-step  | PCM            | 0.284 (1) | 5.645     | 0.228 (2) |
> | 8-step  | SDXL Lightning | 0.282     | 5.75 (2)  | 0.229 (1) |
> | 8-step  | SDXL TURBO     | 0.283 (2) | 5.59      | 0.225     |
> | 8-step  | CTM            | 0.276     | 5.88 (1)  | 0.218     |
> | 8-step  | LCM            | 0.277     | 5.57      | 0.223     |
> | 8-step  | PCM            | 0.285 (1) | 5.676     | 0.229 (2) |
> | 16-step | SDXL Lightning | 0.28 (2)  | 5.72 (2)  | 0.225 (2) |
> | 16-step | SDXL TURBO     | 0.277     | 5.56      | 0.219     |
> | 16-step | CTM            | 0.274     | 5.85 (1)  | 0.215     |
> | 16-step | LCM            | 0.276     | 5.64      | 0.221     |
> | 16-step | PCM            | 0.284 (1) | 5.646     | 0.228 (1) |
>
> Q2:  Regarding the training stability,  we did not observe any training instability, even when using a very small batch size. For example, we trained the SDXL version of PCM with a batch size of 16 using fp16 mixed precision and achieved comparable performance to SDXL-Lightning one-step, which uses a batch size of 512 (32 times larger). We are willing to submit our training code for SD v1-5, SDXL, and SD3 if needed. All models showed clear improvements within 2,000 iterations and consistently improved with longer training times. We believe this stability is due to our specially designed adversarial loss rather than the standard GAN loss.  However, as shown in our ablation study in Fig. 7, when replacing our proposed adversarial loss with normal GAN loss, we observe unstable training. We analyze the phenomenon from the perspective of distribution mismatch between pretraining datasets and distillation datasets in our paper (Theorem 5 & 6.).
>
> [1] Human preference score v2: A solid benchmark for evaluating human preferences of text-to-image synthesis.
>
> [2] Pick-a-pic: An open dataset of user preferences for text-to-image generation. NeurIPS 2023.
>
> [3] Laion-aesthetics.

---

> > ### Comment · Reviewer_tuze · 2024-08-11
> >
> > Thank you for the rebuttal. I will maintain my original score.

---

> > > ### Author Response · Authors · 2024-08-11
> > > **Sincerely thank you for the very positive review!**
> > >
> > > Thank you for maintaining the very positive score. We sincerely appreciate the time and effort you invested in reviewing our paper. Your feedback has been exceptionally valuable, and we are deeply grateful for your support.
> > >
> > > We are always here should you have any further questions or need additional information.
> > >
> > > Best regards,
> > > The Authors

---

### Official Review · Reviewer_sXTg · 2024-07-14

**Soundness:** 3
**Presentation:** 2
**Contribution:** 2
**Rating:** 6
**Confidence:** 3

**Summary:**

The paper investigates three issues of Consistency Models on Latent space, thereby making a proposal named Phased Consistency Model that handles these weaknesses, supported by theoretical proofs and derivations. To assess the efficacy of their proposed solution, the authors conduct extensive experiments on two main scenarios: text-to-image and text-to-video generation. Additionally, details ablation studies and analysis on negative prompting, the self-consistency property and adversarial design to provide a strong motivation for the proposed approach.

**Strengths:**

1. The method separates consistency objective by splitting the ODE trajectory into sub-trajectories, therefore it has better ability on preserving image consistency when varying inference steps in deterministic sampling. Theoretical proofs provide good bounds for these objectives.
2. Reasonable arguments come from theoretical perspectives to address the CFG scale problem of Consistency Models.
3. The proposed Adversarial Consistency Loss for distribution consistency distillation makes sense in design of discriminator’s architecture.

**Weaknesses:**

1. Pre-defined number of sub-trajectories seems to fix the number of inference steps, which reduces the model’s flexibility.
2. Lack of explanation for the design of formula of Adversarial Consistency Loss, which employs ReLU function.
3. Need an ablation study to investigate impact of Adversarial Consistency Loss, for example what is model’s behaviour and result without this loss?
4. Detailed hyper-parameters used in settings, for example preconditioning weight, Adversarial Consistency Loss weight.

**Questions:**

See the weakness above

**Limitations:**

The authors analyse the limitations of their method in case of 1-step generation in a few lines.

---

> ### Author Rebuttal · Authors · 2024-08-06
>
> Thank you for your recognition in our work and for providing constructive feedback on our paper.
>
> Q1: Pre-defined number of sub-trajectories seems to fix the number of inference steps.
>
> 1. Note that inherently, each sub-trajectory of PCM can be perceived as a normal CM. Therefore, within each sub-trajectory, we can still support more step refinement just like normal CM. Specifically,  our one-step model has the same flexibility as normal CM (**CM is a subset of PCM**). But for the best efficiency, we only apply one step for each sub-trajectory.
> 2. We show in the paper that except for one-step generation, all the other step models can be tuned through a small LoRA adapter with only 4% storage burden of original diffusion models. Therefore, users can switch the inference steps easily.
>
> Q2: ReLU Function
>
> This is a typo here, and it should be $\textrm{ReLU}(1 + D (\tilde{x}_s, c, s)) + \textrm{ReLU}(1 - D(\hat{x}_s, c, s))$, where $D$ is the discriminator, $c$ is the text conditions and $s$ is the timestep conditions. The $\textrm {ReLU}$ function is equivalent to normally applied hinge loss. We apologize if it caused any confusion.
>
> Q3: Ablation study on the adversarial loss.
>
> Please refer to Fig. 7 , Fig. 9, and Fig. 31 (In the supplementary) in our submission.  In Fig. 7, we show that the adversarial loss greatly improves the FID at low-step regime and coverages the similar performance of PCM without applying the adversarial loss when the number of phases grows. In Fig.9, we show the visual ablation study on the discriminator choice and the comparison of normal GAN loss and our proposed adversarial loss. In Fig. 31, we show the side-by-side 2-step generation results with and without applying the adversarial loss. In Fig.31, we can observe a clear visual quality improvement. We also conduct a human preference metric comparison in the following table. The introduced adversarial loss consistency improves the human evaluation metrics consistently across different inference steps.
>
> | Step | Methods     | HPS   | AES   | PICKSCORE |
> | ---- | ----------- | ----- | ----- | --------- |
> | 1    | PCM w/ adv  | 0.280 | 5.620 | 0.225     |
> |      | PCM w/o adv | 0.251 | 4.994 | 0.206     |
> | 2    | PCM w/ adv  | 0.282 | 5.688 | 0.225     |
> |      | PCM w/o adv | 0.275 | 5.502 | 0.220     |
> | 4    | PCM w/ adv  | 0.284 | 5.645 | 0.228     |
> |      | PCM w/o adv | 0.281 | 5.576 | 0.225     |
> | 8    | PCM w/ adv  | 0.285 | 5.676 | 0.229     |
> |      | PCM w/o adv | 0.283 | 5.637 | 0.227     |
> | 16   | PCM w/ adv  | 0.284 | 5.646 | 0.228     |
> |      | PCM w/o adv | 0.283 | 5.620 | 0.227     |
>
> Q4: Hyper-parameters
>
> For all training settings, we apply the loss as $\mathcal L_{\textrm{PCM}} + 0.1 \mathcal L_{\textrm {PCM}}^{adv} $. That is the $\lambda$ is set to $0.1$ in all training settings. For the $\lambda (t)$ in $\mathcal L_{\textrm{PCM}}$ (Equation 5), we use 1 for simplicity.

---

> > ### Comment · Reviewer_sXTg · 2024-08-12
> >
> > Thank you for addressing my concern. I decide to raise score to 6.

---

> ### Author Response · Authors · 2024-08-12
> **Sincerely thank you for raising the score!**
>
> We are grateful for your decision to raise the score. Your recognition and feedback mean a lot to us.
>
> Please feel free to reach out if there’s anything else we can assist with.
>
> Best regards,
>
> The Authors

---

### Author Rebuttal · Authors · 2024-08-06

We sincerely thank all reviewers for their valuable comments and suggestions. We are sincerely grateful to the reviewers for dedicating their time and effort to review our work.  We are delighted to see reviewers commenting on our paper with "detailed proofs", "solid and comprehensive experimental results", "clear and straight-forward illustrations", and "versatile approach".

---



In this rebuttal, we try our best to solve the concerns of reviewers. We summarize the important and common concerns in the following:

**Human Evaluation Metrics. (Reviewer tuze, Reviewer ukQK)**: We re-test the generation results of our method and all compared baselines listed in the paper across 1, 2, 4, 8, and 16 steps with widely applied human evaluation metrics including HPS-v2 [1],  PickScore [2], Laion Aesthetic Score [3]. Our method consistently achieves the best or comparable performance with compared methods in all compared settings, significantly surpassing the consistency model baseline LCM [4].

**Experiments on CIFAR [5] and ImageNet [6] for more comprehensive comparison. (Reviewer ukQK)**

We additionally implement the core idea "phasing" technique into CIFAR-10 (unconditional generation) and ImageNet (conditional generation). We try our best to run as many experiments as possible during the limited rebuttal period.

For CIFAR-10, we train our models with a batch-size 256 (512 used in official CM [7]) for 200, 000 iterations (800, 000 iterations used in official CM), without applying any additional GAN training. Our two-step results achieve FID 2.14 with an inception score of 9.76, significantly surpassing the CM (FID 2.93). Our four-step results achieve FID 2.04 with an inception score of 9.90, achieving comparable performance with 35-step results of EDM [8] (FID 2.04, inception score of 9.88).

For ImageNet, due to the limited time and GPU resources, we can only afford to run a tentative experiment. Specifically, we run a 4-phased PCM by adapting the official repo of CM with a normal L2 loss. For the last phase, we apply LPIPS loss. We first train the model for 100,000 iterations with a batch-size 64 and further train the model for 50, 000 iterations with a batch-size 1024. Note that the official CM train the model with batch-size 2048 for 800, 000 iterations. Therefore the current training budget for our method is 3.5% of the offical CM. However, our 4-step model achieves FID 3.7,  surpassing the 4-step results of official CM with LIPIPS loss (FID 4.3) and L2 loss (FID 5.2). We believe this is clear evidence of the effectiveness of the "Phasing" ODE trajectory.

We list a table in the following rebuttal for a more comprehensive and detailed comparison with CM, CTM [9], and other baselines.

**Training Stability. (Reviewer tuze)** We didn't observe any training instability even training our approach with a very small bach-size. For example, we train the SDXL version PCM with a batch-size 16 using fp16 mixed precisions but achieve comparable performance with SDXL-Lightning [10] one-step which uses batch size 512 (32 times larger). We are willing to submit our training code on SD v1-5, SDXL, and SD3 if needed. All models can observe clear improvements within 2,000 iterations and consistently improve as the training time is longer. We believe the stability can be attributed to our specially designed adversarial loss instead of normal GAN loss.

**Impact of Adversarial Loss. (Reviewer sXTg)**

Please refer to Fig. 7 , Fig. 9, and Fig. 31 (In the supplementary) in our submission.  In Fig. 7, we show that the adversarial loss greatly improves the FID at low-step regime and coverages the similar performance of PCM without applying the adversarial loss when the number of phases grows. In Fig. 9, we show the visual ablation study on the discriminator choice and the comparison of normal GAN loss and our proposed adversarial loss. In Fig. 31, we show the side-by-side 2-step generation results with and without applying the adversarial loss. In Fig. 31, we can observe a clear visual quality improvement. Our further quantitative human evaluation metrics in the following rebuttal also aligns with these visual example ablations.



Please refer to the following rebuttals for other specific concerns and more details. We looking forward to your further reply and discussion.

Sincerely,

Authors of submission 2513

---



[1] Human preference score v2: A solid benchmark for evaluating human preferences of text-to-image synthesis.

[2] Pick-a-pic: An open dataset of user preferences for text-to-image generation. NeurIPS 2023.

[3] Laion-aesthetics.

[4] Latent Consistency Models: Synthesizing High-Resolution Images with Few-Step Inference.

[5] Learning multiple layers of features from tiny images.

[6] Imagenet: A large-scale hierarchical image database. CVPR 2009.

[7] Consistency Models. ICML 2023.

[8] Elucidating the Design Space of Diffusion-Based Generative Models. NeurIPS 2022.

[9] Consistency Trajctory Models: Learning Probability Flow ODE. ICLR 2024.

[10] SDXL-Lightning: Progressive Adversarial Diffusion Distillation

---

### Author Response · Authors · 2024-08-11
**Regarding  the accusation of Reviewer ukQK**

Dear AC and Reviewers,

We are writing to **strongly** defend the plagiarism accusation of Reviewer ukQK. A plagiarism accusation is very serious and shall not be made based on the incomplete understanding. We defend Reviewer ukQK's plagiarism accusation on our paper as follows:

1. Reviewer ukQK thinks our Equations 2, 3 are copied from TCD. However, `our Equation 2 is a direct conclusion formula from the previous paper DPM-Solver [2]`, which we have cited. In our submission, we write the sentence as the following:

   > For the above-discussed PF-ODE, there exists an exact solution [Citation to DPM-Solver [2]] from timestep t and s,
   >
   > [Equation 2]

   In addition, our equation 3 is also a direct adaption of first-order approximation, which is equivalent to DDIM in implementation format. This is also a clear conclusion by DPM-Solver.

Both TCD and PCM clearly cited DPM-Solver [2].  The important conclusion of DPM-Solver (NeurIPS 2022) shall not be considered as contributions of TCD (arXiv 2024).

For the accusation from Line 134 to Line 161.

- From Line 134 to 139, it follows the standard definition of Consistency Models [1], which is not proposed by TCD.

- From Line 143 to 150, it follows the standard conclusion of DPM-Solver [2], which is not proposed by TCD.

- From Line 151 to 157, it is our clarification on the epsilon prediction meaning of consistency models. We do not find any relevant discussion in the TCD paper.

- From Line 158 to 161, we do find some similar discussion on the boundary condition in the TCD paper. But please note that it is a natural conclusion from the exact solution formula proposed in DPM-Solver.

Thereby, we believe the plagiarism accusation on our paper copying TCD's formulas is groundless.

---

> ### Author Response · Authors · 2024-08-11
>
> 2. Reviewer ukQK thinks we have the same training objective as TCD. Instead, our design is **totally and inherently different** from the training objective of TCD and CTM, which we have explained in our submission and the rebuttals.
>
> Firstly, from a high-level perspective, we enforce the model to follow self-consistency within the pre-defined sub-phases (i.e. Our ODE is firstly split into multiphases along the time axis, and then we learn CM within the sub-trajectory).  Therefore, our model enforces multiphase consistency. Instead, TCD just  follows CTM and learns to follow trajectory consistency. That is, they enforce their model to learn how to move from arbitrary initial timesteps to arbitrary other timesteps.
>
>  We provide a not strict but very straightforward example to better illustrate the difference. Assume, the model has 1000 timesteps (0, 1, ..., 999) in total and we hope to train a 4-step model of PCM. Then we have
>
>    The learning objectives of PCM:
>
>    $f(x_{999}, 999) =  x_{749}$
>
>    $f(x_{749}, 749) = x_{499}$
>
>    $f(x_{499}, 499) = x_{249}$
>
>    $f(x_{249}, 249) = x_0$
>
>    The learning objectives of TCD:
>
>    $ f(x_{999}, 999, 749) = x_{749} $
>
>    $ f(x_{999}, 999, 499) = x_{499} $
>
>    $ f(x_{999}, 999, 249) = x_{249} $
>
>    $ f(x_{999}, 999, 0) = x_0 $
>
>    $ f(x_{749}, 749, 499) = x_{499} $
>
>    $ f(x_{749}, 749, 249) = x_{249} $
>
>    $ f(x_{749}, 749, 0) = x_0 $
>
>    $ f(x_{499}, 499, 249) = x_{249} $
>
>    $ f(x_{499}, 499, 0) = x_0 $
>
>    $ f(x_{249}, 249, 0) = x_0 $
>
>  In the above example, PCM only maps one timestep to a predefined corresponding middle timestep (e.g., $999 \rightarrow 749$ ), which is pre-defined. That is, PCM is a one-to-one mapping. But for TCD, it has to accept an additional target timestep to indicate which timestep prediction it hopes to predict, and learn to predict all the other timesteps. That is, TCD is a one-to-N mapping. Therefore, PCM only has $O(N)$  training objectives but TCD has $\mathcal O(N + (N-1) + \cdots + 1) = \mathcal O(N^2)$.  This is also the direct reason why it requires a target timestep embedding.
>
> We list a more straightforward illustration of the differences of the loss functions,
>
> | Methods | timesteps sampled                                            | adjacent timesteps | target timesteps                                             |
> | ------- | ------------------------------------------------------------ | ------------------ | ------------------------------------------------------------ |
> | CM      | $t_{n+1}\sim\textrm{Uniform}(\{t_1, t_2, \dots\})$           | $t_{n}$            | $s:=t_1:=0$ (`Fixed`)                                        |
> | CTM     | $t_{n+1}\sim\textrm{Uniform}(\{t_1, t_2, \dots\})$           | $t_{n}$            | $s\sim\textrm{Uniform}(\{t_1, t_2, \dots, t_{n}\})$ (`Random`) |
> | TCD     | $t_{n+1}\sim\textrm{Uniform}(\{t_1, t_2, \dots\})$ (Sampled from all possible timesteps) | $t_{n}$            | $s\sim\textrm{Uniform}(\{t_1, t_2, \dots, t_{n}\})$ (`Random`) |
> | PCM     | $t_{n+1}\sim\textrm{Uniform}(\{s_m, \dots, s_{m+1}\})$ (Sampled within pre-defined phases) | $t_{n}$            | $s := s_m$ (`fixed`)                                                     |
>
> For the accusation of Equation 29 in TCD, please note that the loss of TCD accepts an additional timestep  `$t_m$`  for training, which is equivalent to the `$s$`  in the third row of the above table.  That is, TCD needs to `randomly choose` the target timestep. While for PCM, we fixed the target timestep (i.e., $s:=s_{m}$) before training and `set them to pre-defined`. `This is exactly the core difference between PCM and TCD, CTM`.
>
> Therefore, `we argue that the training design of PCM and TCD, CTM are inherently different and the reviewer ukQK might misunderstand key aspects of our method`.

---

> ### Author Response · Authors · 2024-08-11
>
> 3. In the following table, we summarize more differences between TCD and PCM.
>
> | Method | Learning Paradigm      | Adversarial Loss | 1-Step Generation | Video Generation | Sampling Method | CFG Analysis | Consistent Generation | Num  Traing Objects | Prediction  | FID / NFE                               |
> | ------ | ---------------------- | ---------------- | ----------------- | ---------------- | --------------- | ------------ | --------------------- | ------------------- | ----------- | --------------------------------------- |
> | TCD    | Trajectory Consistency | No               | No                | No               | Gamma-Sampling  | No           | No                    | $\mathcal O(N^2)$   | 1-N mapping | Worse when NFE > 4                      |
> | PCM    | Multiphase Consistency | yes              | yes               | yes              | Deterministic   | Yes          | Yes                   | $\mathcal O(N)$     | 1-1 mapping | Continue to be better as the NFE larger |
>
>
>
> In addition to the above accusation of plagiarism, we find the reviewer also made judgments on our method that we believed to be unfair.
>
> 4. The model of CTM requires a target timestep embedding. But the reviewer asked us the question `Why does CTM need target timestep embeddings`. It is clearly stated in the paper of CTM and also shown in the official code of CTM. Besides, we also set up an independent section to discuss why CTM needs target timestep embeddings (Ln 602 - 615, titled exactly "Why CTM Needs Target Timestep Embeddings?"). The comment might suggest that the reviewer may `have a gap` in understanding the related works and might not have fully grasped the details of our paper.
>
> 5. The motivation of our paper is to analyze the limitations of previous LCMs and generalize the design space to solve these limitations (as stated in the abstract, introduction, teaser figure). The LCM is only for high-resolution `text-conditioned` image and video generation. Therefore, we believe we have efficiently validate the effectiveness of PCM.
>
>    For the reviewers' request to run CIFAR and ImageNet which are low-resolution unconditional generation and class-conditional generation tasks and have no direct relations to LCM, we have tried our best to run as many experiments as possible and provide clear evidence of the superiority of PCM compared to CM and CTM in the fair comparison setting. We have validated our methods on  `five baselines` including CIFAR, ImageNet, SD v1-5, SDXL, and Text-to-Video. It is noting that all compared methods only validate their methods for `at most two baselines`, but the reviewer still claims that our method lacks validation.
>
> We believe the contribution of our paper is underestimated. Please kindly take the time to look at our detailed rebuttal.
>
>
>
> Thanks,
>
> Paper 2513 Authors
>
> [1] Consistency Models. ICML 2023
> [2] DPM-Solver: A Fast ODE Solver for Diffusion Probabilistic Model Sampling in Around 10 Steps. NeurIPS 2022

---

> ### Comment · Reviewer_ukQK · 2024-08-11
>
> I disagree with the author's current response.
>
> Given that the author was aware of [1], deliberately using the same reparameterization scheme without proper citation clearly constitutes plagiarism.
>
> I am **not questioning the innovation or contribution of this work**. I am simply questioning why [1] was not mentioned at all in this manuscript submitted by authors, despite the striking similarity of both submitted manuscript and [1]. This seems to go beyond just a lack of citation.
>
> Also I totally agree about `The important conclusion of DPM-Solver (NeurIPS 2022) shall not be considered as contributions of TCD (arXiv 2024).` But I also do believe TCD as the first work to apply a reparameterization method [2] for large-scale text-to-image distillation does exhibit a degree of innovation and contribution. (If using [2] for distillation doesn't qualify as contribution, then this reparameterization should have been placed in the *related work* section rather than the *method* section of the submitted manuscript.)
>
> Furthermore, the author's explanation is intriguing. `From Line 158 to 161, we do find some similar discussion on the boundary condition in the TCD paper. But please note that it is a natural conclusion from the exact solution formula proposed in DPM-Solver.`  I'm wondering if the author was aware of the TCD paper, does this not suggest the possibility of plagiarism, as I previously mentioned:  Uncited Paraphrasing and Inadequate Paraphrase?
>
> Lastly, I find it somewhat ironic that the author seems to suggest that if [1] cites [2], and their submitted work also cites [2], then their work can avoid citing [1], even though the author(s) know their work and [1] share substantial similarities in terms of tasks, comparisons, and experiments, while [2] is from an entirely different field compared to [1] and their work.
>
> [1] Trajectory consistency distillation
>
> [2] Dpm-solver: A fast ode solver for diffusion probabilistic model sampling in around 10 steps

---

> > ### Author Response · Authors · 2024-08-12
> >
> > **Parameterization**
> >
> > Our final parameterization is finally parameterized in the same format as DDIM for simplicity (as stated in the paper).
> >
> > For distillation-based acceleration, using the format of DDIM for distillation is a common practice. We list works using the format for large-scale text-to-image distillation in the following:
> >
> > 1. Progressive Distillation for Fast Sampling of Diffusion Models. ICLR 2022
> > 2. On Distillation of Guided Diffusion Models. CVPR 2023
> > 3. BOOT : Data-free Distillation of Denoising Diffusion Models with Bootstrapping. arXiv 2023
> >
> > We list the works using the format of DDIM for distillation in unconditional and class-conditional  in the following
> >
> > 1. TRACT: Denoising Diffusion Models with Transitive Closure Time-Distillation. arXiv 2023
> > 2. BOOT : Data-free Distillation of Denoising Diffusion Models with Bootstrapping. arXiv 2023
> >
> > Therefore, we believe It is not TCD (arXiv 2024) that first applied the parameterization for either large-scale text-to-image distillation or unconditional image generation and class-conditional generation.  And we would say it's not strange to potentially have a similar parameterization. The important thing is the discussion in the paper on how to understand and explain the parameterization, which we believe we have an entirely different discussion with TCD (Ln 151 -- 157).
> >
> > **Boundary condition**
> >
> > The DPM-Solver has shown that the denoising from $t$ to $s$ is to add an integral of weighted epsilon from $t$ to $s$. So it is obvious that the integral from $s$ to $s$ (i.e., the integral length is 0, $\int_{s}^s f = 0$, where $f$ can be any function) will equal 0. We believe this is an evident and natural conclusion and can be easily observed with minor exposure to calculus.  We admit we did not notice relevant discussion in TCD. As stated in our previous reply, we are willing to add citations to it.
> >
> > Sincerely,

---

### Decision · Program_Chairs · 2024-09-25

**Decision:**

Accept (poster)

**Comment:**

One reviewer recommended rejection and also flagged a potential plagiarism concern. Upon further investigation the AC did not find evidence of plagiarism.

Amongst the other reviewers, one recommended strong accept and the other recommended weak accept. As mentioned by these reviewers, the paper introduces a novel method of generating multiple self consistent sub-trajectories and achieves strong distillation performance on image generation while also showing applicability on video generation tasks. The AC agrees with these evaluations and recommends acceptance.